# Lightweight Dataset Pruning without Full Training via Example Difficulty and Prediction Uncertainty

Yeseul Cho [* 1]   Baekrok Shin [* 1]   Changmin Kang [1]   Chulhee Yun [1]

## Abstract

Recent advances in deep learning rely heavily on massive datasets, leading to substantial storage and training costs. Dataset pruning aims to alleviate this demand by discarding redundant examples. However, many existing methods require training a model with a full dataset over a large number of epochs before being able to prune the dataset, which ironically makes the pruning process more expensive than just training the model on the entire dataset. To overcome this limitation, we introduce the **Difficulty and Uncertainty-Aware Lightweight (DUAL)** score, which aims to identify important samples from the early training stage by considering both example difficulty and prediction uncertainty. To address a catastrophic accuracy drop at an extreme pruning ratio, we further propose a pruning ratio-adaptive sampling using Beta distribution. Experiments on various datasets and learning scenarios such as image classification with label noise and image corruption, and model architecture generalization demonstrate the superiority of our method over previous state-of-the-art (SOTA) approaches. Specifically, on ImageNet-1k, our method reduces the time cost for pruning to 66% compared to previous methods while achieving a SOTA 60% test accuracy at a 90% pruning ratio. On CIFAR datasets, the time cost is reduced to just 15% while maintaining SOTA performance. Implementation is available at `github/dual-pruning`.

## 1. Introduction

Advancements in deep learning have been significantly driven by large-scale datasets. However, recent studies have revealed a power-law relationship between the generalization capacity of deep neural networks and the size of their training data (Hestness et al., 2017; Rosenfeld et al., 2019; Gordon et al., 2021), meaning that the improvement of model performance becomes increasingly cost-inefficient as we scale up the dataset size.

Fortunately, Sorscher et al. (2022) demonstrate that the power-law scaling of error can be reduced to exponential scaling with Pareto optimal data pruning. The main goal of dataset pruning is to identify and retain the most informative samples while discarding redundant data points for training neural networks. This approach can alleviate storage and computational costs as well as training efficiency.

However, many existing pruning methods require training a model with a full dataset over a number of epochs to measure the importance of each sample, which ironically makes the pruning process more expensive than just training the model once on the original large dataset. For instance, several score-based methods (Toneva et al., 2018; Pleiss et al., 2020; Paul et al., 2021; He et al., 2024; Zhang et al., 2024) require training as they utilize the dynamics from the whole training process. Some geometry-based methods (Xia et al., 2022; Yang et al., 2024) leverage features from the penultimate layer of the trained model, therefore training a model is also required. Hybrid methods (Zheng et al., 2022; Maharana et al., 2023; Tan et al., 2025), which address the difficulty and diversity of samples simultaneously, still hold the same limitation as they use existing score metrics. Having to compute the dot product of learned features to get the neighborhood information makes them become even more expensive to utilize.

In order to address this issue, we introduce the **Difficulty and Uncertainty-Aware Lightweight** (**DUAL**) score, which can measure the importance of samples in the early stage of the training process by considering example difficulty and the prediction uncertainty. Additionally, for the high pruning ratio—when the selected subset is scarce—we propose **pruning-ratio-adaptive Beta sampling**, to intentionally include easier samples which have lower scores to achieve a better representation of the data distribution (Sorscher et al., 2022; Zheng et al., 2022; Acharya et al., 2024).

---

[*]Equal contribution  [1]Kim Jaechul Graduate School of AI, KAIST, Seoul, South Korea. Correspondence to: Chulhee Yun <chulhee.yun@kaist.ac.kr>.

*Proceedings of the 42$^{nd}$ International Conference on Machine Learning*, Vancouver, Canada. PMLR 267, 2025. Copyright 2025 by the author(s).

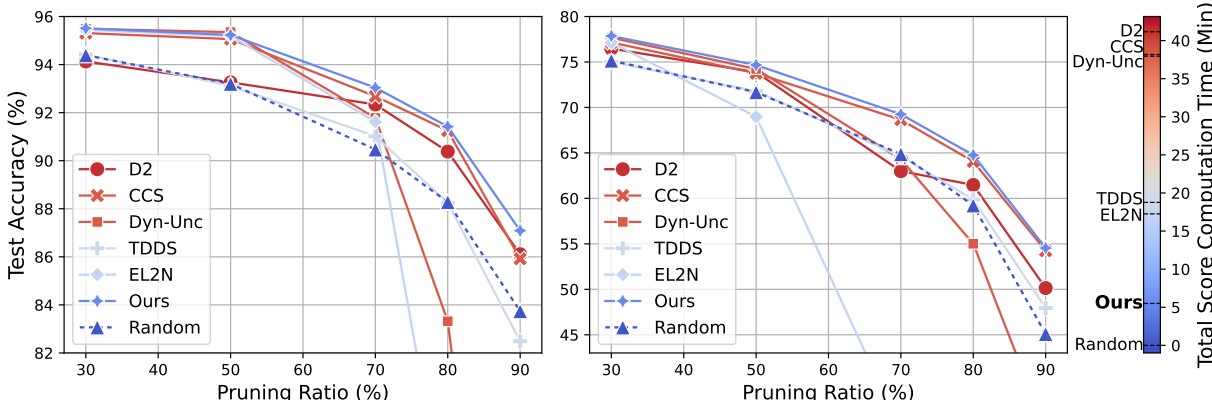

*Figure 1.* Test accuracy comparison on CIFAR datasets (**Left**: Results for CIFAR-10, **Right**: Results for CIFAR-100). The color represents the total computation time, including the time spent training the original dataset for score calculation, for each pruning method. Blue indicates lower computation time, while red indicates higher computation time. Our method demonstrates its ability to minimize computation time while maintaining SOTA performance.

Experiments conducted on CIFAR and ImageNet datasets under various learning scenarios verify the superiority of our method. Specifically, on ImageNet-1k, our method reduces the time cost to 66% compared to previous methods while achieving a SOTA performance, 60% test accuracy at the pruning ratio of 90%. On the CIFAR datasets, as illustrated in Figure 1, our method reduces the time cost to just 15% while maintaining SOTA performance. Especially, our proposed method shows a notable performance when the dataset contains noise.

## 2. Related Works

Data pruning aims to remove redundant examples, keeping the most informative subset of dataset, namely the coreset. Research in this area can be broadly categorized into two groups: *score-based* and *geometry-based* methods. Score-based methods define metrics to measure the importance of data points, then prioritize high-scoring samples. On the other hand, geometry-based methods focus on presenting a better representation of the true data distribution. Recent studies propose *hybrid* methods which incorporate the example difficulty score with the diversity of the coreset.

**Score-based.** EL2N (Paul et al., 2021) calculates L2 norms of the error vector, as an approximation of the gradient norm. Entropy (Coleman et al., 2020) quantifies the information contained in the predicted probabilities at the end of training. However, the outcomes of such "snapshot" methods need multiple runs to be stabilized, as shown in Figure 7, Appendix B. AUM (Pleiss et al., 2020) accumulates the gap between the target probability and the second-highest prediction probability. Methods that utilize training dynamics throughout the entire training offer more stable measures. Forgetting (Toneva et al., 2018) score counts the number of

forgetting events, where a correct prediction is flipped to a wrong prediction during training process. Dyn-Unc (He et al., 2024), which strongly inspired our approach, prioritizes the most uncertain samples rather than typical easy or hard samples during model training. The uncertainty is measured by the variation of predictions in a sliding window, and the score averages the variation throughout the whole training process. TDDS (Zhang et al., 2024) averages differences of Kullback-Leibler divergence loss of non-target probabilities for $T$ training epochs, where $T$ is highly dependent on the pruning ratio. The information from training dynamics proves useful for pruning because it allows one to differentiate hard but useful samples from noisy ones (He et al., 2024). However, despite its stability and effectiveness, previous methods fail to guarantee cost-effectiveness as they require at least one full training of the model on the entire dataset.

**Geometry-based.** Geometry-based methods concentrate on providing a better representation by minimizing the redundancy of selected samples. SSP (Sorscher et al., 2022) selects the samples most distant from k-means cluster centers, while Moderate (Xia et al., 2022) prefers samples near the median. However, these methods often compromise generalization performance, since they underestimate the effect of difficult examples.

Recently, hybrid approaches have emerged that harmonize both difficulty and diversity. CCS (Zheng et al., 2022) partitions difficulty scores into bins and selects an equal number of samples from each bin to ensure balanced representation. $\mathbb{D}^2$ (Maharana et al., 2023) employs a message-passing mechanism with a graph structure where nodes represent difficulty scores and edges encode neighboring representations, facilitating effective sample selection. BOSS (Acharya et al., 2024) introduces a Beta function for importance sampling

based on difficulty scores, which resembles our pruning ratio-adaptive sampling; we discuss the differences in Section 3.3. SIMS (Grosz et al., 2024) defines SIM score using class separability, data integrity, and model uncertainty, and then integrates sampling strategy. Our DUAL pruning defines a score metric with difficulty and uncertainty, and it becomes a hybrid approach when the score is combined with our proposed Beta sampling.

## 3. Proposed Methods

### 3.1. Preliminaries

Let $\mathcal{D} \coloneqq \{(\boldsymbol{x}_1, y_1), \cdots, (\boldsymbol{x}_n, y_n)\}$ be a labeled dataset of $n$ training samples, where $\boldsymbol{x} \in \mathcal{X} \subset \mathbb{R}^d$ and $y \in \mathcal{Y} \coloneqq \{1, \cdots, C\}$ are the data point and the label, respectively. $C$ is a positive integer and indicates the number of classes. For each labeled data point $(\boldsymbol{x}, y) \in \mathcal{D}$, denote $\mathbb{P}_k(y \mid \boldsymbol{x})$ as the prediction probability of $y$ given $\boldsymbol{x}$, for the model trained with $k$ epochs. Let $\mathcal{S} \subset \mathcal{D}$ be the subset retained after pruning. Pruning ratio $r$ is the ratio of the size of $\mathcal{D} \setminus \mathcal{S}$ to $\mathcal{D}$, or $r = 1 - \frac{|\mathcal{S}|}{|\mathcal{D}|}$.

The Dynamic Uncertainty (Dyn-Unc) score (He et al., 2024) prefers the most uncertain samples rather than easy-to-learn or hard-to-learn samples during model training. The uncertainty score is defined as the average of prediction variance throughout training. They first define the uncertainty in a sliding window of length $J$:

$$\mathrm{U}_k(\boldsymbol{x}, y) \coloneqq \sqrt{\frac{\sum_{j=0}^{J-1} \left[ \mathbb{P}_{k+j}(y \mid \boldsymbol{x}) - \bar{\mathbb{P}}_k \right]^2}{J - 1}} \qquad (1)$$

where $\bar{\mathbb{P}}_k \coloneqq \frac{\sum_{j=0}^{J-1} \mathbb{P}_{k+j}(y|\boldsymbol{x})}{J}$ is the average prediction of the model over the window $[k, k + J - 1]$. Then taking the average of the uncertainty throughout the whole training process leads to Dyn-Unc score:

$$\mathrm{U}(\boldsymbol{x}, y) = \frac{\sum_{k=1}^{T-J+1} \mathrm{U}_k(\boldsymbol{x}, y)}{T - J + 1}. \qquad (2)$$

### 3.2. Difficulty & Uncertainty-Aware Lightweight Score

Following the approach of Swayamdipta et al. (2020) and He et al. (2024), we analyze data points from ImageNet-1k based on the mean and standard deviation of predictions during training, as shown in Figure 2. We observe data points typically flow along the moon from bottom to top. Data points starting from the bottom left region with low prediction mean and low standard deviation move to the middle region with increased mean and standard deviation, and those starting at the middle region drift toward to the upper left region with high prediction mean and smaller standard deviation. This phenomenon is closely aligned with existing observations that neural networks typically learn

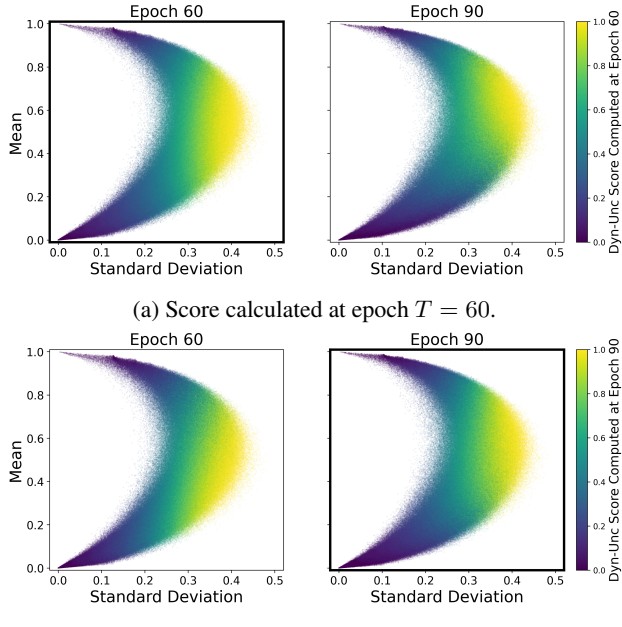

(a) Score calculated at epoch $T = 60$.

(b) Score calculated at epoch $T = 90$.

*Figure 2.* The left column ("Epoch 60") shows the prediction mean and standard deviation, computed using the predicted target probabilities up to epoch 60. The right column ("Epoch 90") shows corresponding values up to epoch 90. In each row, samples are colored by the normalized Dyn-Unc score computed at epoch 60 for Figure 2a and at epoch 90 for Figure 2b. The epoch at which the score was computed is indicated by a bold outline for each row.

easy samples first, then treat harder samples later (Bengio et al., 2009; Arpit et al., 2017; Jiang et al., 2020; Shen et al., 2022). In other words, we see that the uncertainty of easy samples rises first, and then more difficult samples start to move and show increased uncertainty score.

Figure 2 further gives a justification for this intuition. In Figure 2a, samples with the highest Dyn-Unc scores calculated at epoch 60 move upward by the end of training at epoch 90. This means that if we measure Dyn-Unc score at the early stage of training, it gives the highest scores to relatively easy samples rather than the most informative samples. It seems undesirable that it results in poor test accuracy on its coreset, as shown in Figure 8 in Appendix B.

To capture the most useful samples that are likely to contribute significantly to Dyn-Unc during the whole training process (of 90 epochs) at the earlier training stage (*e.g.* epoch of 60), we need to target the samples located near the bottom-right region of the moon-shaped distribution, as Figure 2b illustrates. Inspired by this observation, we design a scoring metric that identifies such samples by taking both the *uncertainty of the predictions* and the *prediction probability* into consideration.

Here, we propose the **Difficulty and Uncertainty-Aware**

**Lightweight (DUAL) score**, a measure that unites example difficulty and prediction uncertainty. We define the DUAL score of a data point $(\boldsymbol{x}, y)$ at $k \in [T - J + 1]$ as

$$\mathrm{DUAL}_k(\boldsymbol{x}, y) \coloneqq$$

$$\underbrace{\left(1 - \bar{\mathbb{P}}_k\right)}_{(a)} \underbrace{\sqrt{\frac{\sum_{j=0}^{J-1}\left[\mathbb{P}_{k+j}(y \mid \boldsymbol{x}) - \bar{\mathbb{P}}_k\right]^2}{J - 1}}}_{(b)} \quad (3)$$

where $\bar{\mathbb{P}}_k \coloneqq \frac{\sum_{j=0}^{J-1} \mathbb{P}_{k+j}(y|\boldsymbol{x})}{J}$ is the average prediction of the model over the window $[k, k + J - 1]$. Note that $\mathrm{DUAL}_k$ is the product of two terms: $(a)$ $1 - \bar{\mathbb{P}}_k$ quantifies the example difficulty averaged over the window; $(b)$ is the standard deviation of the prediction probability over the same window, estimating the prediction uncertainty.

Finally, the DUAL score of $(\boldsymbol{x}, y)$ is defined as the mean of $\mathrm{DUAL}_k$ scores over all windows:

$$\mathrm{DUAL}(\boldsymbol{x}, y) = \frac{\sum_{k=1}^{T-J+1} \mathrm{DUAL}_k(\boldsymbol{x}, y)}{T - J + 1}. \quad (4)$$

The DUAL score reflects training dynamics by leveraging prediction probability across several epochs, providing a reliable estimation to identify the most uncertain examples.

A theoretical analysis of a toy example further verifies the intuition above. Consider a linearly separable binary classification task $\{(\boldsymbol{x}_i \in \mathbb{R}^n, y_i \in \{\pm 1\})\}_{i=1}^N$, where $N = 2$ with $\|\boldsymbol{x}_1\| \ll \langle \boldsymbol{x}_1, \boldsymbol{x}_2 \rangle < \|\boldsymbol{x}_2\|$. Without loss of generality, we set $y_1 = y_2 = +1$. A linear classifier, $f(\boldsymbol{x}; \boldsymbol{w}) = \boldsymbol{w}^\top \boldsymbol{x}$, is employed as the model in our analysis. The parameter $\boldsymbol{w}$ is initialized at zero and updated by gradient descent. Soudry et al. (2018) prove that the parameter of linear classifiers diverges to infinity, but directionally converges to the $L_2$ maximum margin separator. This separator is determined by the support vectors closest to the decision boundary. If a valid pruning method encounters this task, then it should retain the point closer to the decision boundary, which is $\boldsymbol{x}_1$ in our case, and prune $\boldsymbol{x}_2$. Due to its large norm, $\boldsymbol{x}_2$ exhibits higher score values in the early training stage, for both Dyn-Unc and DUAL scores. It takes some time for the model to make prediction on $\boldsymbol{x}_1$ with large confidence to increase its uncertainty level as well as prediction mean, and the scores for $\boldsymbol{x}_1$ eventually become larger than those for $\boldsymbol{x}_2$ as training proceeds. In Theorem 3.1, we show through a rigorous analysis that the moment of such a flip in order happens strictly earlier for DUAL than for uncertainty.

**Theorem 3.1** (Informal). *Define $\sigma(z) \coloneqq (1 + e^{-z})^{-1}$. Let $S_{t;J}^{(i)}$ be the standard deviation and $\mu_{t;J}^{(i)}$ be the mean of $\sigma(f(\boldsymbol{x}_i; \boldsymbol{w}_t))$ within a window from time $t$ to $t + J - 1$. Denote $T_v$ and $T_{vm}$ as the first time when $S_{t;J}^{(1)} > S_{t;J}^{(2)}$ and $S_{t;J}^{(1)}(1 - \mu_{t;J}^{(1)}) > S_{t;J}^{(2)}(1 - \mu_{t;J}^{(2)})$ occurs, respectively. If the learning rate is small enough, then $T_{vm} < T_v$.*

Technical details about Theorem 3.1 are provided in Appendix D, together with an empirical verification of the time-efficiency of DUAL pruning over Dyn-Unc.

Empirically, as shown in Figure 3, the DUAL score targets data points in the bottom-right region during the early training phase, which eventually evolve to the middle-rightmost part by the end of training. This verifies that DUAL pruning identifies the most uncertain region faster than Dyn-Unc *both in theory and practice*. The differences arise from the additional consideration of example difficulty in our method. We believe this adjustment leads to improved generalization performance compared to Dyn-Unc, as verified through various experiments in later sections.

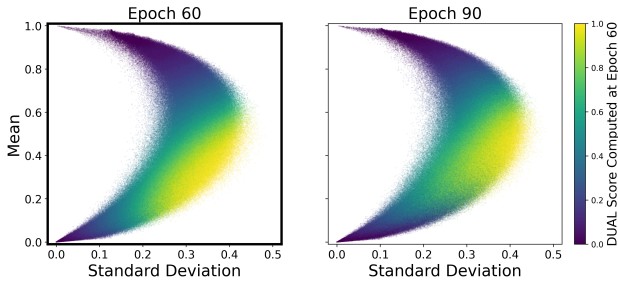

*Figure 3.* Our DUAL score targets similar uncertain samples in the early epoch of 60 (highlighted in bold). Selected samples are finally located in the most uncertain region when the whole training processes are considered.

However, score-based approaches including our method, suffer from accuracy drop at the high pruning ratio due to biased representations. To address this, we propose an additional sampling strategy that adaptively selects samples regarding the coreset size.

### 3.3. Pruning Ratio-Adaptive Sampling

Since the distribution of difficulty scores is dense in high-score samples, selecting only the highest-score samples may result in a biased model (Zhou et al., 2023; Maharana et al., 2023; Choi et al., 2024). To address this, we design a sampling method to determine the subset $\mathcal{S} \subset \mathcal{D}$, rather than simply pruning the samples with the lowest scores. We introduce a Beta distribution that varies with the pruning ratio. The primary objective of this method is to ensure that the selected subsets gradually include more easy samples into the coreset as the pruning ratio increases.

However, the concepts of "easy" and "hard" cannot be distinguished solely based on uncertainty or DUAL score. To address this, we use the *prediction mean* again for sampling. We utilize the Beta probability density function (PDF) to define the selection probability of each sample. First, we assign each data point a corresponding PDF value based

*Table 1.* Comparison of test accuracy between the DUAL score method and existing coreset selection techniques using ResNet-18 on CIFAR-10 and CIFAR-100 datasets. Training the model on the full dataset achieves an average test accuracy of 95.30% on CIFAR-10 and 78.91% on CIFAR-100. The best result in each pruning ratio is highlighted in bold.

| Dataset ($\rightarrow$) | CIFAR10 | | | | | CIFAR100 | | | | |
|---|---|---|---|---|---|---|---|---|---|---|
| Pruning Rate ($\rightarrow$) | 30% | 50% | 70% | 80% | 90% | 30% | 50% | 70% | 80% | 90% |
| Random | 94.39$_{\pm 0.23}$ | 93.20$_{\pm 0.12}$ | 90.47$_{\pm 0.17}$ | 88.28$_{\pm 0.17}$ | 83.74$_{\pm 0.21}$ | 75.15$_{\pm 0.28}$ | 71.68$_{\pm 0.31}$ | 64.86$_{\pm 0.39}$ | 59.23$_{\pm 0.62}$ | 45.09$_{\pm 1.26}$ |
| Entropy | 93.48$_{\pm 0.06}$ | 92.47$_{\pm 0.17}$ | 89.54$_{\pm 0.18}$ | 88.53$_{\pm 0.19}$ | 82.57$_{\pm 0.36}$ | 75.20$_{\pm 0.25}$ | 70.90$_{\pm 0.35}$ | 61.70$_{\pm 0.47}$ | 56.24$_{\pm 0.51}$ | 42.25$_{\pm 0.39}$ |
| Forgetting | 95.48$_{\pm 0.14}$ | 94.94$_{\pm 0.21}$ | 89.55$_{\pm 0.65}$ | 75.47$_{\pm 1.27}$ | 46.64$_{\pm 1.90}$ | 77.52$_{\pm 0.26}$ | 70.93$_{\pm 0.37}$ | 49.66$_{\pm 0.20}$ | 39.09$_{\pm 0.41}$ | 26.87$_{\pm 0.73}$ |
| EL2N | 95.44$_{\pm 0.06}$ | 95.19$_{\pm 0.11}$ | 91.62$_{\pm 0.14}$ | 74.70$_{\pm 0.45}$ | 38.74$_{\pm 0.75}$ | 77.13$_{\pm 0.23}$ | 68.98$_{\pm 0.35}$ | 34.59$_{\pm 0.48}$ | 19.52$_{\pm 0.79}$ | 8.89$_{\pm 0.28}$ |
| AUM | 90.62$_{\pm 0.09}$ | 87.26$_{\pm 0.11}$ | 81.28$_{\pm 0.26}$ | 76.58$_{\pm 0.35}$ | 67.88$_{\pm 0.53}$ | 74.34$_{\pm 0.14}$ | 69.57$_{\pm 0.21}$ | 61.12$_{\pm 0.20}$ | 55.80$_{\pm 0.33}$ | 45.00$_{\pm 0.37}$ |
| Moderate | 94.26$_{\pm 0.09}$ | 92.79$_{\pm 0.09}$ | 90.45$_{\pm 0.21}$ | 88.90$_{\pm 0.17}$ | 85.52$_{\pm 0.29}$ | 75.20$_{\pm 0.25}$ | 70.90$_{\pm 0.35}$ | 61.70$_{\pm 0.47}$ | 56.24$_{\pm 0.51}$ | 42.25$_{\pm 0.39}$ |
| Dyn-Unc | 95.49$_{\pm 0.21}$ | **95.35**$_{\pm 0.12}$ | 91.78$_{\pm 0.65}$ | 83.32$_{\pm 0.94}$ | 59.67$_{\pm 1.79}$ | 77.67$_{\pm 0.14}$ | 74.23$_{\pm 0.22}$ | 64.30$_{\pm 0.13}$ | 55.01$_{\pm 0.55}$ | 34.57$_{\pm 0.69}$ |
| TDDS | 94.42$_{\pm 0.13}$ | 93.11$_{\pm 0.14}$ | 91.02$_{\pm 0.19}$ | 88.25$_{\pm 0.24}$ | 82.49$_{\pm 0.28}$ | 75.02$_{\pm 0.37}$ | 71.80$_{\pm 0.33}$ | 64.61$_{\pm 0.24}$ | 59.88$_{\pm 0.21}$ | 47.93$_{\pm 0.21}$ |
| CCS | 95.31$_{\pm 0.22}$ | 95.06$_{\pm 0.15}$ | 92.68$_{\pm 0.17}$ | 91.25$_{\pm 0.21}$ | 85.92$_{\pm 0.39}$ | 77.15$_{\pm 0.28}$ | 73.83$_{\pm 0.21}$ | 68.65$_{\pm 0.31}$ | 64.06$_{\pm 0.21}$ | 54.23$_{\pm 0.48}$ |
| D2 | 94.13$_{\pm 0.20}$ | 93.26$_{\pm 0.16}$ | 92.34$_{\pm 0.18}$ | 90.38$_{\pm 0.34}$ | 86.11$_{\pm 0.21}$ | 76.47$_{\pm 0.29}$ | 73.88$_{\pm 0.28}$ | 62.99$_{\pm 0.28}$ | 61.48$_{\pm 0.34}$ | 50.14$_{\pm 0.90}$ |
| DUAL | 95.25$_{\pm 0.17}$ | 94.95$_{\pm 0.22}$ | 91.75$_{\pm 0.98}$ | 82.02$_{\pm 1.85}$ | 54.95$_{\pm 0.42}$ | 77.43$_{\pm 0.18}$ | 74.62$_{\pm 0.47}$ | 66.41$_{\pm 0.52}$ | 56.57$_{\pm 0.57}$ | 34.38$_{\pm 1.39}$ |
| DUAL+$\beta$ sampling | **95.51**$_{\pm 0.06}$ | 95.23$_{\pm 0.08}$ | **93.04**$_{\pm 0.43}$ | **91.42**$_{\pm 0.35}$ | **87.09**$_{\pm 0.36}$ | **77.86**$_{\pm 0.12}$ | **74.66**$_{\pm 0.12}$ | **69.25**$_{\pm 0.22}$ | **64.76**$_{\pm 0.23}$ | **54.54**$_{\pm 0.09}$ |

on its prediction mean and weight this probability using the DUAL score. The weighted probability with the DUAL score is then normalized to the range $[0, 1]$, then used as the sampling probability. We clarify that sampling probability is for selecting samples, *not for pruning*. Therefore, for each pruning ratio $r$, we randomly select $(1 - r) \cdot n$ samples without replacement, where sampling probabilities are given according to the prediction mean and DUAL score as described. The detailed algorithm for our proposed pruning method is provided in Algorithm 1, Appendix C.

We design the Beta PDF function to assign a sampling probability concerning a prediction mean as follows:

$$\beta_r = C \cdot (1 - \mu_{\mathcal{D}})(1 - r^{c_{\mathcal{D}}})$$
$$\alpha_r = C - \beta_r, \tag{5}$$

where $C > 0$ is a fixed constant, and the $\mu_D$ stands for the prediction mean of the highest score sample. Recalling that the mean of Beta distribution is $\frac{\alpha_r}{\alpha_r + \beta_r}$, the above choice makes the mean of Beta distribution moves progressively with $r$, starting from $\mu_{\mathcal{D}}$ ($r \simeq 0$, small pruning ratio) to one. In other words, with growing $r$, this Beta distribution becomes skewed towards the easier region ($r \rightarrow 1$, large pruning ratio), which in turn gives more weight to easy samples.

The tendency of evolving should be different with datasets, thus a hyperparameter $c_{\mathcal{D}} \geq 1$ is used to control the rate of evolution of the Beta distribution. Specifically, the choice of $c_{\mathcal{D}}$ depends on the complexity of the initial dataset. For smaller and more complex datasets, setting $c_{\mathcal{D}}$ to a smaller value retains more easy samples. For larger and simpler datasets, setting $c_{\mathcal{D}}$ to a larger value allows more uncertain samples to be selected. (For your intuitive understanding, please refer to Figure 20 and Figure 21 in the Appendix C.) This is also aligned with the previous findings from Sorscher

et al. (2022); if the initial dataset is small, the coreset is more effective when it contains easier samples, while for a relatively large initial dataset, including harder samples can improve generalization performance. More descriptions for our Beta sampling are provided in Appendix C.

***Remark.*** BOSS (Acharya et al., 2024) also uses the Beta distribution to sample easier data points during pruning, similar to our approach. However, a key distinction lies in how we define the Beta distribution's parameters, $\alpha_r$ and $\beta_r$. While BOSS adjusts these parameters to make the mode of the Beta distribution's PDF scale linearly with the pruning ratio $r$, we employ a non-linear combination. This non-linear approach has the crucial advantage of maintaining an almost stationary PDF at low pruning ratios. This stability is especially beneficial when the dataset becomes easier where there is no need to focus on easy examples. Furthermore, unlike previous methods, we define PDF values based on the prediction mean, rather than any difficulty score, which is another significant difference.

***Remark.*** SIMS (Grosz et al., 2024) also proposes a ratio-adaptive sampling strategy, applying importance weights over the original score distribution. However, it assumes a normal distribution of scores, which does not hold in practice (see Figure 2 of (Grosz et al., 2024)). In contrast, our sampling method, by not relying on any specific score distribution, remains robust across diverse datasets.

## 4. Experiments

### 4.1. Experimental Settings

We assessed the performance of our proposed method in three key scenarios: image classification, image classification with noisy labels and corrupted images. In addition, we validate cross-architecture generalization on three-layer CNN, VGG-16 (Simonyan & Zisserman, 2015), ResNet-18 and ResNet-50 (He et al., 2015).

**Hyperparameters** For training CIFAR-10 and CIFAR-100, we train ResNet-18 for 200 epochs with a batch size of 128. SGD optimizer with momentum of 0.9 and weight decay of 0.0005 is used. The learning rate is initialized as 0.1 and decays with the cosine annealing scheduler. As Zhang et al. (2024) show that smaller batch size boosts performance at high pruning rates, we also halved the batch size for 80% pruning, and for 90% we reduced it to one-fourth. For ImageNet-1k, ResNet-34 is trained for 90 epochs with a batch size of 256 across all pruning ratios. An SGD optimizer with a momentum of 0.9, a weight decay of 0.0001, and an initial learning rate of 0.1 is used, combined with a cosine annealing scheduler.

**Baselines** The baselines considered in this study are listed as follows:[1] (1) Random; (2) Entropy (Coleman et al., 2020); (3) Forgetting (Toneva et al., 2018); (4) EL2N (Paul et al., 2021); (5) AUM (Pleiss et al., 2020); (6) Moderate (Xia et al., 2022); (7) Dyn-Unc (He et al., 2024); (8) TDDS (Zhang et al., 2024); (9) CCS (Zheng et al., 2022); and (10) $\mathbb{D}^2$ (Maharana et al., 2023). To ensure a fair comparison, all methods were trained using a common set of base hyperparameters (e.g., learning rate, batch size, number of epochs), while any method-specific hyperparameters were set to the optimal values reported for each score metric in their respective original works. Technical details are provided in the Appendix A.1.

## 4.2. Image Classification Benchmarks

Table 1 presents the test accuracy for image classification results on CIFAR-10 and CIFAR-100. Our pruning method consistently outperforms other baselines, particularly when combined with Beta sampling. While the DUAL score exhibits competitive performance in lower pruning ratios, its coreset accuracy degrades with more aggressive pruning. Our Beta sampling effectively mitigates this performance drop here.

Notably, the DUAL score requires training a single model for *only 30 epochs* for computation, significantly reducing the computational cost. In contrast, the second-best methods, Dyn-Unc and CCS, rely on scores computed over 200 epochs—a full training cycle on the original dataset—which makes them significantly less efficient. Even when accounting for subset selection, score computation, and subset training, the total time remains lower than training the full dataset once, as illustrated in Figure 4. Specifically, on CIFAR-10, our method achieves lossless pruning up to a 50% pruning ratio while saving 35.5% of total training time.

---

[1]Infomax (Tan et al., 2025) was excluded as it employs different base hyperparameters in the original paper compared to other baselines and does not provide publicly available code. See Appendix A.1 for more discussion.

*Table 2.* Comparison of test accuracy of DUAL score with existing coreset selection methods using ResNet34 for ImageNet-1K. The model trained with the full dataset achieves 73.1% test accuracy. The best result in each pruning raio is highlighted in bold.

| Pruning Rate ($\rightarrow$) | 30% | 50% | 70% | 80% | 90% |
|---|---|---|---|---|---|
| **Random** | 72.2 | 70.3 | 66.7 | 62.5 | 52.3 |
| **Entropy** | 72.3 | 70.8 | 64.0 | 55.8 | 39.0 |
| **Forgetting** | 72.6 | 70.9 | 66.5 | 62.9 | 52.3 |
| **EL2N** | 72.2 | 67.2 | 48.8 | 31.2 | 12.9 |
| **AUM** | 72.5 | 66.6 | 40.4 | 21.1 | 9.9 |
| **Moderate** | 72.0 | 70.3 | 65.9 | 61.3 | 52.1 |
| **Dyn-Unc** | 70.9 | 68.3 | 63.5 | 59.1 | 49.0 |
| **TDDS** | 70.5 | 66.8 | 59.4 | 54.4 | 46.0 |
| **CCS** | 72.3 | 70.5 | 67.8 | 64.5 | 57.3 |
| **D2** | 72.9 | 71.8 | 68.1 | 65.9 | 55.6 |
| **DUAL** | 72.8 | 71.5 | 68.6 | 64.7 | 53.1 |
| **DUAL+$\beta$ sampling** | **73.3** | **72.3** | **69.4** | **66.5** | **60.0** |

We also evaluate our pruning method on the large-scale dataset, ImageNet-1k. The DUAL score is computed during training, specifically at epoch 60, which is 33% earlier than the original train epoch used to compute scores for other baseline methods. As shown in Table 2, Dyn-Unc performs worse than random pruning across all pruning ratios, and we attribute this undesirable performance to its limited total training epochs (only 90), which is insufficient for Dyn-Unc to fully capture the training dynamics of each sample. In contrast, our DUAL score, combined with Beta sampling, outperforms all competitors while requiring the least computational cost. By considering both training dynamics and the difficulty of examples, DUAL can effectively identify uncertain samples early in the training process, even with limited training dynamics than full training. Remarkably, for 90% pruning on Imagenet-1K, it maintains a test accuracy of 60.0%, surpassing the previous state-of-the-art (SOTA) by a large margin.

## 4.3. Experiments under More Realistic Scenarios

### 4.3.1. LABEL NOISE AND IMAGE CORRUPTION

Data affected by label noise or image corruption are difficult and unnecessary samples that hinder model learning and degrade generalization performance. Therefore, filtering out these samples through data pruning is crucial. Most data pruning methods, however, either focus solely on selecting difficult samples based on example difficulty (Paul et al., 2021; Pleiss et al., 2020; Coleman et al., 2020) or prioritize dataset diversity (Zheng et al., 2022; Xia et al., 2022), making them unsuitable for effectively pruning such noisy and corrupted samples.

In contrast, methods that select uncertain samples while

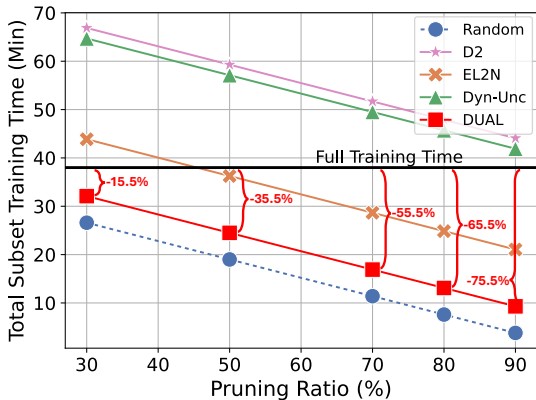

*Figure 4.* Comparison in total time spent (full dataset training, score estimation, and subset training) on CIFAR datasets. While other methods remain ineffective as they require more than full training, our method achieves a 15.5% time reduction with only 30% pruning, approaching the efficiency of random pruning.

considering training dynamics, such as Forgetting (Toneva et al., 2018) and Dyn-Unc (He et al., 2024), demonstrate robustness by pruning both the hardest and easiest samples, ultimately improving generalization performance, as illustrated in Figure 5a. However, since noisy samples tend to be memorized after useful samples are learned (Arpit et al., 2017; Jiang et al., 2020), there is a possibility that those noisy samples may still be treated as uncertain in the later stages of training and thus be included in the selected subset.

The DUAL score aims to identify high-uncertainty samples early in training by considering both training dynamics and example difficulty. Noisy data, typically under-learned compared to other challenging samples during this phase, exhibit lower uncertainty (Figure 13, Appendix B.1). Consequently, our method effectively prunes these noisy samples.

To verify this, we evaluate our method by introducing a specific proportion of symmetric label noise (Patrini et al., 2017; Xia et al., 2020; Li et al., 2022) and applying five different types of image corruptions (Wang et al., 2018; Hendrycks & Dietterich, 2019; Xia et al., 2021). We use CIFAR-100 with ResNet-18 and Tiny-ImageNet with ResNet-34 for these experiments. On CIFAR-100, we test label noise and image corruption ratios of 20%, 30%, and 40% using a model trained for 30 epochs. For Tiny-ImageNet, we use a 20% ratio of label noise and image corruption. We prune the label-noise-added dataset using a model trained for 50 epochs and the image-corrupted dataset with a model trained for 30 epochs using DUAL pruning—both significantly lower than the 200 epochs used by other methods. For detailed experimental settings, please refer to Appendix A.2.

As shown in Figure 5, the left plot demonstrates that DUAL pruning effectively removes mislabeled data at a ratio close

to the optimal. Notably, when the pruning ratio is 10%, nearly *all pruned samples are mislabeled data*. Consequently, as observed in Figure 5b, DUAL pruning leads to improved test accuracy compared to training on the full dataset, even up to a pruning ratio of 70%. At lower pruning ratios, performance improves as mislabeled data are effectively removed, highlighting the advantage of our approach in handling label noise. Similarly, for image corruption, our method prunes more corrupted data across all corruption rates compared to other methods, as shown in Figure 15, 16 in Appendix B.2. As a result, this leads to higher test accuracy, as demonstrated in Figure 5c.

Detailed results, including exact numerical values for different corruption rates and Tiny-ImageNet experiments, can be found in Appendix B.1 and B.2. Across all experiments, DUAL pruning consistently shows *strong noise robustness* and outperforms other methods by a substantial margin.

### 4.3.2. CROSS-ARCHITECTURE GENERALIZATION

Next, we evaluate the ability to transfer scores across various model architectures. Especially, if we can get high quality example scores for pruning by using a simpler architecture than one for the training, our DUAL pruning would become even more efficient in time and computational cost. Therefore, we focus on the cross-architecture generalization from relatively small networks to larger ones with three-layer CNN, VGG-16, ResNet-18, and ResNet-50. Results are summarized in Table 3.

Competitors are selected from each categorized group of the pruning approach: EL2N from difficulty-based, Dyn-Unc from uncertainty-based, and CCS from the geometry-based group. Standard deviations are omitted here due to space limit; please refer to Appendix B.3 for details.

*Table 3.* Cross-architecture generalization performance on CIFAR-100 from ResNet-18 to ResNet-50. '(R50)' marker denotes the score is computed on ResNet-50, serving as a baseline. DUAL + $\beta$ means our Beta sampling with DUAL scores. We report an average of five runs.

| Pruning Rate ($\rightarrow$) | ResNet-18 $\rightarrow$ ResNet-50 | | | |
|---|---|---|---|---|
| | 30% | 50% | 70% | 90% |
| Random | 77.17 | 73.74 | 66.66 | 40.48 |
| EL2N | 79.46 | 74.85 | 58.75 | 16.19 |
| Dyn-Unc | **79.90** | 75.78 | 61.75 | 25.08 |
| CCS | 77.24 | 73.81 | 66.66 | 40.31 |
| DUAL | 79.48 | **76.47** | **68.56** | 29.82 |
| DUAL + $\beta$ | 79.53 | 75.08 | 67.54 | **50.34** |
| DUAL (R50) | 79.60 | 76.64 | 68.60 | 29.84 |
| DUAL (R50) + $\beta$ | 79.63 | 76.49 | 70.37 | 50.27 |

Specifically, when pruning 90% of the original dataset, we find that other methods all fail, showing worse test accura-

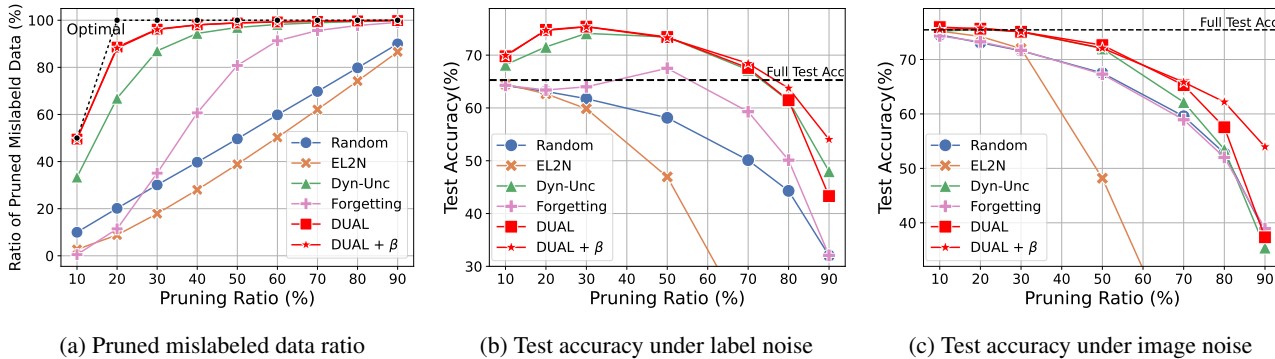

(a) Pruned mislabeled data ratio      (b) Test accuracy under label noise      (c) Test accuracy under image noise

*Figure 5.* The left figure shows the ratio of pruned mislabeled data under 20% label noise on CIFAR-100 trained with ResNet-18. When label noise is 20%, the optimal value (black dashed line) corresponds to pruning 100% of mislabeled data at a 20% pruning ratio. The middle and right figures depict test accuracy under 20% label noise and 20% image corruption, respectively. Our method effectively prunes mislabeled data near the optimal value while maintaining strong generalization performance. Results are averaged over five random seeds.

cies than random pruning. However, our proposed pruning method consistently shows the powerful ability to generalize across the various network architectures, outperforming or being on par with other computationally expensive baselines. More results are provided in Appendix B.3.

### 4.4. Ablation Studies

**Hyperparameter Analysis** In this section, we investigate the robustness of our hyperparameters, $T$, $J$, and $c_{\mathcal{D}}$. We fix $J$ across all experiments, as it has minimal impact on selection, indicating its robustness (Figure 9, Appendix B). In Figure 6, we assess the robustness of $T$ by varying it from 20 to 200 on CIFAR-100. We find that while $T$ is highly robust in early epochs, increasing it eventually degrades generalization. This is expected, as larger $T$ overemphasizes difficult samples due to our difficulty-aware selection. Thus, pruning in earlier epochs (30 to 50) proves more effective and robust. For $c_{\mathcal{D}}$, we vary it from 3 to 6 and observe consistent performance, indicating robustness to its choice as well.

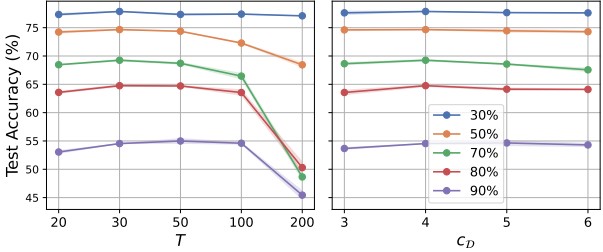

*Figure 6.* **Left**: T varying while $J = 10$ and $c_{\mathcal{D}} = 4$. **Right**: $c_{\mathcal{D}}$ varying while $T = 30$ and $J = 10$. Three runs are averaged.

**Beta sampling with existing scores** Next, we study the impact of our proposed pruning-ratio-adaptive Beta sam-

pling on existing score metrics. We apply our Beta sampling strategy to other score-based methods, including Forgetting, EL2N, and Dyn-Unc, on the CIFAR10 and CIFAR100 datasets. Compared to vanilla threshold pruning, which selects only the highest-scoring samples, we observe that previous methods become competitive when Beta sampling is adjusted (see Table 4). For the case of random pruning combined with Beta sampling, we do not use any score but select samples only with Beta sampling.

*Table 4.* Comparison on CIFAR-10 and CIFAR-100 for 90% pruning rate. We report average accuracy with five runs. The best performance is in bold in each column.

| | CIFAR-10 | |
|---|---|---|
| Method | Thresholding | $\beta$-Sampling |
| Random | **83.74** $\pm$ 0.21 | 83.31 (-0.43) $\pm$ 0.14 |
| EL2N | 38.74 $\pm$ 0.75 | 87.00 (+48.26) $\pm$ 0.45 |
| Forgetting | 46.64 $\pm$ 1.90 | 85.67 (+39.03) $\pm$ 0.13 |
| Dyn-Unc | 59.67 $\pm$ 1.79 | 85.33 (+25.66) $\pm$ 0.20 |
| Ours | 54.95 $\pm$ 0.42 | **87.09** (+32.14) $\pm$ 0.36 |

| | CIFAR-100 | |
|---|---|---|
| Method | Thresholding | $\beta$-Sampling |
| Random | **45.09** $\pm$ 1.26 | 51.76 (+6.67) $\pm$ 0.25 |
| EL2N | 8.89 $\pm$ 0.28 | 53.97 (+45.08) $\pm$ 0.63 |
| Forgetting | 26.87 $\pm$ 0.73 | 52.40 (+25.53) $\pm$ 0.43 |
| Dyn-Unc | 34.57 $\pm$ 0.69 | 51.85 (+17.28) $\pm$ 0.35 |
| Ours | 34.28 $\pm$ 1.39 | **54.54** (+20.26) $\pm$ 0.09 |

Even with random pruning, our Beta sampling continues to perform well. Notably, EL2N, which performs poorly on its own, becomes significantly more effective when combined with our sampling method. Similar improvements are also seen with Forgetting and Dyn-Unc scores. This is be-

cause our proposed Beta sampling enhances the diversity of selected samples, especially when used with example difficulty-based methods. More results conducted at 80% are included in the Appendix B.6.

**Additional Analysis** In addition to the main results presented in this paper, we conducted various experiments to further explore the effectiveness of our method, which is located in Appendix B. For instance, to list a few, these investigations include: (i) calculating the Spearman rank correlation between individual DUAL scores and their average DUAL score across five runs to assess score consistency; and (ii) analyzing coreset performance under a time budget. The findings from these analyses are presented in Figure 7 and Figure 8, respectively, within Appendix B.

Furthermore, detailed results on extreme cases of label noise (ranging from 20% to 40% for CIFAR-100 and 20% for Tiny-ImageNet) are presented in Appendix B.1. Similar comprehensive results for various image corruptions can be found in Appendix B.2. The generalization performance of our method across other network architectures is further detailed in Appendix B.3. Additionally, results for long-tailed data classification using the CIFAR-10-LT and CIFAR-100-LT datasets are provided in Appendix B.4. Lastly, a comparison with dynamic pruning methods such as Qin et al. (2024) and Yuan et al. (2025) is provided in Appendix B.5.

## 5. Conclusion

This paper introduces Difficulty and Uncertainty-Aware Lightweight (DUAL), a novel scoring metric for cost-effective pruning. DUAL is the first metric to combine difficulty and uncertainty into a single measure, and its effectiveness in identifying the most informative samples early in training is further supported by theoretical analysis. Further, we propose pruning-ratio-adaptive sampling to consider the sample diversity when the pruning ratio is extremely high. Our proposed DUAL score, combined with Beta sampling, shows remarkable performance, particularly under label noise and image corruption by effectively distinguishing noisy samples. Future work could explore extending this approach to unsupervised settings.

## Acknowledgement

This work was supported by two Institute of Information & communications Technology Planning & Evaluation (IITP) grants (No. RS-2019-II190075, Artificial Intelligence Graduate School Program (KAIST); No. RS-2024-00457882, National AI Research Lab Project) funded by the Korean government (MSIT), and a National Research Foundation of Korea (NRF) grant (No. NRF-2019R1A5A1028324) funded by the Korean government (MSIT).

## Impact Statement

This paper presents work on data pruning to advance machine learning, with the potential for positive societal impact through improved efficiency.

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

# A. Technical Details

## A.1. Details on Baseline Implementation

**EL2N** (Paul et al., 2021) is defined the error L2 norm between the true labels and predictions of model. Then examples with low scores are pruned out. We calculate error norm at epoch 20 from five independent runs, then the average was used for EL2N score.

**Forgetting** (Toneva et al., 2018) is defined as the number of forgetting events, where the model prediction goes wrong after the correct prediction, up until the end of training. Rarely are unforgotten samples pruned out.

**AUM** (Pleiss et al., 2020) accumulates the margin, which means the gap between the target probabilities and the second largest prediction of model. They calculate the margin at every epoch and then transform it into an AUM score at the end of the training. Here samples with small margin are considered as mislabeled samples, thus data points with small AUM scores are eliminated.

**Entropy** (Coleman et al., 2020) is calculated as the entropy of prediction probabilities at the end of training, then the samples which have high entropy are selected into coreset.

**Dyn-Unc** (He et al., 2024) is also calculated at the end of training, with the window length J set as 10. Samples with high uncertainties are selected into the subset after pruning.

**TDDS** (Zhang et al., 2024) adapts different hyperparameter for each pruning ratio. As they do not provide full information for implementation, we have no choice but set parameters for the rest case arbitrarily. The provided setting for (pruning ratio, computation epoch $T$, the length of sliding window $K$) is $(0.3, 70, 10)$, $(0.5, 90, 10)$, $(0.7, 80, 10)$, $(0.8, 30, 10)$, and $(0.9, 10, 5)$ for CIFAR-100, and for ImageNet-$(0.3, 20, 10)$, $(0.5, 20, 10)$, $(0.7, 30, 20)$. Therefore, we set the parameter for CIFAR-10 as the same with CIFAR-100, and 80%, 90% pruning on ImageNet-1K, we set them as $(30, 20)$, following the choice for 70% pruning.

**CCS** (Zheng et al., 2022) for stratified sampling method, we adapt AUM score as the original CCS paper does. They assign different hard cutoff rate for each pruning ratio, For CIFAR10, the cutoff rate is $(30\%, 0)$, $(50\%, 0)$, $(70\%, 10\%)$, $(80\%, 10\%)$, $(90\%, 30\%)$. For CIFAR100 and ImageNet-1K, we set them as the same with the original paper. As explicitly mentioned in Appendix B of Zheng et al. (2022), we use the AUM score calculated at the end of training. This means scores are computed at epoch 200 for the CIFAR-10/100 datasets and at epoch 90 for the ImageNet-1K dataset.

**D2** (Maharana et al., 2023) for $\mathbb{D}^2$ pruning, we set the initial node using forgetting scores for CIFAR-10 and CIFAR-100, we set the number of neighbors $k$, and message passing weight $\gamma$ as the same with the original paper.

*Remark*. As detailed by Zhang et al. (2024) (Section 5.2), TDDS employs 90 epochs for initial full-dataset training. Subsequently, an exhaustive search is conducted to determine an optimal epoch for score computation (e.g., 30 epoch for pruning ImageNet by 70-90%). In our evaluations, we utilized these reported optimal epochs for TDDS across all pruning ratios. While this approach leads to a shorter score computation period for TDDS (excluding the significant overhead of the exhaustive search itself), it is crucial to note that our proposed method consistently achieves significantly higher test accuracy. This advantage is demonstrated by both their reported results and our reproduced experiments.

Note that, Infomax (Tan et al., 2025) was excluded as it employs different base hyperparameters in the original paper compared to other baselines and does not provide publicly available code. Additionally, implementation details, such as the base score metric used to implement Infomax, are not provided. As we intend to compare other baseline methods with the same training hyperparameters, we do not include the accuracies of Infomax in our tables. To see if we can match the performance of Infomax, we tested our method with different training details. For example, if we train the subset using the same number of iterations (not epoch) as the full dataset and use a different learning rate tuned for our method, then an improved accuracy of 59% is achievable for 90% pruning on CIFAR-100, which surpasses the reported performance of Infomax. For the ImageNet-1K dataset, our method outperforms Infomax without any base hyperparameter tuning, while also being cost-effective.

## A.2. Detailed Experimental Settings

Here we clarify the technical details in our works. For training the model on full-dataset and the selected subset, all parameters are used identically only except for batch sizes. For CIFAR-10/100, we train ResNet-18 for 200 epochs with batch size of 128, for each pruning ratio $\{30\%, 50\%, 70\%, 80\%, 90\%\}$ we use different batch sizes with $\{128, 128, 128,$

64, 32}. We set the initial learning rate as 0.1, optimizer as SGD with momentum 0.9, and scheduler as cosine annealing scheduler with weight decay 0.0005. For training ImageNet, we use ResNet-34 as the network architecture. For all coresets with different pruning rates, we train models for 300,000 iterations with a 256 batch size. We use the SGD optimizer with 0.9 momentum and 0.0001 weight decay, using a 0.1 initial learning rate. The cosine annealing learning rate scheduler was used for training. For fair comparison, we use the same parameters across all pruning methods, including ours. All experiments were conducted using an NVIDIA A6000 GPU. We also attach the implementation in the supplementary material.

For calculating DUAL score, we need three parameters $T$, $J$, and $c_{\mathcal{D}}$, each means score computation epoch, the length of sliding window, and hyperparameter regarding the train dataset. We fix $J$ as 10 for all experiments. We use ($T$, $J$, $c_{\mathcal{D}}$) for each dataset as followings. For CIFAR-10, we use (30, 10, 5.5), for CIFAR-100, (30, 10, 4), and for ImageNet-1K, (60, 10, 11). We first roughly assign the term $c_{\mathcal{D}}$ based on the size of initial dataset and by considering the relative difficulty of each, we set $c_{\mathcal{D}}$ for CIFAR-100 smaller than that of CIFAR-10. For the ImageNet-1K dataset, which contains 1,281,167 images, the size of the initial dataset is large enough that we do not need to set $c_{\mathcal{D}}$ to a small value to in order to intentionally sample easier samples. Also, note that we fix the value of $C$ of Beta distribution at 15 across all experiments. A more detailed distribution, along with a visualization, can be found in Appendix C.

Experiments with label noise and image corruption on CIFAR-100 are conducted under the same settings as described above, except for the hyperparameters for DUAL pruning. For label noise experiments, we set $T$ to 50 and $J$ to 10 across all label noise ratio. For $c_{\mathcal{D}}$, we set it to 6 for 20% and 30% noise, 8 for 40% noise. For image corruption experiments, we set $T$ to 30, $J$ to 10, and $c_{\mathcal{D}}$ to 6 across all image corruption ratio.

For the Tiny-ImageNet case, we train ResNet-34 for 90 epochs with a batch size of 256 across all pruning ratios, using a weight decay of 0.0001. The initial learning rate is set to 0.1 with the SGD optimizer, where the momentum is set to 0.9, combined with a cosine annealing learning rate scheduler. For the hyperparameters used in DUAL pruning, we set $T$ to 60, $J$ to 10, and $c_{\mathcal{D}}$ to 6 for the label noise experiments. For the image corruption experiments, we set $T$ to 60, $J$ to 10, and $c_{\mathcal{D}}$ to 2. We follow the ImageNet-1K hyperparameters to implement the baselines.

# B. More Results on Experiments

We evaluate our proposed DUAL score through a wide range of analyses in this section. In Appendix B.1 and B.2, we demonstrate the robustness of the DUAL score through intensive experiments. In Appendix B.3, we investigate the cross-architecture performance of our method. In Appendix B.6, we show the effectiveness of our Beta sampling when combined with other existing scores, compared to previous sampling strategies.

We first investigate the stability of our DUAL score. We calculate the Spearman rank correlation of individual scores and the average across five runs, following Paul et al. (2021). As shown in Figure 7, snapshot-based methods such as EL2N and Entropy exhibit relatively low correlation compared to methods using training dynamics. In particular, the DUAL score shows minimal variation across runs with a high Spearman rank correlation. This shows strong stability across random seeds.

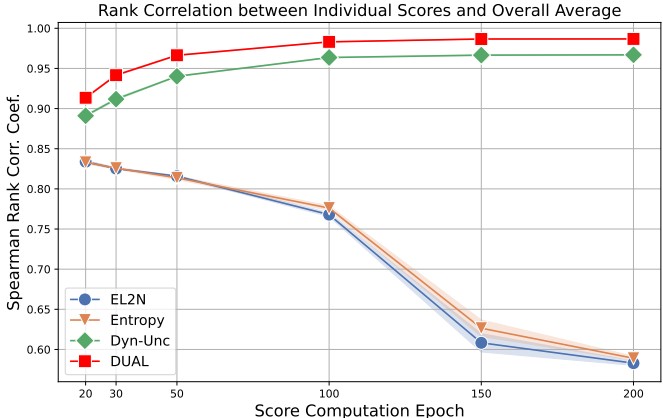

*Figure 7.* Average of Spearman rank correlation among independent runs and overall average of five runs. DUAL score is calculated at 30th epoch.

Next, we compute the Dyn-Unc, TDDS, and AUM scores at the 30th epoch, as we do for our method, and then compare the test accuracy on the coreset. Our pruning method, using the DUAL score and ratio-adaptive Beta sampling, outperforms the others by a significant margin, as illustrated in Figure 8. We see that using epoch of 30 results in insufficient training dynamics for the others, thus it negatively impacts their performance.

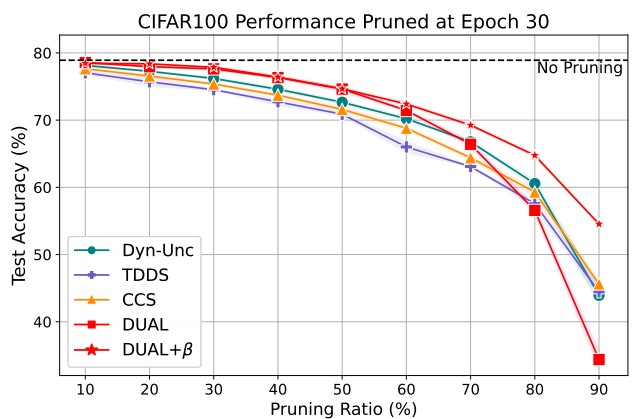

*Figure 8.* Test accuracy comparison under limited computation budget (epoch 30)

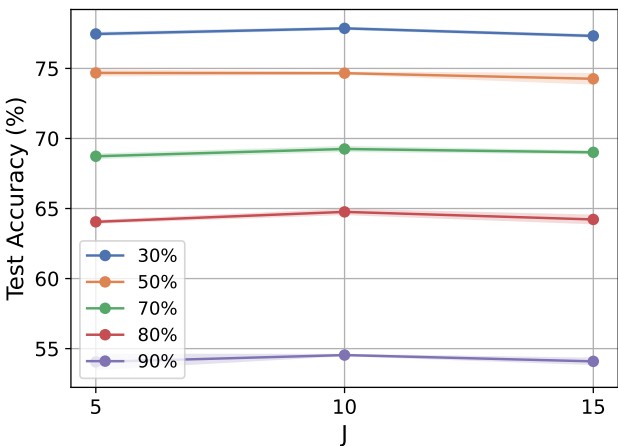

*Figure 9.* J varies from 5 to 15, showing minimal differences, which demonstrates its robustness. We fix $T = 30$, $C_{\mathcal{D}} = 4$. Runs are averaged over three runs.

Figure 10 is a visualization of samples kept and pruned by our method. Samples kept by our method are more recognizable. The black swan on the grass and the sun-shaped balloon are rare cases, while the others are more easily recognizable. Samples pruned by our method are either typical, confusing, or mislabeled. The first and fifth examples are mislabeled, the second and third are typical, and the fourth is confusing.

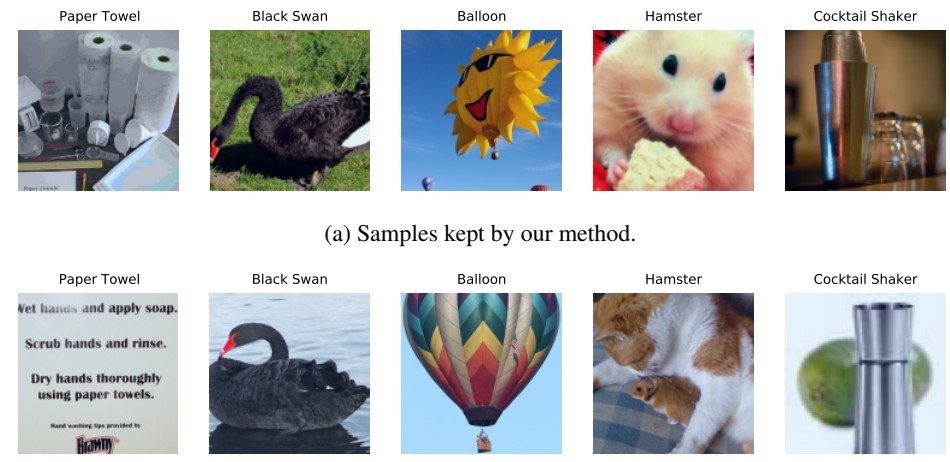

(a) Samples kept by our method.

(b) Samples pruned by our method.

*Figure 10.* Illustrations of samples kept and pruned by our method at the pruning ratio of 30%. The pruned samples are likely either typical, confusing, or mislabeled, while the kept ones are certainly recognizable.

We compare the subset selected at the high pruning ratios by previous SOTA methods, namely CCS and D2. First, we examine the total amount of overlap by counting the number of samples in the intersection of each method in Table 5. Furthermore, for intuitive understanding, we visualize the selected subset by each method for the 70%, 80% pruning cases on CIFAR-100 in Figure 11. We can see that DUAL+$\beta$ seems to include more difficult examples than others.

*Table 5.* Comparison of subset overlap ratio over different pruning methods at high pruning rates. Let $S$ and $T$ denote the subsets selected by the two methods. The subset overlap ratio is computed as $\frac{|S \cap T|}{|S|}$ (which is equal to $\frac{|S \cap T|}{|T|}$).

| Pruning Rate | CCS & DUAL+$\beta$ | D2 & DUAL+$\beta$ | CCS & D2 |
|:---:|:---:|:---:|:---:|
| 70% | 0.41 | 0.37 | 0.42 |
| 80% | 0.31 | 0.27 | 0.34 |
| 90% | 0.19 | 0.17 | 0.21 |

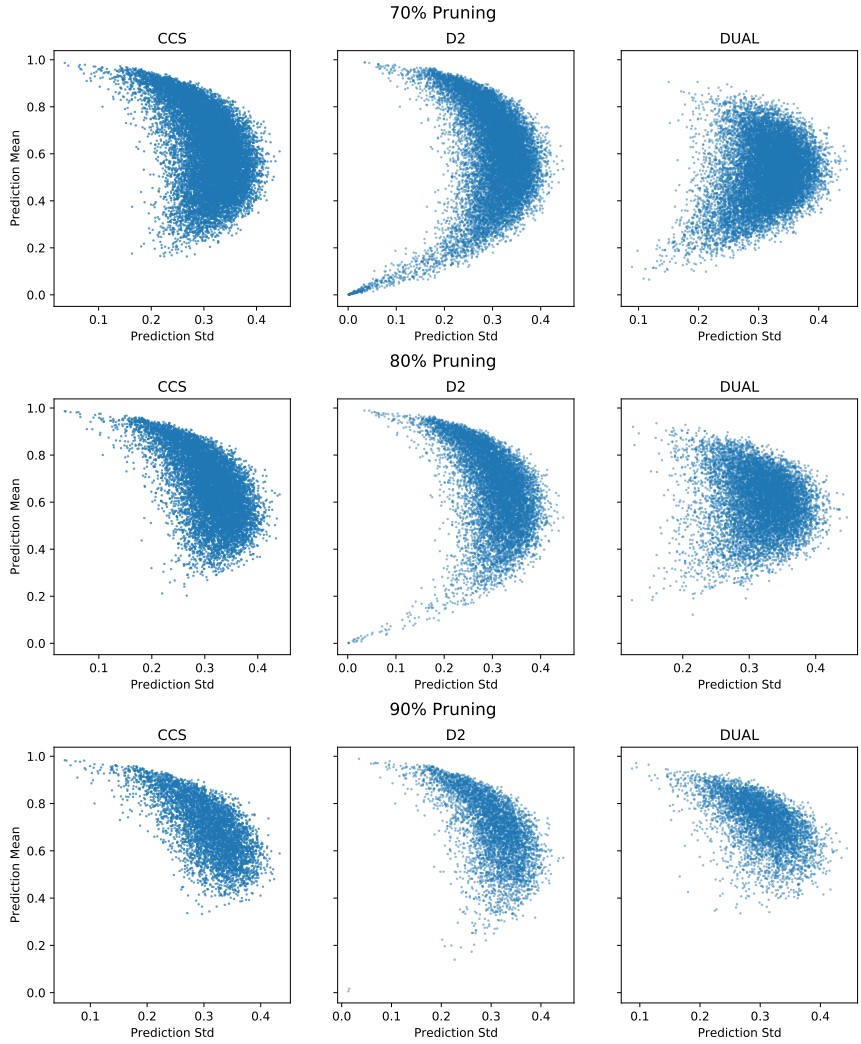

*Figure 11.* Comparison of selected subset over different methods at high pruning rates.

## B.1. Image Classification Under Label Noise

We evaluated the robustness of our DUAL pruning method against label noise. We introduced symmetric label noise by replacing the original labels with labels from other classes randomly. For example, if we apply 20% label noise to a dataset with 100 classes, 20% of the data points are randomly selected, and each label is randomly reassigned to another label with a probability of $1/99$ for the selected data points.

Even under 30% and 40% random label noise, our method achieves the best performance and accurately identifies the noisy labels, as can be seen in Figure 12. By examining the proportion of noise removed, we can see that our method operates close to optimal.

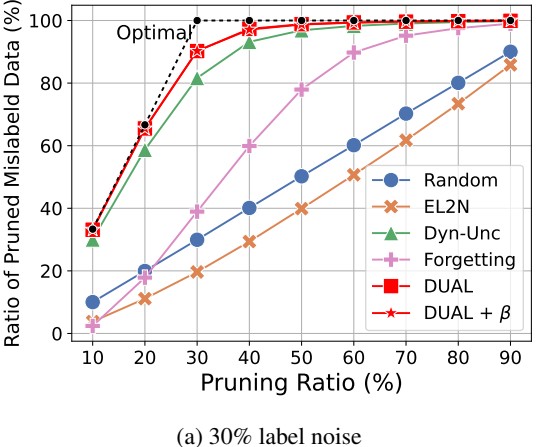 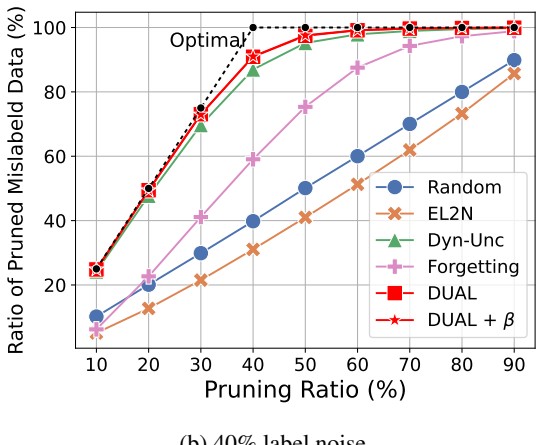

(a) 30% label noise

(b) 40% label noise

*Figure 12.* Ratio of pruned mislabeled data under 30% and 40% label noise on CIFAR-100

Figure 13 shows a scatter plot of the CIFAR-100 dataset under 20% label noise. The model is trained for 30 epochs, and we compute the prediction mean (y-axis) and standard deviation (x-axis) for each data point. Red dots represent the 20% mislabeled data. These points remain close to the origin (0,0) during the early training phase. Therefore, pruning at this stage allows us to remove mislabeled samples nearly optimally while selecting the most uncertain ones.

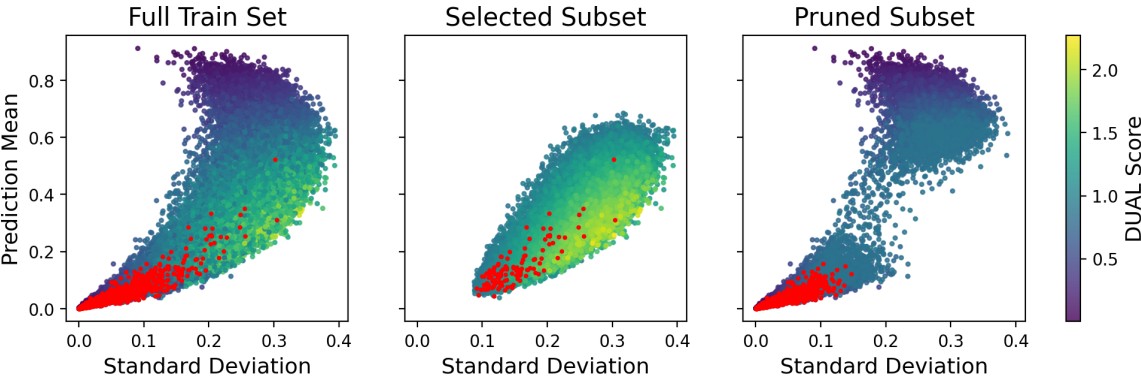

*Figure 13.* Pruning ratio is set to 50%. Only 116 data points over 10,000 mislabeled data are selected as a subset where red dots indicate mislabeled data.

We evaluated the performance of our proposed method across a wide range of pruning levels, from 10% to 90%, and compared the final accuracy with that of baseline methods. As shown in the Table 6-9, our method consistently outperforms the competition with a substantial margin in most cases. For a comprehensive analysis of performance under noisy conditions, please refer to Tables 6 to 8 for CIFAR-100, which show results for 20%, 30%, and 40% noise, respectively. Additionally, the results for 20% label noise in Tiny-ImageNet are shown in Table 9.

*Table 6.* Comparison of test accuracy of DUAL pruning with existing coreset selection methods under 20% label noise using ResNet-18 for CIFAR-100. The model trained with the full dataset achieves **65.28%** test accuracy on average. Results are averaged over five runs.

| Pruning Rate ($\rightarrow$) | 10% | 20% | 30% | 50% | 70% | 80% | 90% |
|---|---|---|---|---|---|---|---|
| **Random** | 64.22 ±0.37 | 63.12 ±0.26 | 61.75 ±0.24 | 58.13 ±0.22 | 50.11 ±0.75 | 44.29 ±1.2 | 32.04 ±0.93 |
| **Entropy** | 63.51 ±0.25 | 60.59 ±0.23 | 56.75 ±0.37 | 44.90 ±0.74 | 24.43 ±0.12 | 16.60 ±0.29 | 10.35 ±0.49 |
| **Forgetting** | 64.29 ±0.26 | 63.40 ±0.14 | 64.00 ±0.27 | 67.51 ±0.52 | 59.29 ±0.66 | 50.11 ±0.91 | 32.08 ±1.15 |
| **EL2N** | 64.51 ±0.35 | 62.67 ±0.28 | 59.85 ±0.31 | 46.94 ±0.75 | 19.32 ±0.87 | 11.02 ±0.45 | 6.83 ±0.21 |
| **AUM** | 64.54 ±0.23 | 60.72 ±0.22 | 50.38 ±0.66 | 22.03 ±0.92 | 5.55 ±0.26 | 3.00 ±0.18 | 1.68 ±0.10 |
| **Moderate** | 64.45 ±0.29 | 62.90 ±0.33 | 61.46 ±0.50 | 57.53 ±0.61 | 49.50 ±1.06 | 43.81 ±0.80 | 29.15 ±0.79 |
| **Dyn-Unc** | 68.17 ±0.26 | 71.56 ±0.27 | 74.12 ±0.15 | **73.43** ±0.12 | 67.21 ±0.27 | 61.38 ±0.27 | 48.00 ±0.79 |
| **TDDS** | 62.86 ±0.36 | 61.96 ±1.03 | 61.38 ±0.53 | 59.16 ±0.94 | 48.93 ±1.68 | 43.83 ±1.13 | 34.05 ±0.49 |
| **CCS** | 64.30 ±0.21 | 63.24 ±0.24 | 61.91 ±0.45 | 58.24 ±0.29 | 50.24 ±0.39 | 43.76 ±1.07 | 30.67 ±0.96 |
| **DUAL** | 69.78 ±0.28 | **74.79** ±0.07 | **75.40** ±0.11 | **73.43** ±0.16 | 67.57 ±0.18 | 61.46 ±0.45 | 43.30 ±1.59 |
| **DUAL+$\beta$ sampling** | **69.95** ±0.60 | 74.68 ±1.22 | 75.37 ±1.33 | 73.29 ±0.84 | **68.43** ±0.77 | **63.74** ±0.35 | **54.04** ±0.92 |

*Table 7.* Comparison of test accuracy of DUAL pruning with existing coreset selection methods under 30% label noise using ResNet-18 for CIFAR-100. The model trained with the full dataset achieves **58.25%** test accuracy on average. Results are averaged over five runs.

| Pruning Rate ($\rightarrow$) | 10% | 20% | 30% | 50% | 70% | 80% | 90% |
|---|---|---|---|---|---|---|---|
| **Random** | 57.67 ±0.52 | 56.29 ±0.55 | 54.70 ±0.60 | 51.41 ±0.38 | 42.67 ±0.80 | 36.86 ±1.01 | 25.64 ±0.82 |
| **Entropy** | 55.51 ±0.42 | 51.87 ±0.36 | 47.16 ±0.58 | 35.35 ±0.49 | 18.69 ±0.76 | 13.61 ±0.42 | 8.58 ±0.49 |
| **Forgetting** | 56.76 ±0.62 | 56.43 ±0.28 | 58.84 ±0.26 | 64.51 ±0.37 | 61.26 ±0.69 | 52.94 ±0.68 | 34.99 ±1.16 |
| **EL2N** | 56.39 ±0.53 | 54.41 ±0.68 | 50.29 ±0.40 | 35.65 ±0.79 | 13.05 ±0.51 | 8.52 ±0.40 | 6.16 ±0.40 |
| **AUM** | 56.51 ±0.56 | 49.10 ±0.72 | 37.57 ±0.66 | 11.56 ±0.46 | 2.79 ±0.23 | 1.87 ±0.24 | 1.43 ±0.12 |
| **Moderate** | 57.31 ±0.75 | 56.11 ±0.45 | 54.52 ±0.48 | 50.71 ±0.42 | 42.47 ±0.29 | 36.21 ±1.09 | 24.85 ±1.72 |
| **Dyn-Unc** | 62.20 ±0.44 | 66.48 ±0.40 | 70.45 ±0.50 | **71.91** ±0.34 | 66.53 ±0.19 | 61.95 ±0.46 | 49.51 ±0.52 |
| **TDDS** | 57.24 ±0.44 | 55.64 ±0.46 | 53.97 ±0.46 | 49.04 ±1.05 | 39.90 ±1.21 | 35.02 ±1.34 | 26.99 ±1.03 |
| **CCS** | 57.26 ±0.48 | 56.52 ±0.23 | 54.76 ±0.52 | 51.29 ±0.32 | 42.33 ±0.78 | 36.61 ±1.31 | 25.64 ±1.65 |
| **DUAL** | 62.42 ±0.48 | **67.52** ±0.40 | **72.65** ±0.17 | 71.55 ±0.23 | 66.35 ±0.14 | 61.57 ±0.44 | 48.70 ±0.19 |
| **DUAL+$\beta$ sampling** | **63.02** ±0.41 | **67.52** ±0.24 | 72.57 ±0.15 | 71.68 ±0.27 | **66.75** ±0.45 | **62.28** ±0.43 | **52.60** ±0.87 |

*Table 8.* Comparison of test accuracy of DUAL pruning with existing coreset selection methods under 40% label noise using ResNet-18 for CIFAR-100. The model trained with the full dataset achieves **52.74%** test accuracy on average. Results are averaged over five runs.

| Pruning Rate ($\rightarrow$) | 10% | 20% | 30% | 50% | 70% | 80% | 90% |
|---|---|---|---|---|---|---|---|
| **Random** | 51.13 ±0.71 | 48.42 ±0.46 | 46.99 ±0.29 | 43.24 ±0.46 | 33.60 ±0.50 | 28.28 ±0.81 | 19.52 ±0.79 |
| **Entropy** | 49.14 ±0.32 | 46.06 ±0.58 | 41.83 ±0.73 | 28.26 ±0.37 | 15.64 ±0.19 | 12.21 ±0.68 | 8.23 ±0.40 |
| **Forgetting** | 50.98 ±0.72 | 50.36 ±0.48 | 52.86 ±0.47 | 60.48 ±0.68 | 61.55 ±0.58 | 54.57 ±0.86 | 37.68 ±1.63 |
| **EL2N** | 50.09 ±0.86 | 46.35 ±0.48 | 41.57 ±0.26 | 23.42 ±0.80 | 9.00 ±0.25 | 6.80 ±0.44 | 5.58 ±0.40 |
| **AUM** | 50.60 ±0.54 | 41.84 ±0.76 | 26.29 ±0.72 | 5.49 ±0.19 | 1.95 ±0.21 | 1.44 ±0.14 | 1.43 ±0.24 |
| **Moderate** | 50.62 ±0.27 | 48.70 ±0.79 | 47.01 ±0.21 | 42.73 ±0.39 | 32.35 ±1.29 | 27.72 ±1.69 | 19.85 ±1.11 |
| **Dyn-Unc** | 54.46 ±0.27 | 59.02 ±0.23 | 63.86 ±0.47 | 69.76 ±0.16 | **65.36** ±0.14 | 61.37 ±0.32 | 50.49 ±0.71 |
| **TDDS** | 50.65 ±0.23 | 48.83 ±0.38 | 46.93 ±0.66 | 41.85 ±0.37 | 33.31 ±0.79 | 29.39 ±0.35 | 21.09 ±0.89 |
| **CCS** | 64.30 ±0.29 | 48.54 ±0.35 | 46.81 ±0.45 | 42.57 ±0.32 | 33.19 ±0.88 | 28.32 ±0.59 | 19.61 ±0.75 |
| **DUAL** | 54.46 ±0.33 | 58.99 ±0.34 | **64.71** ±0.44 | 69.87 ±0.28 | 64.21 ±0.21 | 59.90 ±0.44 | 49.61 ±0.27 |
| **DUAL+$\beta$ sampling** | **54.53** ±0.06 | **59.65** ±0.41 | 64.67 ±0.34 | **70.09** ±0.33 | 65.12 ±0.46 | 60.62 ±0.30 | **51.51** ±0.41 |

*Table 9.* Comparison of test accuracy of DUAL pruning with existing coreset selection methods under 20% label noise using ResNet-34 for Tiny-ImageNet. The model trained with the full dataset achieves **42.24%** test accuracy on average. Results are averaged over three runs.

| Pruning Rate ($\rightarrow$) | 10% | 20% | 30% | 50% | 70% | 80% | 90% |
|---|---|---|---|---|---|---|---|
| **Random** | 41.09 ±0.29 | 39.24 ±0.39 | 37.17 ±0.23 | 32.93 ±0.45 | 26.12 ±0.63 | 22.11 ±0.42 | 13.88 ±0.60 |
| **Entropy** | 40.69 ±0.06 | 38.14 ±0.92 | 35.93 ±1.56 | 31.24 ±1.76 | 23.65 ±2.05 | 18.53 ±2.10 | 10.52 ±1.64 |
| **Forgetting** | 43.60 ±0.65 | 44.82 ±0.20 | 45.65 ±0.48 | 46.05 ±0.07 | 41.08 ±0.53 | 34.89 ±0.12 | 24.58 ±0.06 |
| **EL2N** | 41.05 ±0.35 | 38.88 ±0.63 | 32.91 ±0.39 | 20.89 ±0.80 | 8.08 ±0.24 | 4.92 ±0.32 | 3.12 ±0.07 |
| **AUM** | 40.20 ±0.27 | 34.68 ±0.35 | 29.01 ±0.12 | 10.45 ±0.85 | 2.52 ±0.75 | 1.30 ±0.23 | 0.79 ±0.40 |
| **Moderate** | 41.23 ±0.38 | 38.58 ±0.60 | 37.60 ±0.66 | 32.65 ±1.18 | 25.68 ±0.40 | 21.74 ±0.63 | 14.15 ±0.73 |
| **Dyn-Unc** | 45.67 ±0.78 | 47.49 ±0.46 | 49.38 ±0.17 | 47.47 ±0.32 | 42.49 ±0.39 | 37.44 ±0.73 | 28.48 ±0.73 |
| **TDDS** | 36.56 ±0.54 | 36.90 ±0.48 | 47.62 ±1.36 | 42.44 ±0.63 | 34.32 ±0.26 | 24.32 ±0.26 | 17.43 ±0.17 |
| **CCS** | 40.49 ±0.67 | 39.06 ±0.24 | 37.67 ±0.46 | 30.83 ±1.02 | 22.38 ±0.70 | 19.66 ±0.58 | 12.23 ±0.64 |
| **DUAL** | **45.76** ±0.67 | **48.20** ±0.20 | **49.94** ±0.17 | **48.19** ±0.27 | 42.80 ±0.74 | **37.90** ±0.59 | 27.80 ±0.49 |
| **DUAL+$\beta$ sampling** | 45.21 ±0.08 | 47.76 ±0.33 | 48.99 ±0.32 | 46.95 ±0.23 | **43.01** ±0.43 | **37.91** ±0.28 | **28.78** ±0.57 |

## B.2. Image Classification Under Image Corruption

We also evaluated the robustness of our proposed method against five different types of realistic image corruption: motion blur, fog, reduced resolution, rectangular occlusion, and Gaussian noise across the corruption rate from 20% to 40%. The ratio of each type of corruption is 4% for 20% corruption, 6% for 30% corruption, and 8% for 40% corruption. Example images for each type of corruption can be found in Figure 14. Motion blur, reduced resolution, and rectangular occlusion are somewhat distinguishable, whereas fog and Gaussian noise are difficult for the human eye to differentiate. Somewhat surprisingly, our DUAL pruning prioritize to remove the most challenging examples, such as fog and Gaussian corrupted images, as shown in Figure 16.

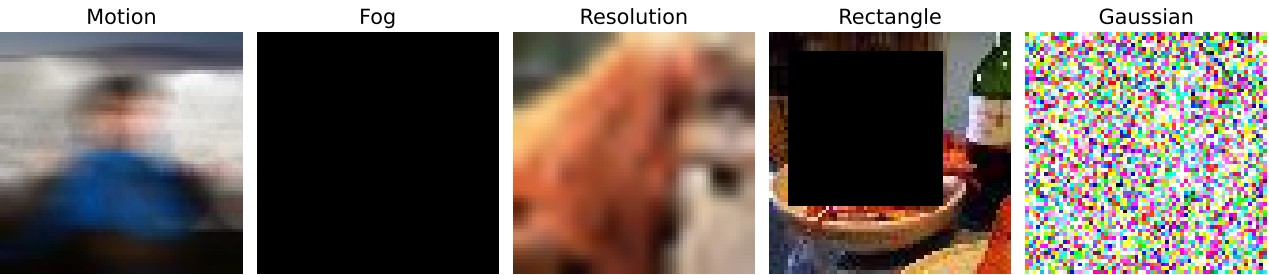

*Figure 14.* Examples of the different types of noise used for image corruption. Here we consider motion blur, fog, resolution, rectangle, and Gaussian noise.

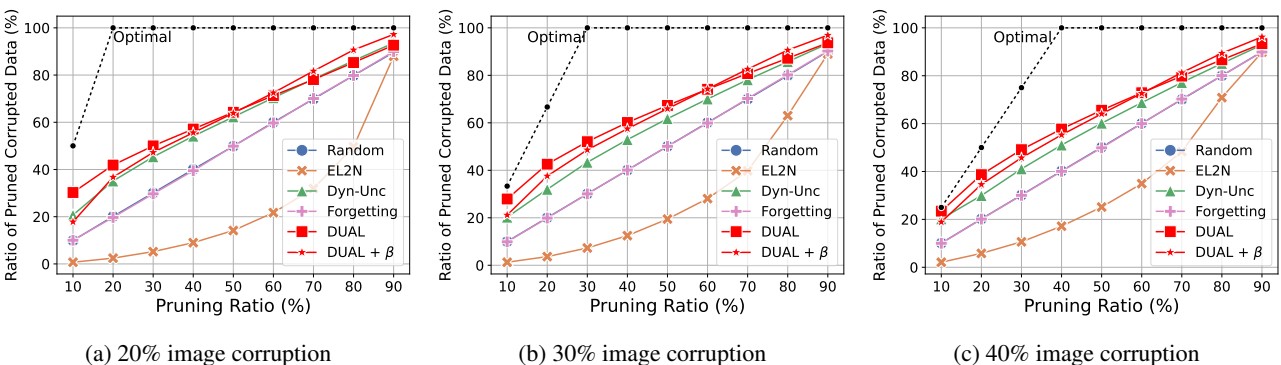

(a) 20% image corruption      (b) 30% image corruption      (c) 40% image corruption

*Figure 15.* Ratio of pruned corrupted samples with corruption rate of 20%, 30% and 40% on CIFAR-100.

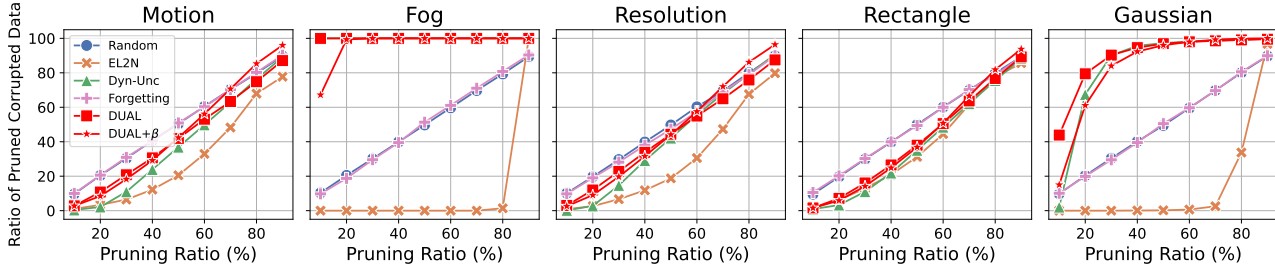

*Figure 16.* Illustration of the different types of noise used for image corruption. DUAL pruning prioritizes removing the most challenging corrupted images, such as fog and Gaussian noise.

We evaluated the performance of our proposed method across a wide range of pruning levels, from 10% to 90%, and compared the final accuracy with that of baseline methods. As shown in the table, our method consistently outperforms the competitors in most cases. For a comprehensive analysis of performance under noisy conditions, please refer to Tables 10 to 12 for CIFAR-100, which show results for 20%, 30%, and 40% corrupted images, respectively. Additionally, the results for 20% image corruption in Tiny-ImageNet are shown in Table 13.

*Table 10.* Comparison of test accuracy of DUAL pruning with existing coreset selection methods under 20% image corrupted data using ResNet-18 for CIFAR-100. The model trained with the full dataset achieves **75.45%** test accuracy on average. Results are averaged over five runs.

| Pruning Rate ($\rightarrow$) | 10% | 20% | 30% | 50% | 70% | 80% | 90% |
|---|---|---|---|---|---|---|---|
| **Random** | 74.54 $\pm$0.14 | 73.08 $\pm$0.27 | 71.61 $\pm$0.14 | 67.52 $\pm$0.32 | 59.57 $\pm$0.52 | 52.79 $\pm$0.68 | 38.26 $\pm$1.32 |
| **Entropy** | 74.74 $\pm$0.25 | 73.15 $\pm$0.26 | 71.15 $\pm$0.13 | 64.97 $\pm$0.52 | 49.49 $\pm$1.40 | 35.92 $\pm$0.64 | 17.91 $\pm$0.45 |
| **Forgetting** | 74.33 $\pm$0.25 | 73.25 $\pm$0.29 | 71.68 $\pm$0.37 | 67.31 $\pm$0.23 | 58.93 $\pm$0.35 | 52.01 $\pm$0.62 | 38.95 $\pm$1.24 |
| **EL2N** | 75.22 $\pm$0.09 | 74.23 $\pm$0.11 | 72.01 $\pm$0.18 | 48.19 $\pm$0.47 | 14.81 $\pm$0.14 | 8.68 $\pm$0.06 | 7.60 $\pm$0.18 |
| **AUM** | 75.26 $\pm$0.25 | 74.47 $\pm$0.31 | 71.96 $\pm$0.22 | 47.50 $\pm$1.39 | 15.35 $\pm$1.79 | 8.98 $\pm$1.37 | 5.47 $\pm$0.85 |
| **Moderate** | 75.25 $\pm$0.23 | 74.34 $\pm$0.31 | 72.80 $\pm$0.25 | 68.75 $\pm$0.40 | 60.98 $\pm$0.39 | 54.21 $\pm$0.93 | 38.72 $\pm$0.30 |
| **Dyn-Unc** | 75.22 $\pm$0.25 | 75.51 $\pm$0.22 | 75.09 $\pm$0.23 | 72.02 $\pm$0.07 | 62.17 $\pm$0.55 | 53.49 $\pm$0.47 | 35.44 $\pm$0.49 |
| **TDDS** | 73.29 $\pm$0.40 | 72.90 $\pm$0.31 | 71.83 $\pm$0.78 | 67.24 $\pm$0.92 | 57.30 $\pm$3.11 | 55.14 $\pm$1.21 | 41.58 $\pm$2.10 |
| **CCS** | 74.31 $\pm$0.14 | 73.04 $\pm$0.23 | 71.83 $\pm$0.25 | 67.61 $\pm$0.48 | 59.61 $\pm$0.64 | 53.35 $\pm$0.71 | 39.04 $\pm$1.14 |
| **DUAL** | **75.95** $\pm$0.19 | 75.66 $\pm$0.23 | **75.10** $\pm$0.23 | **72.64** $\pm$0.27 | 65.29 $\pm$0.64 | 57.55 $\pm$0.55 | 37.34 $\pm$1.70 |
| **DUAL+$\beta$ sampling** | 75.50 $\pm$0.21 | **75.78** $\pm$0.15 | **75.10** $\pm$0.13 | 72.08 $\pm$0.22 | **65.84** $\pm$0.37 | **62.20** $\pm$0.72 | **53.96** $\pm$0.35 |

*Table 11.* Comparison of test accuracy of DUAL pruning with existing coreset selection methods under 30% image corrupted data using ResNet-18 for CIFAR-100. The model trained with the full dataset achieves **73.77%** test accuracy on average. Results are averaged over five runs.

| Pruning Rate ($\rightarrow$) | 10% | 20% | 30% | 50% | 70% | 80% | 90% |
|---|---|---|---|---|---|---|---|
| **Random** | 72.71 $\pm$0.34 | 71.28 $\pm$0.31 | 69.84 $\pm$0.24 | 65.42 $\pm$0.33 | 56.72 $\pm$0.56 | 49.71 $\pm$0.65 | 35.75 $\pm$1.41 |
| **Entropy** | 72.94 $\pm$0.09 | 71.14 $\pm$0.14 | 68.74 $\pm$0.20 | 61.34 $\pm$0.59 | 42.70 $\pm$1.02 | 29.46 $\pm$1.68 | 12.55 $\pm$0.66 |
| **Forgetting** | 72.67 $\pm$0.21 | 71.22 $\pm$0.08 | 69.65 $\pm$0.45 | 65.25 $\pm$0.33 | 56.47 $\pm$0.31 | 49.07 $\pm$0.32 | 34.62 $\pm$1.15 |
| **EL2N** | 73.33 $\pm$0.08 | 71.99 $\pm$0.11 | 67.72 $\pm$0.50 | 37.57 $\pm$0.70 | 10.75 $\pm$0.28 | 9.08 $\pm$0.30 | 7.75 $\pm$0.08 |
| **AUM** | 73.73 $\pm$0.19 | 72.99 $\pm$0.22 | 70.93 $\pm$0.33 | 57.13 $\pm$0.42 | 28.98 $\pm$0.50 | 19.73 $\pm$0.28 | 12.18 $\pm$0.46 |
| **Moderate** | **74.02** $\pm$0.28 | 72.70 $\pm$0.30 | 71.51 $\pm$0.26 | 67.35 $\pm$0.16 | 59.47 $\pm$0.34 | 52.95 $\pm$0.60 | 37.45 $\pm$1.21 |
| **Dyn-Unc** | 73.86 $\pm$0.21 | 73.78 $\pm$0.20 | **73.78** $\pm$0.12 | 71.01 $\pm$0.23 | 61.56 $\pm$0.46 | 52.51 $\pm$1.08 | 35.47 $\pm$1.34 |
| **TDDS** | 71.58 $\pm$0.50 | 71.45 $\pm$0.68 | 69.92 $\pm$0.25 | 65.12 $\pm$2.08 | 55.79 $\pm$2.16 | 53.85 $\pm$0.94 | 40.51 $\pm$1.34 |
| **CCS** | 72.58 $\pm$0.12 | 71.38 $\pm$0.35 | 69.83 $\pm$0.26 | 65.45 $\pm$0.23 | 56.65 $\pm$0.45 | 49.75 $\pm$0.90 | 34.63 $\pm$1.79 |
| **DUAL** | 73.96 $\pm$0.20 | **74.07** $\pm$0.43 | 73.74 $\pm$0.18 | **71.23** $\pm$0.08 | 64.76 $\pm$0.32 | 57.47 $\pm$0.51 | 37.93 $\pm$2.38 |
| **DUAL+$\beta$ sampling** | 73.91 $\pm$0.17 | 73.80 $\pm$0.48 | 73.59 $\pm$0.19 | 71.12 $\pm$0.29 | **65.18** $\pm$0.44 | **61.07** $\pm$0.47 | **52.61** $\pm$0.47 |

*Table 12.* Comparison of test accuracy of DUAL pruning with existing coreset selection methods under 40% image corrupted data using ResNet-18 for CIFAR-100. The model trained with the full dataset achieves **72.16%** test accuracy on average. Results are averaged over five runs.

| Pruning Rate ($\rightarrow$) | 10% | 20% | 30% | 50% | 70% | 80% | 90% |
|---|---|---|---|---|---|---|---|
| **Random** | 70.78 $\pm$0.25 | 69.30 $\pm$0.29 | 67.98 $\pm$0.26 | 63.23 $\pm$0.26 | 53.29 $\pm$0.64 | 45.76 $\pm$0.85 | 32.63 $\pm$0.61 |
| **Entropy** | 70.74 $\pm$0.18 | 68.90 $\pm$0.37 | 66.19 $\pm$0.46 | 57.03 $\pm$0.60 | 35.62 $\pm$1.58 | 22.50 $\pm$1.03 | 7.46 $\pm$0.52 |
| **Forgetting** | 70.54 $\pm$0.10 | 69.17 $\pm$0.30 | 67.41 $\pm$0.28 | 62.77 $\pm$0.15 | 52.89 $\pm$0.36 | 44.94 $\pm$0.66 | 30.48 $\pm$0.49 |
| **EL2N** | 71.57 $\pm$0.28 | 69.24 $\pm$0.16 | 62.95 $\pm$0.52 | 28.33 $\pm$0.47 | 9.48 $\pm$0.21 | 8.86 $\pm$0.21 | 7.58 $\pm$0.16 |
| **AUM** | 71.66 $\pm$0.23 | 69.75 $\pm$0.30 | 62.10 $\pm$0.46 | 26.56 $\pm$0.62 | 8.93 $\pm$0.19 | 5.82 $\pm$0.09 | 4.15 $\pm$0.11 |
| **Moderate** | **72.10** $\pm$0.14 | 71.55 $\pm$0.25 | 69.84 $\pm$0.39 | 65.74 $\pm$0.21 | 56.96 $\pm$0.52 | 49.04 $\pm$0.74 | 34.87 $\pm$0.57 |
| **Dyn-Unc** | 71.86 $\pm$0.12 | 71.65 $\pm$0.18 | **71.79** $\pm$0.27 | 69.17 $\pm$0.44 | 59.69 $\pm$0.30 | 51.36 $\pm$0.70 | 34.02 $\pm$0.45 |
| **TDDS** | 70.02 $\pm$0.43 | 69.27 $\pm$0.74 | 68.03 $\pm$0.55 | 63.42 $\pm$0.77 | 55.28 $\pm$1.93 | 51.44 $\pm$1.36 | 38.42 $\pm$0.80 |
| **CCS** | 70.84 $\pm$0.41 | 69.08 $\pm$0.41 | 68.11 $\pm$0.09 | 63.36 $\pm$0.16 | 53.21 $\pm$0.54 | 46.27 $\pm$0.52 | 32.72 $\pm$0.52 |
| **DUAL** | 71.90 $\pm$0.27 | **72.38** $\pm$0.27 | **71.79** $\pm$0.11 | **69.69** $\pm$0.18 | 63.35 $\pm$0.29 | 56.57 $\pm$1.07 | 37.78 $\pm$0.73 |
| **DUAL+$\beta$ sampling** | 71.96 $\pm$0.13 | 71.92 $\pm$0.22 | 71.69 $\pm$0.18 | 69.23 $\pm$0.15 | **63.73** $\pm$0.43 | **59.75** $\pm$0.32 | **51.51** $\pm$0.68 |

*Table 13.* Comparison of test accuracy of DUAL pruning with existing coreset selection methods under 20% image corrupted data using ResNet-34 for Tiny-ImageNet. The model trained with the full dataset achieves **57.12%** test accuracy on average. Results are averaged over three runs.

| Pruning Rate ($\rightarrow$) | 10% | 20% | 30% | 50% | 70% | 80% | 90% |
|---|---|---|---|---|---|---|---|
| **Random** | 49.59 ±0.93 | 48.64 ±0.94 | 45.64 ±0.53 | 41.58 ±0.66 | 33.98 ±0.55 | 28.88 ±0.67 | 18.59 ±0.25 |
| **Entropy** | 50.34 ±0.19 | 48.02 ±0.49 | 44.80 ±0.30 | 36.58 ±0.19 | 25.20 ±0.53 | 16.55 ±0.40 | 3.32 ±0.26 |
| **Forgetting** | 46.81 ±0.26 | 41.16 ±0.28 | 35.58 ±0.17 | 26.80 ±0.18 | 17.66 ±0.23 | 12.61 ±0.04 | 6.01 ±0.19 |
| **EL2N** | 50.66 ±0.27 | 47.76 ±0.25 | 42.15 ±1.02 | 23.42 ±0.26 | 8.07 ±0.09 | 6.57 ±0.36 | 3.75 ±0.13 |
| **AUM** | 51.11 ±0.73 | 47.70 ±0.51 | 42.04 ±0.81 | 20.85 ±0.79 | 6.87 ±0.24 | 3.75 ±0.21 | 2.27 ±0.11 |
| **Moderate** | 51.43 ±0.76 | 49.85 ±0.23 | 47.85 ±0.31 | 42.31 ±0.40 | 35.00 ±0.49 | 29.63 ±0.67 | 19.51 ±0.72 |
| **Dyn-Unc** | 51.61 ±0.19 | 51.47 ±0.34 | **51.18** ±0.58 | **48.88** ±0.85 | 42.52 ±0.34 | 37.85 ±0.47 | 26.26 ±0.70 |
| **TDDS** | 51.53 ±0.40 | 49.81 ±0.21 | 48.98 ±0.27 | 45.81 ±0.16 | 38.05 ±0.70 | 33.04 ±0.39 | 22.66 ±1.28 |
| **CCS** | 50.26 ±0.78 | 48.00 ±0.41 | 45.38 ±0.63 | 40.98 ±0.23 | 33.49 ±0.04 | 27.18 ±0.66 | 15.37 ±0.54 |
| **DUAL** | 51.22 ±0.40 | **52.06** ±0.55 | 50.88 ±0.64 | 47.03 ±0.56 | 40.03 ±0.09 | 34.92 ±0.15 | 20.41 ±1.07 |
| **DUAL+$\beta$ sampling** | **52.15** ±0.25 | 51.11 ±0.34 | 50.21 ±0.36 | 46.85 ±0.27 | **42.97** ±0.28 | **38.30** ±0.06 | **27.45** ±0.50 |

## B.3. Cross-architecture generalization

In this section, we investigate the cross-architecture generalization ability of our proposed method. Specifically, we calculate the example score on one architecture and test its coreset performance on a different architecture. This evaluation, with results presented in Tables 14 through 18, aims to assess the transferability of these scores across diverse architectural designs.

*Table 14.* Cross-architecture generalization performance on CIFAR-100 from three layer CNN to ResNet-18. We report an average of five runs. 'R18 → R18' stands for score computation on ResNet-18, as a baseline.

| | 3-layer CNN → ResNet-18 | | | |
|---|---|---|---|---|
| Pruning Rate (→) | 30% | 50% | 70% | 90% |
| Random | 75.15 ±0.28 | 71.68 ±0.31 | 64.86 ±0.39 | 45.09 ±1.26 |
| EL2N | 76.56 ±0.65 | 71.78 ±0.32 | 56.57 ±1.32 | 22.84 ±3.54 |
| Dyn-Unc | **76.61** ±0.75 | 72.92 ±0.57 | 65.97 ±0.53 | 44.25 ±2.47 |
| CCS | 75.29 ±0.20 | 72.06 ±0.19 | **66.11** ±0.15 | 36.98 ±1.47 |
| DUAL | **76.61** ±0.08 | **73.55** ±0.12 | 65.97 ±0.18 | 39.00 ±2.51 |
| DUAL+$\beta$ sampling | 76.36 ±0.18 | 72.46 ±0.41 | 65.50 ±0.53 | **48.91** ±0.60 |
| DUAL (R18→R18) | 77.43 ±0.18 | 74.62 ±0.47 | 66.41 ±0.52 | 34.38 ±1.39 |
| DUAL (R18→R18) +$\beta$ sampling | 77.86 ±0.12 | 74.66 ±0.12 | 69.25 ±0.22 | 54.54 ±0.09 |

*Table 15.* Cross-architecture generalization performance on CIFAR-100 from three layer CNN to VGG-16. We report an average of five runs. 'V16 → V16' stands for score computation on VGG-16, as a baseline.

| | 3-layer CNN → VGG-16 | | | |
|---|---|---|---|---|
| Pruning Rate (→) | 30% | 50% | 70% | 90% |
| Random | 69.47 ±0.27 | 65.52 ±0.54 | 57.18 ±0.68 | 34.69 ±1.97 |
| EL2N | 70.35 ±0.64 | 63.66 ±1.49 | 46.12 ±6.87 | 20.85 ±9.03 |
| Dyn-Unc | 71.18 ±0.96 | 67.06 ±0.94 | 58.87 ±0.83 | 31.57 ±3.29 |
| CCS | 69.56 ±0.33 | 65.26 ±0.50 | 57.60 ±0.80 | 23.92 ±1.85 |
| DUAL | **71.75** ±0.16 | **67.91** ±0.27 | 59.08 ±0.64 | 29.16 ±2.28 |
| DUAL+$\beta$ sampling | 70.78 ±0.41 | 67.47 ±0.44 | **60.33** ±0.32 | **43.92** ±1.15 |
| DUAL (V16→V16) | 73.63 ±0.62 | 69.66 ±0.45 | 58.49 ±0.77 | 32.96 ±1.12 |
| DUAL (V16→V16) +$\beta$ sampling | 72.77 ±0.41 | 68.93 ±0.23 | 61.48 ±0.36 | 42.99 ±0.62 |

*Table 16.* Cross-architecture generalization performance on CIFAR-100 from VGG-16 to ResNet-18. We report an average of five runs. 'R18 → R18' stands for score computation on ResNet-18, as a baseline.

| | VGG-16 → ResNet-18 | | | |
|---|---|---|---|---|
| Pruning Rate (→) | 30% | 50% | 70% | 90% |
| Random | 75.15 ±0.28 | 71.68 ±0.31 | 64.86 ±0.39 | 45.09 ±1.26 |
| EL2N | 76.42 ±0.27 | 70.44 ±0.48 | 51.87 ±1.27 | 25.74 ±1.53 |
| Dyn-Unc | **77.59** ±0.19 | 74.20 ±0.22 | 65.24 ±0.36 | 42.95 ±1.14 |
| CCS | 75.19 ±0.19 | 71.56 ±0.28 | 64.83 ±0.25 | **46.08** ±1.23 |
| DUAL | 77.40 ±0.36 | **74.29** ±0.12 | 63.74 ±0.30 | 36.87 ±2.27 |
| DUAL+ $\beta$ sampling | 76.67 ±0.15 | 73.14 ±0.29 | **65.69** ±0.57 | 45.95 ±0.52 |
| DUAL (R18→R18) | 77.43 ±0.18 | 74.62 ±0.47 | 66.41 ±0.52 | 34.38 ±1.39 |
| DUAL (R18→R18) +$\beta$ sampling | 77.86 ±0.12 | 74.66 ±0.12 | 69.25 ±0.22 | 54.54 ±0.09 |

*Table 17.* Cross-architecture generalization performance on CIFAR-100 from ResNet-18 to VGG-16. We report an average of five runs.
'V16 → V16' stands for score computation on VGG-16, as a baseline.

| | ResNet-18 → VGG-16 | | | |
| --- | --- | --- | --- | --- |
| Pruning Rate (→) | 30% | 50% | 70% | 90% |
| Random | 70.99 ±0.33 | 67.34 ±0.21 | 60.18 ±0.52 | 41.69 ±0.72 |
| EL2N | 72.43 ±0.54 | 65.36 ±0.68 | 43.35 ±0.81 | 19.92 ±0.89 |
| Dyn-Unc | 73.34 ±0.29 | 69.24 ±0.39 | 57.67 ±0.52 | 31.74 ±0.80 |
| CCS | 71.18 ±0.16 | 67.35 ±0.38 | 59.77 ±0.43 | 41.06 ±1.03 |
| DUAL | 73.44 ±0.29 | 69.87 ±0.35 | 60.07 ±0.47 | 29.74 ±1.70 |
| DUAL +$\beta$ sampling | **73.50** ±0.27 | **70.43** ±0.26 | **64.48** ±0.47 | **49.61** ±0.49 |
| DUAL (V16→V16) | 73.63 ±0.61 | 69.66 ±0.45 | 58.49 ±0.77 | 32.96 ±1.12 |
| DUAL (V16→V16)+$\beta$ sampling | 72.66 ±0.17 | 68.80 ±0.34 | 60.40 ±0.68 | 41.51 ±0.47 |

*Table 18.* Cross-architecture generalization performance on CIFAR-100 from VGG-16 to ResNet-50. We report an average of five runs.
'R50 → R50' stands for score computation on ResNet-50, as a baseline

| | VGG-16 → ResNet-50 | | | |
| --- | --- | --- | --- | --- |
| Pruning Rate (→) | 30% | 50% | 70% | 90% |
| Random | 71.13 ±6.52 | 70.31 ±1.20 | 61.02 ±1.68 | 41.03 ±3.74 |
| EL2N | 76.30 ±0.69 | 67.11 ±3.09 | 44.88 ±3.65 | 25.05 ±1.76 |
| Dyn-Unc | **77.91** ±0.54 | **73.52** ±0.41 | 62.37 ±0.62 | 39.10 ±4.04 |
| CCS | 75.40 ±0.64 | 70.44 ±0.49 | 60.10 ±1.24 | 41.94 ±3.01 |
| DUAL | 77.50 ±0.53 | 71.81 ±0.48 | 60.68 ±1.67 | 34.88 ±3.47 |
| DUAL +$\beta$ sampling | 76.67 ±0.15 | 73.14 ±0.29 | **65.69** ±0.57 | **45.95** ±0.52 |
| DUAL (R50→R50) | 77.82 ±0.64 | 73.66 ±0.85 | 52.12 ±2.73 | 26.13 ±1.96 |
| DUAL (R50→R50)+$\beta$ sampling | 77.57 ±0.23 | 73.44 ±0.87 | 65.17 ±0.96 | 47.63 ±2.47 |

## B.4. Image Classification on Long-tailed Distributions

We also conducted experiments on long-tailed versions of CIFAR-10 and CIFAR-100, following the procedure of Cao et al. (2019). These long-tailed datasets were constructed using an imbalance ratio, $\rho$, defined as the ratio between the sample sizes of the most frequent class ($n_{\max}$) and the least frequent class ($n_{\min}$), i.e. $\rho = n_{\max}/n_{\min}$. The class distribution in this long-tailed setup exhibits an exponential decay in sample sizes across classes. For all pruning ratios, we compared our method against several baselines: Random, EL2N, Dyn-Unc, and CCS. The results presented in Table 19 and 20 demonstrate that DUAL pruning (along with the Beta sampling) achieves superior performance compared to other baselines.

*Table 19.* Test accuracy on long-tailed imbalance on CIFAR-10. The test accuracy on a full dataset is 89.98 ($\rho = 10$) and 75.03 ($\rho = 100$). We report the average performance across three runs.

| | CIFAR-10-LT | | | | | | | | | |
|---|---|---|---|---|---|---|---|---|---|---|
| Imbalance Ratio | 10 | | | | | 100 | | | | |
| Pruning Rate | 30% | 50% | 70% | 80% | 90% | 30% | 50% | 70% | 80% | 90% |
| Random | 42.48 ±0.45 | 28.20 ±0.07 | 18.85 ±0.24 | 10.00 ±0.00 | 10.00 ±0.00 | 28.23 ±0.09 | 19.36 ±0.18 | 10.00 ±0.00 | 10.00 ±0.00 | 10.00 ±0.00 |
| EL2N | 89.42 ±0.20 | 87.59 ±0.97 | 68.15 ±3.44 | 52.90 ±1.87 | 33.25 ±0.41 | 72.70 ±1.58 | 66.06 ±4.27 | 52.90 ±2.88 | 41.79 ±2.61 | 30.30 ±0.58 |
| Dyn-Unc | 89.64 ±0.28 | 87.60 ±0.39 | 67.60 ±4.34 | 53.05 ±0.88 | 39.16 ±1.94 | **74.40** ±1.32 | **70.22** ±1.60 | 51.89 ±3.08 | 41.27 ±2.34 | 31.24 ±0.23 |
| CCS | 84.42 ±0.89 | 73.04 ±1.20 | 47.07 ±0.68 | 37.38 ±0.36 | 27.91 ±0.96 | 63.18 ±1.56 | 45.46 ±1.33 | 32.66 ±0.63 | 29.38 ±0.71 | 24.10 ±0.97 |
| DUAL | **89.67** ±0.40 | **88.75** ±0.36 | 75.38 ±3.41 | 56.70 ±2.83 | 43.58 ±2.45 | 72.94 ±1.14 | 69.66 ±0.73 | 52.80 ±1.00 | 38.32 ±1.28 | 25.30 ±1.28 |
| DUAL + $\beta$ | 89.49 ±0.21 | 88.12 ±0.61 | **76.00** ±2.79 | **78.31** ±2.26 | **71.27** ±1.44 | 73.81 ±2.06 | 68.89 ±0.24 | **52.95** ±2.79 | **46.49** ±1.80 | **36.43** ±1.00 |

*Table 20.* Test accuracy on long-tailed imbalance on CIFAR-100. The test accuracy on a full dataset is 62.92 ($\rho = 10$) and 41.67 ($\rho = 100$). We report the average performance across three runs.

| | CIFAR-100-LT | | | | | | | | | |
|---|---|---|---|---|---|---|---|---|---|---|
| Imbalance Ratio | 10 | | | | | 100 | | | | |
| Pruning Rate | 30% | 50% | 70% | 80% | 90% | 30% | 50% | 70% | 80% | 90% |
| Random | 32.89 ±0.23 | 18.79 ±0.75 | 8.26 ±0.41 | 5.43 ±0.07 | 3.23 ±0.05 | 22.88 ±0.87 | 11.45 ±0.11 | 5.90 ±0.15 | 3.96 ±0.05 | 2.48 ±0.02 |
| EL2N | 57.57 ±0.50 | 47.23 ±0.46 | 21.38 ±0.33 | 13.92 ±0.97 | 9.54 ±0.19 | 37.59 ±2.13 | 24.76 ±1.87 | 12.33 ±0.54 | 9.42 ±0.26 | 6.64 ±0.02 |
| Dyn-Unc | 58.09 ±0.85 | 46.68 ±0.69 | 25.95 ±2.16 | 20.80 ±0.68 | 13.48 ±0.50 | **37.82** ±1.08 | 26.88 ±0.38 | 15.41 ±0.42 | 12.47 ±0.55 | 9.52 ±0.12 |
| CCS | 46.51 ±0.56 | 34.85 ±0.79 | 18.08 ±0.80 | 11.34 ±0.30 | 6.06 ±0.43 | 27.46 ±0.27 | 17.85 ±0.76 | 11.43 ±0.33 | 8.25 ±0.66 | 4.34 ±0.53 |
| DUAL | **58.50** ±0.27 | 54.11 ±0.27 | 39.15 ±1.43 | 30.10 ±0.97 | 18.80 ±1.17 | 36.35 ±0.66 | 30.19 ±1.58 | 20.47 ±0.30 | 17.76 ±0.47 | 12.52 ±0.56 |
| DUAL + $\beta$ | 58.05 ±0.34 | **54.88** ±0.36 | **43.53** ±0.66 | **35.87** ±1.75 | **27.13** ±1.49 | 37.04 ±0.97 | **32.25** ±0.45 | **21.94** ±1.27 | **19.38** ±0.77 | **15.42** ±0.32 |

**B.5. Comparison with Dynamic Pruning Methods**

In this section, we present several experiments comparing recent dynamic pruning methods, such as those by Yuan et al. (2025) and Qin et al. (2024), with static approaches, including DUAL pruning. We first highlight two key differences between static and dynamic data pruning.

- Compared to static pruning, dynamic pruning maintains access to the entire original dataset throughout training, allowing it to fully leverage all available information in the original dataset.

- While both aim to improve training efficiency, their underlying goals differ slightly. Static data pruning seeks to identify a "fixed" subset that reduces the dataset size while preserving as much information about the original dataset as possible. This subset can then serve as a new, independent dataset, reusable across various model architectures and experimental setups. In contrast, dynamic data pruning enhances training efficiency within a single training session by pruning data dynamically on the fly. However, this approach requires storing the entire original dataset, making dynamic pruning less memory-efficient and not reusable.

**Standard Training.**   We conducted experiments on CIFAR-10 and CIFAR-100 with ResNet-18, using the same hyperparameters as described in Section 4. We first tested dynamic random pruning, which dynamically prunes randomly selected samples from the entire dataset at each epoch. Notably, dynamic random pruning significantly outperformed all static baselines, achieving test accuracies of 91.82% on CIFAR-10 and 72.8% on CIFAR-100 at a pruning ratio of 90%. We also evaluated the methods in Yuan et al. (2025); Qin et al. (2024) and the results are provided in Figure 17. Overall, dynamic methods consistently outperform static baselines. However, at lower pruning ratios (e.g., on CIFAR-10), DUAL can outperform dynamic methods under a similar computational budget.

We believe this performance gap stems from differences in accessible information: static methods are limited to 10% of the data, while dynamic methods use the full dataset. Consequently, the performance gap widens even further at aggressive pruning ratios. To validate this, we plot how often each sample was seen during training. The plot in Figure 18 shows that static methods are confined to a subset, while dynamic ones use nearly all data—rendering direct comparison somewhat unfair. Indeed, dynamic pruning methods might be better compared with scheduled batch-selection approaches, such as curriculum learning, rather than static pruning methods.

**Label Noise Setting.**   We also evaluated these methods under label noise conditions. In fact, Yuan et al. (2025) conclude that IES cannot prune any samples (corrupted or not) when label noise is introduced. Similarly, InfoBatch (Qin et al., 2024) tends to retain harder (and often noisy) samples, as it removes only easy examples during training. In contrast, DUAL effectively filters noisy samples, improving performance even beyond full-data training.

We conducted experiments on CIFAR-100 with a 40% label noise setting (full-train test accuracy: 52.74%) to verify this explanation. DUAL achieves over 70% test accuracy at a 50% pruning ratio as can be seen in Table 8 in Appendix B.1, whereas InfoBatch achieves only 51.24% accuracy with a similar number of iterations. Under similar iterations, random dynamic pruning achieves 51.81% test accuracy, which still outperforms random static pruning. Lastly, IES (Yuan et al., 2025) prunes only 1.7% of samples during training (consistent with the original report in their paper), resulting in 51.95% test accuracy. Furthermore, our static method can create fixed subsets in which nearly all noisy samples have been removed, resulting in high-quality datasets that can be preserved for future use.

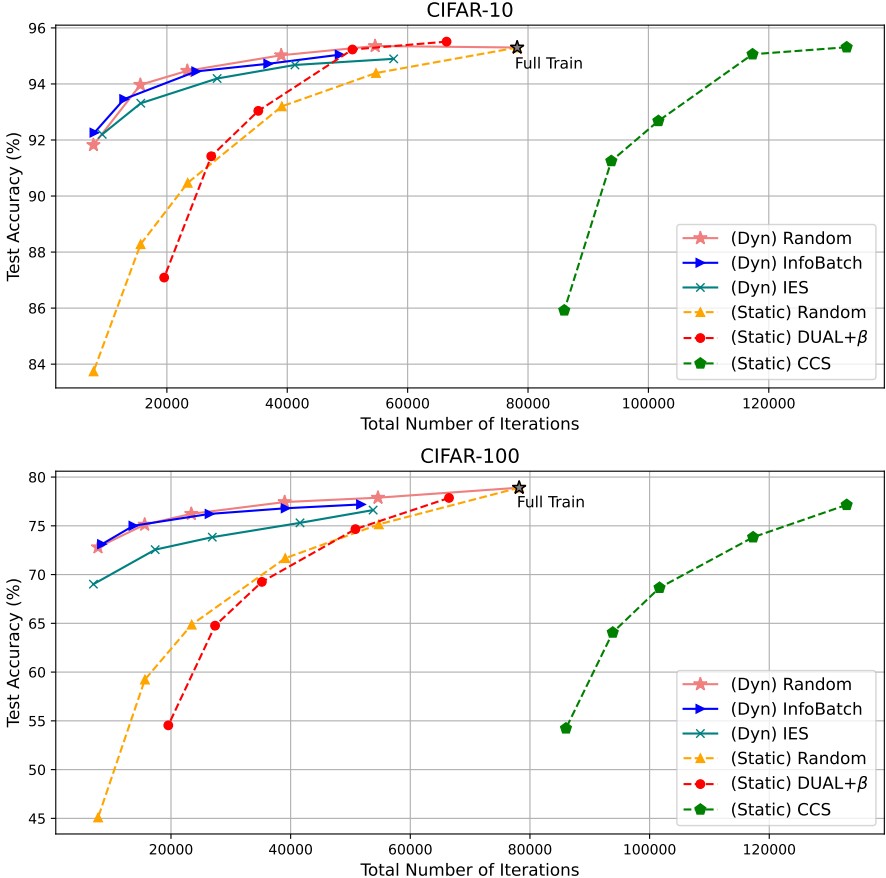

*Figure 17.* This figure shows test accuracy (y-axis) versus the total number of iterations needed to fully train each subset (x-axis). The top figure corresponds to CIFAR-10, and the bottom to CIFAR-100. Results are averaged over five random seeds. For the static pruning methods, we plotted the number of iterations and test accuracy at pruning ratios of 30%, 50%, 70%, 80%, and 90%, which correspond to the markers on each line from right to left. We adjust the total training epochs of InfoBatch following the procedure in the original paper, in order to evaluate its performance when the total number of iterations is reduced. For IES, we adjust the pruning threshold to evaluate its performance under a reduced total number of training iterations. Results demonstrate that dynamic pruning methods—including even random dynamic pruning—outperform static baselines in accuracy while requiring fewer iterations. Among static methods, DUAL is both the most efficient and the best-performing. However, it does not achieve as favorable a time-performance trade-off as dynamic methods. Still, at low pruning ratios—where more information is available—DUAL performs comparably to dynamic approaches.

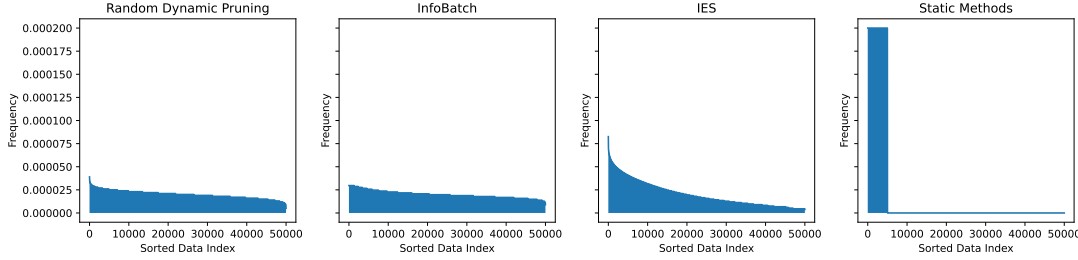

*Figure 18.* This figure shows the selection frequency of each sample at a high pruning ratio (90%). The x-axis represents data indices sorted in descending order by selection frequency, and the y-axis indicates the normalized selection ratio—i.e., how often each data point was selected, divided by the total number of data points seen during training (values sum to 1). Methods like Random Dynamic Pruning, InfoBatch, and IES use nearly the entire dataset during training, making them incomparable to static pruning methods, which are restricted to only 10% of the data.

## B.6. Ablation Study on Beta Sampling

We study the impact of our Beta sampling on existing score metrics. We apply our Beta sampling strategy to forgetting, EL2N, and Dyn-Unc scores of CIFAR10 and 100. By comparing Beta sampling with the vanilla threshold pruning using scores, with results presented in Tables 21 and 22, we observe that prior score-based methods become competitive, outperforming random pruning when Beta sampling is adjusted.

*Table 21.* Comparison on CIFAR-10 and CIFAR-100 for $90\%$ pruning rate. We report average accuracy with five runs. The best performance is in bold in each column.

| | CIFAR-10 | | CIFAR-100 | |
|---|---|---|---|---|
| Method | Thresholding | $\beta$-Sampling | Thresholding | $\beta$-Sampling |
| Random | **83.74** $\pm$ 0.21 | 83.31 (-0.43) $\pm$ 0.14 | **45.09** $\pm$ 1.26 | 51.76 (+6.67) $\pm$ 0.25 |
| EL2N | 38.74 $\pm$ 0.75 | 87.00 (+48.26) $\pm$ 0.45 | 8.89 $\pm$ 0.28 | 53.97 (+45.08) $\pm$ 0.63 |
| Forgetting | 46.64 $\pm$ 1.90 | 85.67 (+39.03) $\pm$ 0.13 | 26.87 $\pm$ 0.73 | 52.40 (+25.53) $\pm$ 0.43 |
| Dyn-Unc | 59.67 $\pm$ 1.79 | 85.33 (+32.14) $\pm$ 0.20 | 34.57 $\pm$ 0.69 | 51.85 (+17.28) $\pm$ 0.35 |
| DUAL | 54.95 $\pm$ 0.42 | **87.09** (+31.51) $\pm$ 0.36 | 34.28 $\pm$ 1.39 | **54.54** (+20.26) $\pm$ 0.09 |

*Table 22.* Comparison on CIFAR-10 and CIFAR-100 for $80\%$ pruning rate. We report average accuracy with five runs. The best performance is in bold in each column.

| | CIFAR-10 | | CIFAR-100 | |
|---|---|---|---|---|
| Method | Thresholding | $\beta$-Sampling | Thresholding | $\beta$-Sampling |
| Random | **88.28** $\pm$ 0.17 | 88.83 (+0.55) $\pm$ 0.18 | **59.23** $\pm$ 0.62 | 61.74 (+2.51) $\pm$ 0.15 |
| EL2N | 74.70 $\pm$ 0.45 | 87.69 (+12.99) $\pm$ 0.98 | 19.52 $\pm$ 0.79 | 63.98 (+44.46) $\pm$ 0.73 |
| Forgetting | 75.47 $\pm$ 1.27 | 90.86 (+15.39) $\pm$ 0.07 | 39.09 $\pm$ 0.41 | 63.29 (+24.20) $\pm$ 0.13 |
| Dyn-Unc | 83.32 $\pm$ 0.94 | 90.80 (+7.48) $\pm$ 0.30 | 55.01 $\pm$ 0.55 | 62.31 (+7.30) $\pm$ 0.23 |
| DUAL | 82.02 $\pm$ 1.85 | **91.42** (+9.68) $\pm$ 0.35 | 56.57 $\pm$ 0.57 | **64.76** (+8.46) $\pm$ 0.23 |

We also study the impact of our pruning strategy with DUAL score combined with Beta sampling. We compare different sampling strategies: vanilla thresholding, stratified sampling (Zheng et al., 2022), and Beta sampling on CIFAR10 and 100, at 80% and 90% pruning rates. The results, presented in Table 23 indicate that our proposed Beta sampling mostly performs the best, especially with the high pruning ratio.

*Table 23.* Comparison on Sampling Strategy

| CIFAR10 | | | | | |
|---|---|---|---|---|---|
| Pruning Rate | 30% | 50% | 70% | 80% | 90% |
| DUAL | 95.35 | 95.08 | 91.95 | 81.74 | 55.58 |
| DUAL + CCS | **95.54** | 95.00 | 92.83 | 90.49 | 81.67 |
| DUAL + $\beta$ | 95.51 | **95.23** | **93.04** | **91.42** | **87.09** |

| CIFAR100 | | | | | |
|---|---|---|---|---|---|
| Pruning Rate | 30% | 50% | 70% | 80% | 90% |
| DUAL | 77.61 | **74.86** | 66.39 | 56.50 | 34.28 |
| DUAL + CCS | 75.21 | 71.53 | 64.30 | 59.09 | 45.21 |
| DUAL + $\beta$ | **77.86** | 74.66 | **69.25** | **64.76** | **54.54** |

# C. Detailed Explanation about Beta Sampling

Here, we provide details on our choice of Beta sampling. Appendix C.1 shows the visualization of the selected data using the Beta sampling. Appendix C.2 presents the full algorithm for our DUAL pruning method with the suggested Beta sampling strategy.

We begin by explaining why the Beta distribution is selected as the sampling distribution. The domain of Beta distribution is $[0, 1]$, which naturally aligns with the range of prediction means. Moreover, the probability density function (PDF) of the Beta distribution can be shaped to decay at both tails, ensuring that samples with extreme scores are rarely selected. While other distributions, such as Gaussian, could also be considered, their support spans $\mathbb{R}$. As a result, they may assign non-negligible probability to values far outside the desired range unless their standard deviation is made extremely small.

We define our sampling distribution $\text{Beta}(\alpha_r, \beta_r)$ as follows:

$$\beta_r = 15 \left(1 - \mu_{\mathcal{D}}\right) \left(1 - r^{c_{\mathcal{D}}}\right)$$
$$\alpha_r = 15 - \beta_r, \tag{6}$$

where $\mu_{\mathcal{D}} \in [0, 1]$ is the probability mean of the highest DUAL score training sample. To ensure stability, we compute this as the average probability mean of the 10 highest DUAL score training samples. Additionally, as mentioned in Appendix A.2, we set the value of $C$ to 15 across all experiments. Recall that the variance of a Beta distribution with parameters is given by $\frac{\alpha\beta}{(\alpha+\beta)^2(\alpha+\beta+1)}$, so increasing $C$ leads to a lower variance, an effect illustrated in Figure 19. This enables a more concentrated sampling distribution in a specific region, improving the effectiveness of the sampling process by reducing unnecessary spread. In an implementation, we add 1 to $\alpha_r$ to provide a more targeted sampling.

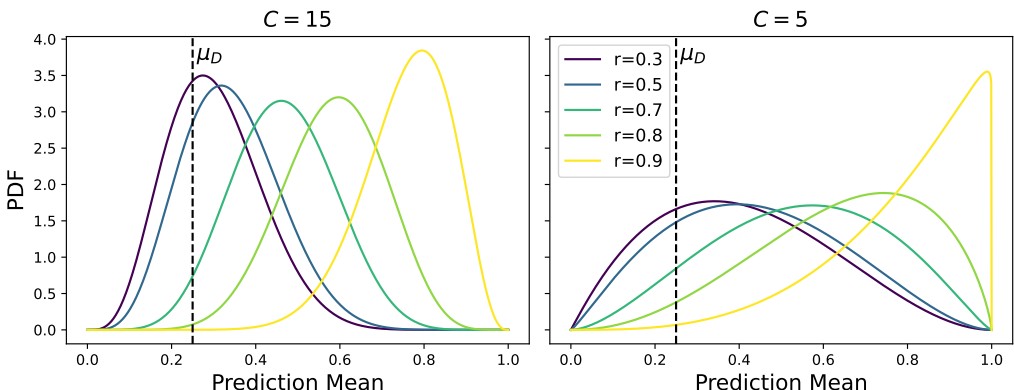

*Figure 19.* Visualization of Beta distribution for varying $C$. Large $C$ enables more concentrated targeting.

Now we justify our choice of parameters $\alpha_r$ and $\beta_r$ in the Beta distribution. When the pruning ratio is set to zero, $\alpha_r$ and $\beta_r$ are configured so that the mean of the Beta distribution (which is $\frac{\alpha}{\alpha+\beta}$) matches the prediction mean of the highest-scoring sample. This allows the sampling to focus on high-score samples at low pruning ratios. To gradually include easier samples as the pruning ratio increases, we set parameters $\alpha_r$ and $\beta_r$ depending on the pruning ratio. While BOSS (Acharya et al., 2024) adjusts these parameters such that the mode of the Beta distribution (which is $\frac{\alpha-1}{\alpha+\beta-2}$ for $\alpha, \beta > 1$) scales linearly with the pruning ratio $r$, we adopt a non-linear scaling by raising $r$ to the power of $c_D$. This results in a PDF that is remains almost stationary at low pruning ratios, but gradually shifts toward easier samples in a polynomial manner as the pruning ratio increases.

The hyperparameter $c_D$ is chosen based on the relative complexity of the dataset. We assumed that a larger dataset, which may have more samples per class, tends to be relatively easier. A higher value of $c_D$ results in smaller $\beta_r$, thereby increasing the mean and reducing the variance of the Beta distribution. For more difficult datasets, sampling easier examples becomes more important, which justifies using a larger $c_D$. Figure 20 illustrates the Beta PDF for different values of $c_D$. In both subplots, we set $\mu_{\mathcal{D}}$ as 0.25. The left subplot shows the PDF with $c_{\mathcal{D}} = 5.5$, which corresponds to the value used in our CIFAR-10 experiments, while the right subplot shows the case where $c_{\mathcal{D}} = 4$, which is used in CIFAR-100.

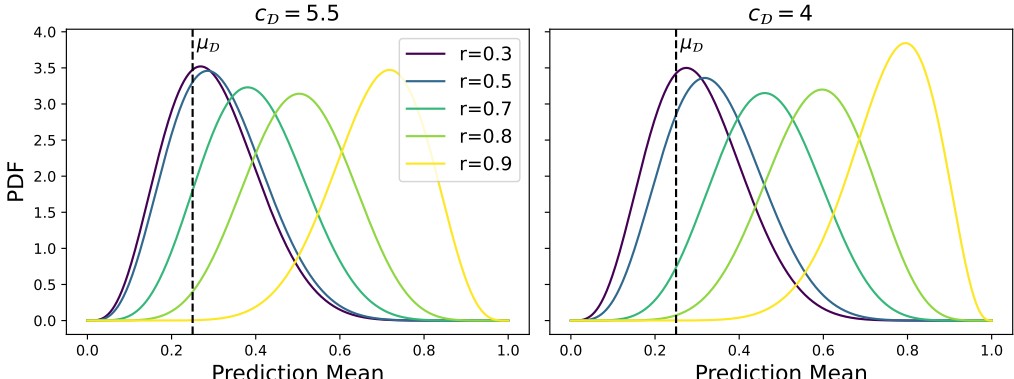

*Figure 20.* Visualization of Beta distribution for varying $c_\mathcal{D}$. Left subplot corresponds to the value used in CIFAR-10, and the right subplot corresponds to the value used in CIFAR-100.

## C.1. Visualization of Selected Data with Beta Sampling

We provide visualizations of Beta sampling in Figure 21. This figure illustrates, in respective columns: (i) sample selection probabilities for the coreset; (ii) examples of selected samples when forming a 70%, 30% and 10% coreset; and (iii) the samples pruned at these corresponding rates. The visualization shows that Beta sampling increasingly favors the selection of easier samples for the coreset as the pruning ratio increases.

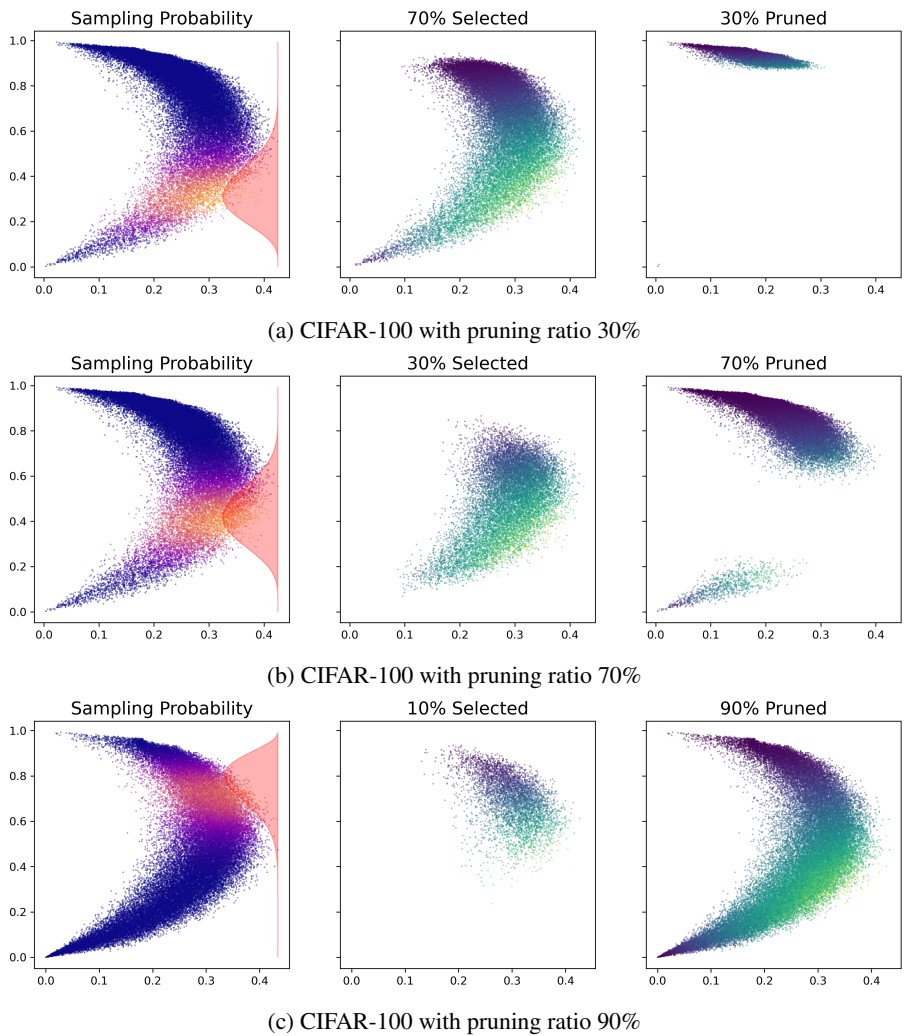

(a) CIFAR-100 with pruning ratio 30%

(b) CIFAR-100 with pruning ratio 70%

(c) CIFAR-100 with pruning ratio 90%

*Figure 21.* Pruning visualization on CIFAR-100.

## C.2. Algorithm of Proposed Pruning Method

The detailed algorithms for DUAL pruning and Beta sampling are as follows:

---

**Algorithm 1** DUAL pruning + $\beta$ sampling

---

**input** Training dataset $\mathcal{D}$, pruning ratio $r$, dataset simplicity $c_{\mathcal{D}}$, training epoch $T$, window length $J$.
**output** Subset $\mathcal{S} \subset \mathcal{D}$ such that $|\mathcal{S}| = (1-r)|\mathcal{D}|$
  **for** $(\boldsymbol{x}_i, y_i) \in \mathcal{D}$ **do**
    **for** $k = 1, \cdots, T - J + 1$ **do**
      $\bar{\mathbb{P}}_k(\boldsymbol{x}_i, y_i) \leftarrow \frac{1}{J} \sum_{j=0}^{J-1} \mathbb{P}_{k+j}(y_i \mid \boldsymbol{x}_i)$ {Example Difficulty}
      $\mathbb{U}_k(\boldsymbol{x}_i, y_i) \leftarrow \sqrt{\frac{1}{J-1} \sum_{j=0}^{J-1} \left[\mathbb{P}_{k+j}(y_i \mid \boldsymbol{x}_i) - \bar{\mathbb{P}}_k(\boldsymbol{x}_i, y_i)\right]^2}$ {Prediction Uncertainty}
      $\text{DUAL}_k(\boldsymbol{x}_i, y_i) \leftarrow (1 - \bar{\mathbb{P}}_k(\boldsymbol{x}_i, y_i)) \times \mathbb{U}_k(\boldsymbol{x}_i, y_i)$
    **end for**
    $\text{DUAL}(\boldsymbol{x}_i, y_i) \leftarrow \frac{1}{T-J+1} \sum_{k=1}^{T-J+1} \text{DUAL}_k(\boldsymbol{x}_i, y_i)$
  **end for**
  **if** $\beta$-sampling **then**
    **for** $(\boldsymbol{x}_i, y_i) \in \mathcal{D}$ **do**
      $\bar{\mathbb{P}}(\boldsymbol{x}_i, y_i) \leftarrow \frac{1}{T} \sum_{k=1}^{T} \mathbb{P}_k(y_i \mid \boldsymbol{x}_i)$
      $\varphi\left(\bar{\mathbb{P}}(\boldsymbol{x}_i, y_i)\right) \leftarrow$ PDF value of $\text{Beta}(\alpha_r, \beta_r)$ from Equation (5)
      $\tilde{\varphi}(\boldsymbol{x}_i) \leftarrow \varphi\left(\bar{\mathbb{P}}(\boldsymbol{x}_i, y_i)\right) \times \text{DUAL}(\boldsymbol{x}_i, y_i)$
    **end for**
    $\tilde{\varphi}(\boldsymbol{x}_i) \leftarrow \frac{\tilde{\varphi}(\boldsymbol{x}_i)}{\sum_{j \in \mathcal{D}} \tilde{\varphi}(\boldsymbol{x}_j)}$
    $\mathcal{S} \leftarrow$ Sample $(1-r)|\mathcal{D}|$ data points according to $\tilde{\varphi}(\boldsymbol{x}_i)$
  **else**
    $\mathcal{S} \leftarrow$ Sample $(1-r)|\mathcal{D}|$ data points with the largest $\text{DUAL}(\boldsymbol{x}_i, y_i)$ score
  **end if**

---

# D. Theoretical Results

Throughout this section, we will rigorously prove Theorem 3.1, providing the intuition that Dyn-Unc takes longer than our method to select informative samples.

## D.1. Proof of Theorem 3.1

Assume that the input and output (or label) space are $\mathcal{X} = \mathbb{R}^n$ and $\mathcal{Y} = \{\pm 1\}$, respectively. Let the model $f : \mathcal{X} \to \mathbb{R}$ be of the form $f(\boldsymbol{x}; \boldsymbol{w}) = \boldsymbol{w}^\top \boldsymbol{x}$ parameterized by $\boldsymbol{w} \in \mathbb{R}^n$ with zero-initialization. Let the loss be the exponential loss, $\ell(z) = e^{-z}$. Exponential loss is reported to induce implicit bias similar to logistic loss in binary classification tasks using linearly separable datasets (Soudry et al., 2018; Gunasekar et al., 2018).

The task of the model is to learn a binary classification. The dataset $\mathcal{D}$ consists only two points, i.e. $\mathcal{D} = \{(\boldsymbol{x}_1, y_1^*), (\boldsymbol{x}_2, y_2^*)\}$, where without loss of generality $y_i^* = 1$ for $i = 1, 2$. The model learns from $\mathcal{D}$ with the gradient descent. The update rule, equipped with a learning rate $\eta > 0$, is:

$$\boldsymbol{w}_0 = 0$$

$$\boldsymbol{w}_{t+1} = \boldsymbol{w}_t - \eta \nabla_{\boldsymbol{w}} \left[ \sum_{i=1}^{2} \ell\left(f\left(\boldsymbol{x}_i; \boldsymbol{w}_t\right)\right) \right]$$

$$= \boldsymbol{w}_t + \eta \left( e^{-\boldsymbol{w}_t^\top \boldsymbol{x}_1} \boldsymbol{x}_1 + e^{-\boldsymbol{w}_t^\top \boldsymbol{x}_2} \boldsymbol{x}_2 \right).$$

For brevity, denote the model output of the $i$-th data point at the $t$-th epoch as $y_t^{(i)} := f(\boldsymbol{x}_i; \boldsymbol{w}_t)$. The update rule for the parameter is simplified as:

$$\boldsymbol{w}_{t+1} = \boldsymbol{w}_t + \eta \left( e^{-y_t^{(1)}} \boldsymbol{x}_1 + e^{-y_t^{(2)}} \boldsymbol{x}_2 \right). \tag{7}$$

We also derive the update rule of model output for each instance:

$$\begin{cases} y_{t+1}^{(1)} = \boldsymbol{w}_{t+1}^\top \boldsymbol{x}_1 = \left( \boldsymbol{w}_t + \eta \left( e^{-y_t^{(1)}} \boldsymbol{x}_1 + e^{-y_t^{(2)}} \boldsymbol{x}_2 \right) \right)^\top \boldsymbol{x}_1 \\ \quad = y_t^{(1)} + \eta e^{-y_t^{(1)}} \|\boldsymbol{x}_1\|^2 + \eta e^{-y_t^{(2)}} \langle \boldsymbol{x}_1, \boldsymbol{x}_2 \rangle, \\ y_{t+1}^{(2)} = y_t^{(2)} + \eta e^{-y_t^{(2)}} \|\boldsymbol{x}_2\|^2 + \eta e^{-y_t^{(1)}} \langle \boldsymbol{x}_1, \boldsymbol{x}_2 \rangle. \end{cases} \tag{8}$$

Assume that $\boldsymbol{x}_2$ is farther from the origin in terms of distance than $\boldsymbol{x}_1$ is, but not too different in terms of angle. Formally,

**Assumption D.1.** $\|\boldsymbol{x}_2\| > 1$, $4\|\boldsymbol{x}_1\|^2 < 2\langle \boldsymbol{x}_1, \boldsymbol{x}_2 \rangle < \|\boldsymbol{x}_2\|^2$. Moreover, $\langle \boldsymbol{x}_1, \boldsymbol{x}_2 \rangle < \|\boldsymbol{x}_1\|\|\boldsymbol{x}_2\|$.

Under these assumptions, as $\langle \boldsymbol{x}_1, \boldsymbol{x}_2 \rangle > 0$, $\mathcal{D}$ is linearly separable. Also, notice that $\boldsymbol{x}_1$ and $\boldsymbol{x}_2$ are not parallel. Our definition of a linearly separable dataset is in accordance with Soudry et al. (2018). A dataset $\mathcal{D}$ is linearly separable if there exists $\boldsymbol{w}^*$ such that $\langle \boldsymbol{x}_i, \boldsymbol{w}^* \rangle > 0, \forall i$.

**Theorem D.2.** *Let $V_{t;J}^{(i)}$ be the variance and $\mu_{t;J}^{(i)}$ be the mean of $\sigma(y_t^{(i)})$ within a window from time $t$ to $t + J - 1$. Denote $T_v$ and $T_{vm}$ as the first time when $V_{t;J}^{(1)} > V_{t;J}^{(2)}$ and $V_{t;J}^{(1)}(1 - \mu_{t;J}^{(i)}) > V_{t;J}^{(2)}(1 - \mu_{t;J}^{(2)})$ occurs, respectively. Under Assumption D.1, if $\eta$ is sufficiently small then $T_{vm} < T_v$.*

By Soudry et al. (2018), the learning is progressed as: $\boldsymbol{w}_t$, $y_t^{(1)}$, and $y_t^{(2)}$ diverges to positive infinity (Lemma 1) but $\boldsymbol{w}_t$ directionally converges towards $L_2$ max margin vector, $\hat{\boldsymbol{w}} = \boldsymbol{x}_1 / \|\boldsymbol{x}_1\|^2$, or $\lim_{t \to \infty} \frac{\boldsymbol{w}_t}{\|\boldsymbol{w}_t\|} = \frac{\hat{\boldsymbol{w}}}{\|\hat{\boldsymbol{w}}\|}$ (Theorem 3). Moreover, the growth of $\boldsymbol{w}$ is logarithmic, i.e. $\boldsymbol{w}_t \approx \hat{\boldsymbol{w}} \log t$. We hereby note that Theorem 3 of Soudry et al. (2018) holds for learning rate $\eta$ smaller than a global constant. Since our condition requires $\eta$ to be sufficiently small, we will make use of the findings of Theorem 3.

**Lemma D.3.** $\Delta y_t := y_t^{(2)} - y_t^{(1)}$ *is a non-negative, strictly increasing sequence. Also, $\lim_{t \to \infty} \Delta y_t = \infty$.*

*Proof.*

1) Since $\boldsymbol{w}_0 = 0$, $y_0^{(1)} = 0 = y_0^{(2)}$ so $\Delta y_0 = 0$. By Equation (8) and Assumption D.1, $\Delta y_1 = y_1^{(2)} - y_1^{(1)} = \eta \left( \|\boldsymbol{x}_2\|^2 - \|\boldsymbol{x}_1\|^2 \right) > 0$.

2)

$$\Delta y_{t+1} - \Delta y_t = \eta \left[ e^{-y_t^{(2)}} \left( \|\boldsymbol{x}_2\|^2 - \langle \boldsymbol{x}_1, \boldsymbol{x}_2 \rangle \right) + e^{-y_t^{(1)}} \left( \langle \boldsymbol{x}_1, \boldsymbol{x}_2 \rangle - \|\boldsymbol{x}_1\|^2 \right) \right]$$
$$=: K_1 e^{-y_t^{(1)}} + K_2 e^{-y_t^{(2)}} > 0,$$

for some positive constant $K_1, K_2$. As $y_t^{(i)} = \boldsymbol{w}_t^\top \boldsymbol{x}_i$ would logarithmically grow in terms of $t$, $e^{-y_t^{(i)}}$ is decreasing in $t$. Moreover, as $y_t^{(1)} = \boldsymbol{w}_t^\top \boldsymbol{x}_1 \approx \hat{\boldsymbol{w}}^\top \boldsymbol{x}_1 \log t = \log t$, $e^{-y_t^{(1)}}$ is (asymptotically) in scale of $t^{-1}$ and so is $\Delta y_{t+1} - \Delta y_t$. Hence, $\{\Delta y_t\}$ is non-negative and increases to infinity. $\qquad \square$

The notation $\Delta y_t := y_t^{(2)} - y_t^{(1)}$ will be used throughout this section. Next, we show that, under Assumption D.1, $y_{t+1}^{(1)} < y_t^{(2)}$ for all $t > 0$.

**Lemma D.4.** *For all $t > 0$, $y_{t+1}^{(1)} < y_t^{(2)}$.*

*Proof.* Notice that:

$$\begin{cases} y_1^{(1)} = \eta \|\boldsymbol{x}_1\|^2 + \eta \langle \boldsymbol{x}_1, \boldsymbol{x}_2 \rangle \\ y_1^{(2)} = \eta \|\boldsymbol{x}_2\|^2 + \eta \langle \boldsymbol{x}_1, \boldsymbol{x}_2 \rangle. \end{cases}$$

1) $y_2^{(1)} < y_1^{(2)}$:

$$\begin{aligned} y_2^{(1)} &= y_1^{(1)} + \eta e^{-y_1^{(1)}} \|\boldsymbol{x}_1\|^2 + \eta e^{-y_1^{(2)}} \langle \boldsymbol{x}_1, \boldsymbol{x}_2 \rangle \\ &= \eta \left( e^{-y_1^{(1)}} + 1 \right) \|\boldsymbol{x}_1\|^2 + \eta \left( e^{-y_1^{(2)}} + 1 \right) \langle \boldsymbol{x}_1, \boldsymbol{x}_2 \rangle \\ &< \eta \times 2 \|\boldsymbol{x}_1\|^2 + \eta \times 2 \langle \boldsymbol{x}_1, \boldsymbol{x}_2 \rangle \\ &< \eta \langle \boldsymbol{x}_1, \boldsymbol{x}_2 \rangle + \eta \|\boldsymbol{x}_2\|^2 = y_1^{(2)}. \end{aligned}$$

2) Assume, for $t > 0$, $y_{t+1}^{(1)} < y_t^{(2)}$.

$$\begin{aligned} y_{t+2}^{(1)} &= y_{t+1}^{(1)} + \eta e^{-y_{t+1}^{(1)}} \|\boldsymbol{x}_1\|^2 + \eta e^{-y_{t+1}^{(2)}} \langle \boldsymbol{x}_1, \boldsymbol{x}_2 \rangle \\ &< y_t^{(2)} + \eta e^{-y_t^{(1)}} \|\boldsymbol{x}_1\|^2 + \eta e^{-y_t^{(2)}} \langle \boldsymbol{x}_1, \boldsymbol{x}_2 \rangle \\ &< y_t^{(2)} + \eta e^{-y_t^{(1)}} \langle \boldsymbol{x}_1, \boldsymbol{x}_2 \rangle + \eta e^{-y_t^{(2)}} \|\boldsymbol{x}_2\|^2 = y_{t+1}^{(2)}. \end{aligned}$$

$\qquad \square$

By Lemma D.4, for all $t > 0$, $\left( y_t^{(2)}, y_{t+1}^{(2)} \right)$ lies entirely on right-hand side of $\left( y_t^{(1)}, y_{t+1}^{(1)} \right)$, without any overlap.

We first analyze the following term: $\frac{y_{t+1}^{(1)} - y_t^{(1)}}{y_{t+1}^{(2)} - y_t^{(2)}}$. Observe that:

$$\begin{aligned} \frac{y_{t+1}^{(1)} - y_t^{(1)}}{y_{t+1}^{(2)} - y_t^{(2)}} &= \frac{\eta e^{-y_t^{(1)}} \|\boldsymbol{x}_1\|^2 + \eta e^{-y_t^{(2)}} \langle \boldsymbol{x}_1, \boldsymbol{x}_2 \rangle}{\eta e^{-y_t^{(2)}} \|\boldsymbol{x}_2\|^2 + \eta e^{-y_t^{(1)}} \langle \boldsymbol{x}_1, \boldsymbol{x}_2 \rangle} \\ &= \frac{\|\boldsymbol{x}_1\|^2 + e^{-\Delta y_t} \langle \boldsymbol{x}_1, \boldsymbol{x}_2 \rangle}{\langle \boldsymbol{x}_1, \boldsymbol{x}_2 \rangle + e^{-\Delta y_t} \|\boldsymbol{x}_2\|^2}. \end{aligned} \qquad (9)$$

It is derived that the fraction is an increasing sequence in terms of $t$. For values $a, b, c, c', d, d' > 0$, $\frac{a+c}{b+d} < \frac{a+c'}{b+d'} \Leftrightarrow ad' + cb + cd' < ad + c'b + c'd$. Taking:

$$\begin{cases} a = \|\boldsymbol{x}_1\|^2 \\ b = \langle \boldsymbol{x}_1, \boldsymbol{x}_2 \rangle \end{cases} \begin{cases} c = e^{-\Delta y_t} \langle \boldsymbol{x}_1, \boldsymbol{x}_2 \rangle \\ d = e^{-\Delta y_t} \|\boldsymbol{x}_2\|^2 \end{cases} \begin{cases} c' = e^{-\Delta y_{t+1}} \langle \boldsymbol{x}_1, \boldsymbol{x}_2 \rangle \\ d' = e^{-\Delta y_{t+1}} \|\boldsymbol{x}_2\|^2 \end{cases},$$

we have

$$
\begin{aligned}
& ad' + cb + cd' \\
&= e^{-\Delta y_{t+1}} \|\boldsymbol{x}_1\|^2 \|\boldsymbol{x}_2\|^2 + e^{-\Delta y_t} \langle \boldsymbol{x}_1, \boldsymbol{x}_2 \rangle^2 + e^{-\Delta y_t} e^{-\Delta y_{t+1}} \langle \boldsymbol{x}_1, \boldsymbol{x}_2 \rangle \|\boldsymbol{x}_2\|^2 \\
&< e^{-\Delta y_t} \|\boldsymbol{x}_1\|^2 \|\boldsymbol{x}_2\|^2 + e^{-\Delta y_{t+1}} \langle \boldsymbol{x}_1, \boldsymbol{x}_2 \rangle^2 + e^{-\Delta y_t} e^{-\Delta y_{t+1}} \langle \boldsymbol{x}_1, \boldsymbol{x}_2 \rangle \|\boldsymbol{x}_2\|^2 \\
&= ad + c'b + c'd.
\end{aligned}
$$

The inequality holds by Lemma D.3 and the Cauchy-Schwarz inequality. Taking the limit of Equation (9) as $t \to \infty$, the ratio converges to:

$$
R := \frac{\|\boldsymbol{x}_1\|^2}{\langle \boldsymbol{x}_1, \boldsymbol{x}_2 \rangle}. \tag{10}
$$

For the later uses, we also define the initial ratio, which is smaller than 1:

$$
R_0 := \frac{y_1^{(1)} - y_0^{(1)}}{y_1^{(2)} - y_0^{(2)}} = \frac{\|\boldsymbol{x}_1\|^2 + \langle \boldsymbol{x}_1, \boldsymbol{x}_2 \rangle}{\langle \boldsymbol{x}_1, \boldsymbol{x}_2 \rangle + \|\boldsymbol{x}_2\|^2} \ (\leq R). \tag{11}
$$

Now we analyze a similar ratio of the one-step difference, but in terms of $\sigma\left(y_t^{(i)}\right)$ instead of $y_t^{(i)}$. There, $\sigma$ stands for the logistic function, $\sigma(z) = (1 + e^{-z})^{-1}$. Notice that $\sigma'(z) = \sigma(z)(1 - \sigma(z))$.

**Lemma D.5.** $\gamma_V(t) := \dfrac{\sigma\left(y_{t+1}^{(1)}\right) - \sigma\left(y_t^{(1)}\right)}{\sigma\left(y_{t+1}^{(2)}\right) - \sigma\left(y_t^{(2)}\right)}$ *monotonically increases to* $+\infty$.

*Proof.*

$$
\begin{aligned}
\gamma_V(t) &= \frac{y_{t+1}^{(1)} - y_t^{(1)}}{y_{t+1}^{(2)} - y_t^{(2)}} \frac{\sigma'\left(\zeta_t^{(1)}\right)}{\sigma'\left(\zeta_t^{(2)}\right)} \quad \left(\text{for some } \begin{cases} \zeta_t^{(1)} \in \left(y_t^{(1)}, y_{t+1}^{(1)}\right) \\ \zeta_t^{(2)} \in \left(y_t^{(2)}, y_{t+1}^{(2)}\right) \end{cases} \text{by the mean value theorem.}\right) \\
&\geq \frac{y_{t+1}^{(1)} - y_t^{(1)}}{y_{t+1}^{(2)} - y_t^{(2)}} \frac{\sigma'\left(y_{t+1}^{(1)}\right)}{\sigma'\left(y_t^{(2)}\right)} \quad (\because \sigma' \text{: decreasing on } \mathbb{R}^+) \\
&= \frac{y_{t+1}^{(1)} - y_t^{(1)}}{y_{t+1}^{(2)} - y_t^{(2)}} \frac{e^{-y_{t+1}^{(1)}} \left(1 + e^{-y_{t+1}^{(1)}}\right)^{-2}}{e^{-y_t^{(2)}} \left(1 + e^{-y_t^{(2)}}\right)^{-2}} \\
&\geq \frac{y_{t+1}^{(1)} - y_t^{(1)}}{y_{t+1}^{(2)} - y_t^{(2)}} \frac{1}{4} e^{y_t^{(2)} - y_{t+1}^{(1)}} \quad \left(\because \left(1 + e^{-z}\right)^{-2} \in [1/4, 1] \text{ on } \mathbb{R}^+\right) \\
&\geq \frac{R_0}{4} e^{y_t^{(2)} - y_{t+1}^{(1)}}.
\end{aligned}
$$

As $y_t^{(2)} - y_{t+1}^{(1)} = y_t^{(2)} - y_t^{(1)} - \eta\left(e^{-y_t^{(1)}} \|\boldsymbol{x}_1\|^2 + e^{-y_t^{(2)}} \langle \boldsymbol{x}_1, \boldsymbol{x}_2 \rangle\right) \to \infty$, $\gamma_V(t) \to \infty$. For the part that proves $\gamma_V(t)$ is increasing, see Appendix D.1.1. $\qquad \square$

Notice that $\gamma_V(0) < 1$. Lemma D.5 implies that there exists (unique) $T_v > 0$ such that for all $t \geq T_v$, $\gamma_V(t) > 1$ holds, or $\sigma\left(y_{t+1}^{(1)}\right) - \sigma\left(y_t^{(1)}\right) > \sigma\left(y_{t+1}^{(2)}\right) - \sigma\left(y_t^{(2)}\right)$. Recall that the (sample) variance of a finite dataset $\mathcal{T} = \{\boldsymbol{x}_1, \cdots, \boldsymbol{x}_n\}$ can be computed as:

$$
\mathrm{Var}[\mathcal{T}] = \frac{1}{n(n-1)} \sum_{i=1}^{n-1} \sum_{j=i+1}^{n} (\boldsymbol{x}_i - \boldsymbol{x}_j)^2.
$$

Hence, for given $J$, (which corresponds to the window size,) for all $t \geq T_v$,

$$
S_{t;J}^{(1)} := \sqrt{\mathrm{Var}\left[\left\{\sigma\left(y_\tau^{(1)}\right)\right\}_{\tau=t}^{t+J-1}\right]} = \sqrt{\frac{1}{J(J-1)}\sum_{k=0}^{J-2}\sum_{l=k+1}^{J-1}\left[\sigma\left(y_{t+l}^{(1)}\right)-\sigma\left(y_{t+k}^{(1)}\right)\right]^2}
$$

$$
> \sqrt{\frac{1}{J(J-1)}\sum_{k=0}^{J-2}\sum_{l=k+1}^{J-1}\left[\sigma\left(y_{t+l}^{(2)}\right)-\sigma\left(y_{t+k}^{(2)}\right)\right]^2}
$$

$$
= \sqrt{\mathrm{Var}\left[\left\{\sigma\left(y_\tau^{(2)}\right)\right\}_{\tau=t}^{t+J-1}\right]} =: S_{t;J}^{(2)}.
$$

It is easily derived that the converse is true: If $\gamma_V(t)$ is increasing and $V_{t;J}^{(1)} > V_{t;J}^{(2)}$ then $\gamma_V(t) > 1$.

We have two metrics: the first is only the variance (which corresponds to the Dyn-Unc score) and the second is the variance multiplied by the mean subtracted from 1 (which corresponds to the DUAL pruning score). Both the variance and the mean are calculated within a window of fixed length. At the early epoch, as the model learns $x_2$ first, both metrics show a smaller value for $x_1$ than that for $x_2$. At the late epoch, now the model learns $x_1$, so the order of the metric values reverses for both metrics.

Our goal is to show that the elapsed time of the second metric for the order to be reversed is shorter than that of the first metric. Let $T_{vm}$ be that time for our metric. We represent the mean of the logistic output within a window of length $J$ and from epoch $t$, computed for $i$-th instance by $\mu_{t;J}^{(i)}$:

$$
\mu_{t;J}^{(i)} := \frac{1}{J}\sum_{\tau=t}^{t+J-1}\sigma\left(y_\tau^{(i)}\right). \tag{12}
$$

For $t \geq T_v$, we see that the inequality still holds:

$$
S_{t;J}^{(1)}\left(1-\mu_{t;J}^{(1)}\right)
$$

$$
= \sqrt{\frac{1}{J(J-1)}\sum_{k=0}^{J-2}\sum_{l=k+1}^{J-1}\left[\sigma\left(y_{t+l}^{(1)}\right)-\sigma\left(y_{t+k}^{(1)}\right)\right]^2}\left[1-\frac{1}{J}\sum_{\tau=t}^{t+J-1}\sigma\left(y_\tau^{(1)}\right)\right]
$$

$$
> \sqrt{\frac{1}{J(J-1)}\sum_{k=0}^{J-2}\sum_{l=k+1}^{J-1}\left[\sigma\left(y_{t+l}^{(2)}\right)-\sigma\left(y_{t+k}^{(2)}\right)\right]^2}\left[1-\frac{1}{J}\sum_{\tau=t}^{t+J-1}\sigma\left(y_\tau^{(2)}\right)\right]
$$

$$
= S_{t;J}^{(2)}\left(1-\mu_{t;J}^{(2)}\right).
$$

as for all $t$, $\sigma\left(y_t^{(2)}\right) > \sigma\left(y_t^{(1)}\right)$. Indeed, $T_{vm} \leq T_v$ holds, but is $T_{vm} < T_v$ true? To verify the question, we reshape the terms for a similar analysis upon $\mu$:

$$
S_{t;J}^{(1)}\left(1-\mu_{t;J}^{(1)}\right) \tag{13}
$$

$$
= \sqrt{\frac{1}{J(J-1)}\sum_{k=0}^{J-2}\sum_{l=k+1}^{J-1}\left[\sigma\left(y_{t+l}^{(1)}\right)-\sigma\left(y_{t+k}^{(1)}\right)\right]^2}\left[\frac{1}{J}\sum_{\tau=t}^{t+J-1}1-\sigma\left(y_\tau^{(1)}\right)\right] \tag{14}
$$

$$
> \sqrt{\frac{1}{J(J-1)}\sum_{k=0}^{J-2}\sum_{l=k+1}^{J-1}\left[\sigma\left(y_{t+l}^{(2)}\right)-\sigma\left(y_{t+k}^{(2)}\right)\right]^2}\left[\frac{1}{J}\sum_{\tau=t}^{t+J-1}1-\sigma\left(y_\tau^{(2)}\right)\right] \tag{15}
$$

$$
= S_{t;J}^{(2)}\left(1-\mu_{t;J}^{(2)}\right). \tag{16}
$$

The intuition is now clear: for any time before $T_v$, we know that the variance of $x_1$ is smaller than that of $x_2$, if the ratio corresponding to $1 - \sigma(y)$ is large, the factors could be canceled out and the inequality still holds. If this case is possible, definitely $T_{vm} < T_v$.

Now let us analyze the ratio of $1 - \sigma\left(y_t^{(i)}\right)$.

**Lemma D.6.** $\gamma_M(t) := \dfrac{1 - \sigma\left(y_t^{(1)}\right)}{1 - \sigma\left(y_t^{(2)}\right)}$ *increases to* $+\infty$.

*Proof.*

$$\gamma_M(t) = \frac{1 + e^{y_t^{(2)}}}{1 + e^{y_t^{(1)}}}$$

$$= e^{\Delta y_t} - \frac{e^{\Delta y_t} - 1}{1 + e^{y_t^{(1)}}}$$

$$\geq e^{\Delta y_t} - \frac{e^{\Delta y_t}}{1 + e^{y_t^{(1)}}}$$

$$= e^{\Delta y_t} \sigma\left(y_t^{(1)}\right).$$

The quantity in the last line indeed diverges to infinity. We now show that $\gamma_M(t)$ is increasing.

$$\gamma_M(t) = e^{\Delta y_t} - \frac{e^{\Delta y_t}}{1 + e^{y_t^{(1)}}} + \frac{1}{1 + e^{y_t^{(1)}}}$$

$$= e^{\Delta y_t} \sigma\left(y_t^{(1)}\right) + 1 - \sigma\left(y_t^{(1)}\right)$$

$$= \left(e^{\Delta y_t} - 1\right) \sigma\left(y_t^{(1)}\right) + 1$$

$$< \left(e^{\Delta y_{t+1}} - 1\right) \sigma\left(y_{t+1}^{(1)}\right) + 1 = \gamma_M(t+1).$$

$\square$

Notice that, for $a > c > 0, b > d > 0, \frac{a-c}{b-d} < \frac{a}{b} \Leftrightarrow \frac{a}{b} < \frac{c}{d}$. Recall from Lemma D.5 that $\gamma_V(t) = \frac{1 - \sigma\left(y_t^{(1)}\right) - \left[1 - \sigma\left(y_{t+1}^{(1)}\right)\right]}{1 - \sigma\left(y_t^{(2)}\right) - \left[1 - \sigma\left(y_{t+1}^{(2)}\right)\right]}$, hence $\gamma_V(t) < \gamma_M(t)$. Moreover,

$$\gamma_V(t) \leq \frac{y_{t+1}^{(1)} - y_t^{(1)}}{y_{t+1}^{(2)} - y_t^{(2)}} \frac{\sigma'\left(y_t^{(1)}\right)}{\sigma'\left(y_{t+1}^{(2)}\right)}$$

$$= \frac{y_{t+1}^{(1)} - y_t^{(1)}}{y_{t+1}^{(2)} - y_t^{(2)}} e^{y_{t+1}^{(2)} - y_t^{(1)}} \left(\frac{1 + e^{-y_{t+1}^{(2)}}}{1 + e^{-y_t^{(1)}}}\right)^2$$

$$\leq \frac{y_{t+1}^{(1)} - y_t^{(1)}}{y_{t+1}^{(2)} - y_t^{(2)}} e^{y_{t+1}^{(2)} - y_t^{(1)}} \left(\frac{1 + e^{-y_{t+1}^{(2)}}}{1 + e^{-y_t^{(1)}}}\right) \quad \because \left(\frac{1 + e^{-y_{t+1}^{(2)}}}{1 + e^{-y_t^{(1)}}}\right) \in (0, 1].$$

$$= \frac{y_{t+1}^{(1)} - y_t^{(1)}}{y_{t+1}^{(2)} - y_t^{(2)}} e^{y_{t+1}^{(2)} - y_t^{(2)}} e^{\Delta y_t} \left(\frac{1 + e^{-y_{t+1}^{(2)}}}{1 + e^{-y_t^{(1)}}}\right)$$

$$\leq \frac{y_{t+1}^{(1)} - y_t^{(1)}}{y_{t+1}^{(2)} - y_t^{(2)}} e^{y_{t+1}^{(2)} - y_t^{(2)}} e^{\Delta y_t} \left(\frac{1 + e^{-y_t^{(2)}}}{1 + e^{-y_t^{(1)}}}\right)$$

$$\leq R e^{y_1^{(2)} - y_0^{(2)}} \gamma_M(t).$$

Now we revisit Equation (13).

$$
\sqrt{\frac{1}{J(J-1)} \sum_{k=0}^{J-2} \sum_{l=k+1}^{J-1} \left[\sigma\left(y_{t+l}^{(1)}\right) - \sigma\left(y_{t+k}^{(1)}\right)\right]^2 \left[\frac{1}{J} \sum_{\tau=t}^{t+J-1} 1 - \sigma\left(y_\tau^{(1)}\right)\right]}
$$
$$
> \sqrt{\frac{1}{J(J-1)} \sum_{k=0}^{J-2} \sum_{l=k+1}^{J-1} \left[\sigma\left(y_{t+l}^{(2)}\right) - \sigma\left(y_{t+k}^{(2)}\right)\right]^2 \left[\frac{1}{J} \sum_{\tau=t}^{t+J-1} 1 - \sigma\left(y_\tau^{(2)}\right)\right]}
$$

(17)

Assume, for the moment, that for some constant $C > 1$, $\sigma\left(y_{t+1}^{(1)}\right) - \sigma\left(y_t^{(1)}\right) > C^{-1}\left[\sigma\left(y_{t+1}^{(2)}\right) - \sigma\left(y_t^{(2)}\right)\right]$ but $1 - \sigma\left(y_t^{(1)}\right) > C^2\left[1 - \sigma\left(y_t^{(2)}\right)\right]$ for all large $t$. Then the ratio of the first term of the left-hand side of Equation (17) to the first term of the right-hand side is greater than $C^{-2}$. Also, the ratio of the second term of the left-hand side of Equation (17) to the second term of the right-hand side is greater than $C^2$. If so, we observe that 1) the inequality in Equation (17) holds, 2) as the condition $\gamma_V(t) \geq 1$ for $T_v$ now changed to $\gamma_V(t) \geq C^{-1}$ for $T_{vm}$, hence $T_{vm} < T_v$ is guaranteed. It remains to find the constant $C$. Recall that, for all $t$,

$$
\gamma_V(t) \leq Re^{y_1^{(2)} - y_0^{(2)}} \gamma_M(t).
$$

If we set $Re^{y_1^{(2)} - y_0^{(2)}} = C^{-3}$, when $\gamma_V(t)$ becomes at least $C^{-1}$, we have $\gamma_M(t) \geq C^2$, satisfying the condition for $T_{vm}$. If the learning rate is sufficiently small, then $\gamma_V(t)$ cannot significantly increase in one step, allowing $\gamma_V(t)$ to fall between $C^{-1}$ and 1. Refer to Figure 23a to observe that the graph of $\gamma_V(t)$ resembles that of a continuously increasing function.

### D.1.1. MONOTONICITY OF $\gamma_V(t)$

Recall that:

$$
\gamma_V(t) := \frac{\sigma\left(y_{t+1}^{(1)}\right) - \sigma\left(y_t^{(1)}\right)}{\sigma\left(y_{t+1}^{(2)}\right) - \sigma\left(y_t^{(2)}\right)}
$$
$$
= \frac{y_{t+1}^{(1)} - y_t^{(1)}}{y_{t+1}^{(2)} - y_t^{(2)}} \frac{\sigma'\left(\zeta_t^{(1)}\right)}{\sigma'\left(\zeta_t^{(2)}\right)}
$$

for some $\zeta_t^{(1)} \in \left(y_t^{(1)}, y_{t+1}^{(1)}\right), \zeta_t^{(2)} \in \left(y_t^{(2)}, y_{t+1}^{(2)}\right)$ by the mean value theorem. The first term is shown to be increasing (to $R$). $\gamma_V(t)$ is increasing if the second term is also increasing in $t$.

Let $\Delta\zeta_t := \zeta_t^{(2)} - \zeta_t^{(1)}$. By Lemma D.4, $\Delta\zeta_t > 0$.

$$
\frac{\sigma'\left(\zeta_t^{(1)}\right)}{\sigma'\left(\zeta_t^{(2)}\right)} = \frac{e^{-\zeta_t^{(1)}}}{e^{-\zeta_t^{(2)}}} \left(\frac{1 + e^{-\zeta_t^{(2)}}}{1 + e^{-\zeta_t^{(1)}}}\right)^2
$$
$$
= e^{\Delta\zeta_t}\left(\frac{1 + e^{-\zeta_t^{(1)} - \Delta\zeta_t}}{1 + e^{-\zeta_t^{(1)}}}\right)^2.
$$

Define $g(x, y) := e^x\left(\frac{1 + e^{-y-x}}{1 + e^{-y}}\right)^2$. The partial derivatives satisfy:

$$
\begin{cases}
\nabla_x g = \frac{(e^y - e^{-x})(e^{x+y} + 1)}{(1+e^y)^2} > 0 \text{ for } x > 0 \text{ if } y > 0 \\
\nabla_y g = \frac{2e^{y-x}(e^x - 1)(e^{x+y} + 1)}{(1+e^y)^3} > 0, \forall y \text{ if } x > 0.
\end{cases}
$$

Notice that $\frac{\sigma'\left(\zeta_t^{(1)}\right)}{\sigma'\left(\zeta_t^{(2)}\right)} = g(\Delta\zeta_t, \zeta_t^{(1)})$. Since $\zeta_t^{(1)} \in \left(y_t^{(1)}, y_{t+1}^{(1)}\right)$ is (strictly) increasing and positive, if we show that $\Delta\zeta_t$ is increasing in $t$, we are done. Our result is that, if $y_{t+1}^{(i)} - y_t^{(i)}$ is small for $i = 1, 2$, $\zeta_t^{(i)} \approx \left(y_t^{(i)} + y_{t+1}^{(i)}\right)/2$ so $\Delta\zeta_t \approx (\Delta y_t + \Delta y_{t+1})/2$, which is indeed increasing.

In particular, if (, assume for now) for all $t$,

$$\zeta_t^{(i)} \in \left( \frac{2y_t^{(i)} + y_{t+1}^{(i)}}{3}, \frac{y_t^{(i)} + 2y_{t+1}^{(i)}}{3} \right) \tag{18}$$

$$\Rightarrow \Delta\zeta_t \in \left( \frac{\Delta y_t + \Delta y_{t+1}}{3} + \frac{y_t^{(2)} - y_{t+1}^{(1)}}{3}, \frac{\Delta y_t + \Delta y_{t+1}}{3} + \frac{y_{t+1}^{(2)} - y_t^{(1)}}{3} \right)$$

$$\Rightarrow \Delta\zeta_t < \frac{\Delta y_t + \Delta y_{t+1}}{3} + \frac{y_{t+1}^{(2)} - y_t^{(1)}}{3}$$

$$< \frac{\Delta y_{t+1} + \Delta y_{t+2}}{3} + \frac{y_{t+1}^{(2)} - y_{t+2}^{(1)}}{3} \tag{\dagger}$$

$$< \Delta\zeta_{t+1}$$

($\dagger$) holds by Assumption D.1:

$$(\dagger) \Leftrightarrow \Delta y_{t+2} - \Delta y_t > y_{t+2}^{(1)} - y_t^{(1)}, \forall t$$

$$\Leftarrow \Delta y_{t+1} - \Delta y_t > y_{t+1}^{(1)} - y_t^{(1)}, \forall t$$

$$\Leftrightarrow \eta \left[ e^{-y_t^{(1)}} \left( \langle \boldsymbol{x}_1, \boldsymbol{x}_2 \rangle - \|\boldsymbol{x}_1\|^2 \right) + e^{-y_t^{(2)}} \left( \|\boldsymbol{x}_2\|^2 - \langle \boldsymbol{x}_1, \boldsymbol{x}_2 \rangle \right) \right] >$$

$$\eta \left[ e^{-y_t^{(1)}} \|\boldsymbol{x}_1\|^2 + e^{-y_t^{(2)}} \langle \boldsymbol{x}_1, \boldsymbol{x}_2 \rangle \right], \forall t.$$

It remains to show Equation (18). To this end, we use Lemma D.7.

**Lemma D.7.** *Let $z_2 > z_1 (\geq 0)$ be reals and $\zeta \in (z_1, z_2)$ be a number that satisfies the following: $\sigma(z_2) - \sigma(z_1) = (z_2 - z_1)\sigma'(\zeta)$. Denote the midpoint of $(z_1, z_2)$ as $m := (z_1 + z_2)/2$. For $(1 \gg)\epsilon > 0$, if $z_2 - z_1 < \mathcal{O}(\sqrt{\epsilon})$ then $|\zeta - m| < \epsilon$.*

*Proof.* Expand the Taylor series of $\sigma$ at $m$ for $z_i$:

$$\sigma(z_i) = \sigma(m) + \sigma'(m)(z_i - m) + \frac{1}{2!}\sigma''(m)(z_i - m)^2 + \frac{1}{3!}\sigma'''(m)(z_i - m)^3 + \mathcal{O}\left(|z_i - m|^4\right)$$

We have:

$$\sigma(z_2) - \sigma(z_1) = \sigma'(m)(z_2 - z_1) + \frac{1}{24}\sigma'''(m)(z_2 - z_1)^3 + \mathcal{O}\left((z_2 - z_1)^5\right)$$

$$\sigma'(\zeta) = \sigma'(m) + \frac{1}{24}\sigma'''(m)(z_2 - z_1)^2 + \mathcal{O}\left((z_2 - z_1)^4\right)$$

Now, expand the Taylor series of $\sigma'$ at $m$ for $\zeta$:

$$\sigma'(\zeta) = \sigma'(m) + \sigma''(m)(\zeta - m) + \frac{1}{2!}\sigma'''(m)(\zeta - m)^2 + \mathcal{O}\left(|\zeta - m|^3\right)$$

Comparing the above two lines gives

$$24\sigma''(m)(\zeta - m) + 12\sigma'''(m)(\zeta - m)^2 = \sigma'''(m)(z_2 - z_1)^2 + \mathcal{O}\left((z_2 - z_1)^3\right)$$

If $\sigma'''(m) = 0$ then $|\zeta - m| = \mathcal{O}\left((z_2 - z_1)^3\right)$, so $z_2 - z_1 = \mathcal{O}(\sqrt{\epsilon})$ is sufficient.

Otherwise, we can solve the above for $\zeta - m$ from the fact that $\sigma''(z) < 0$ for $z > 0$:

$$12\sigma'''(m)(\zeta - m) = -12\sigma''(m) - \sqrt{(12\sigma''(m))^2 + 12\sigma'''(m)\left[\sigma'''(m)(z_2 - z_1)^2 + \mathcal{O}\left((z_2 - z_1)^3\right)\right]}$$

$$= \frac{12\sigma'''(m)^2(z_2 - z_1)^2}{24\sigma''(m)} + \mathcal{O}\left((z_2 - z_1)^3\right)$$

The last equality is from the Taylor series $\sqrt{1 + \frac{a}{x^2}} - 1 = \frac{a}{2x^2} + \mathcal{O}\left(a^2 x^{-4}\right)$, or $\sqrt{x^2 + a} - x = \frac{a}{2x} + \mathcal{O}\left(a^2 x^{-3}\right)$. We have $|\zeta - m| = \Theta\left((z_2 - z_1)^2\right)$. $\qquad\qquad\qquad\qquad\qquad\qquad\qquad\qquad\qquad\qquad\qquad\qquad\qquad\qquad\square$

For $|\zeta_t^{(i)} - \left(y_t^{(i)} + y_{t+1}^{(i)}\right)/2| < \left(y_{t+1}^{(i)} - y_t^{(i)}\right)/6$, it suffices to have $y_{t+1}^{(i)} - y_t^{(i)} < \mathcal{O}\left(\sqrt{\left(y_{t+1}^{(i)} - y_t^{(i)}\right)/6}\right)$. This generally holds for sufficiently small $\eta$.

## D.2. Experimental Results under Synthetic Setting

This section displays the figures plotted from the experiments on the synthetic dataset. We choose $\mathcal{X} = \mathbb{R}^2$ and $\mathcal{D} = \{((0.1, 0.1), 1), ((10, 5), 1)\}$. We fix $J = 10$ and $\eta = 0.01$ (unless specified). The total time of training $T$ is specified for each figure for neat visualization.

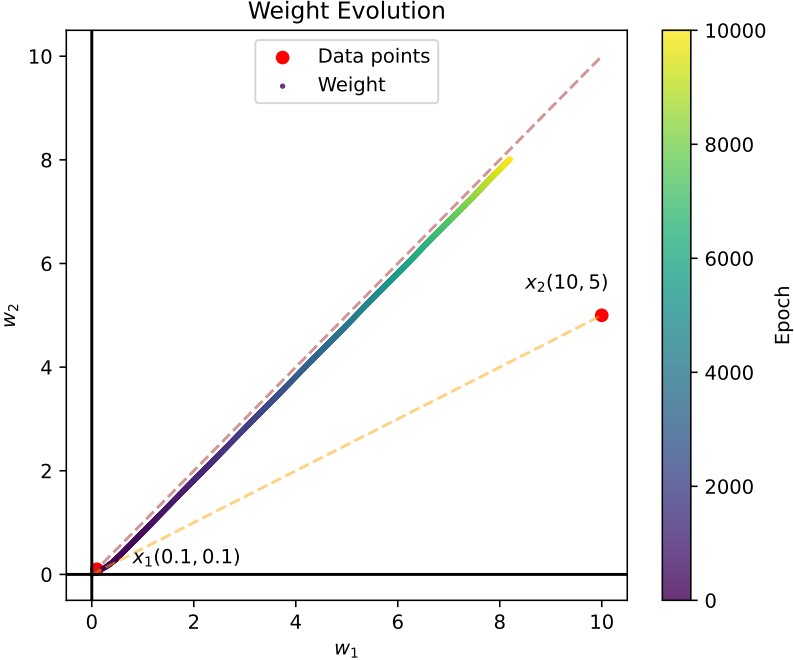

*Figure 22.* Illustration of the evolution of the weight as the model learns from the two-point dataset. Observe that the weight learns $\boldsymbol{x}_2$ first (closer to the orange dashed line), but gradually moves towards $\boldsymbol{x}_1$ (closer to the brown dashed line). Here $T = 10,000$.

We also empirically validate our statements of Appendix D.1.1. Figure 23 shows that $\gamma_V(t)$ and $\Delta\zeta_t$ are indeed increasing functions. Figure 24 shows that $\zeta_t^{(i)}$ is sufficiently close to the midpoint of the interval it lies in, $\left(y_t^{(i)}, y_{t+1}^{(i)}\right)$.

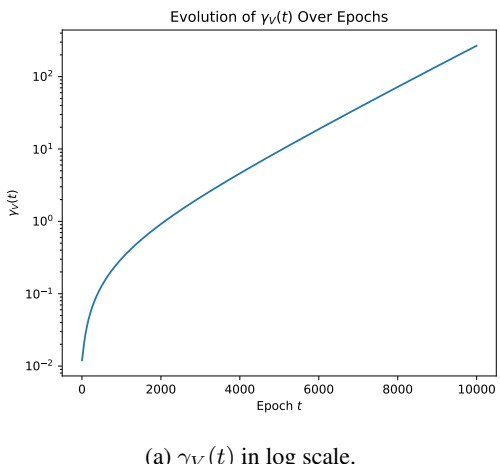 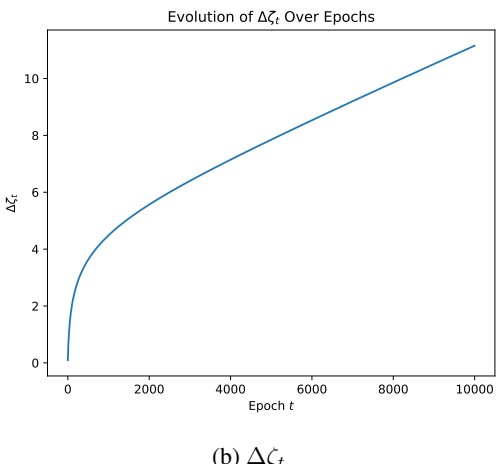

(a) $\gamma_V(t)$ in log scale.

(b) $\Delta\zeta_t$

*Figure 23.* Empirical validations of the critical statements in Appendix D.1.1. We ran experiments and plot the results that both $\gamma_V(t)$ (left—in log scale) and $\Delta\zeta_t$ (right) are an increasing sequence in terms of $t$. Here, we set $\eta = 0.0005$. The reason is that if the learning rate is larger, $\sigma(y_t^{(2)})$ quickly saturates to 1, leading to a possibility of division by zero in $\gamma_V(t)$ and degradation in numerical stability of $\Delta\zeta_t$. Moreover, notice that the graph of $\gamma_V(t)$ in the log scale closely resembles that of $\Delta\zeta_t$ in the original scale.

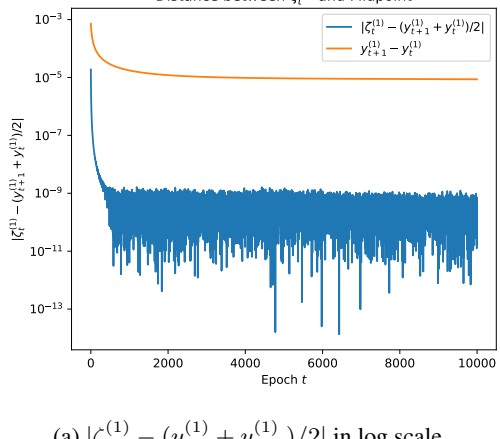 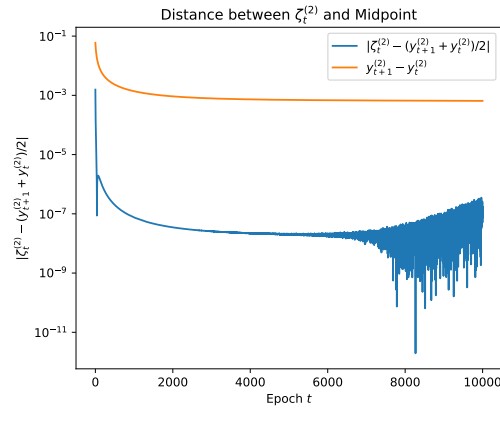

(a) $|\zeta_t^{(1)} - (y_t^{(1)} + y_{t+1}^{(1)})/2|$ in log scale.

(b) $|\zeta_t^{(2)} - (y_t^{(2)} + y_{t+1}^{(2)})/2|$ in log scale.

*Figure 24.* Empirical validations of the critical statements in Appendix D.1.1. We ran experiments and plot the results that both $\zeta_t^{(1)}$ (left) and $\zeta_t^{(2)}$ (right) are extremely close to the midpoint $(y_t^{(1)} + y_{t+1}^{(1)})/2$ and $(y_t^{(2)} + y_{t+1}^{(2)})/2$, compared to the interval length, respectively. In both plots, the blue line is the true distance while the orange line is the interval length. Here, we set $\eta = 0.0005$ for the same reasoning of Figure 23. Empirically, the noise introduced by MVT is too small to deny that $\Delta\zeta_t$ is an increasing sequence.

We also show that we can observe the "flow" of the moon plot as in Figure 2 for the synthetic dataset.

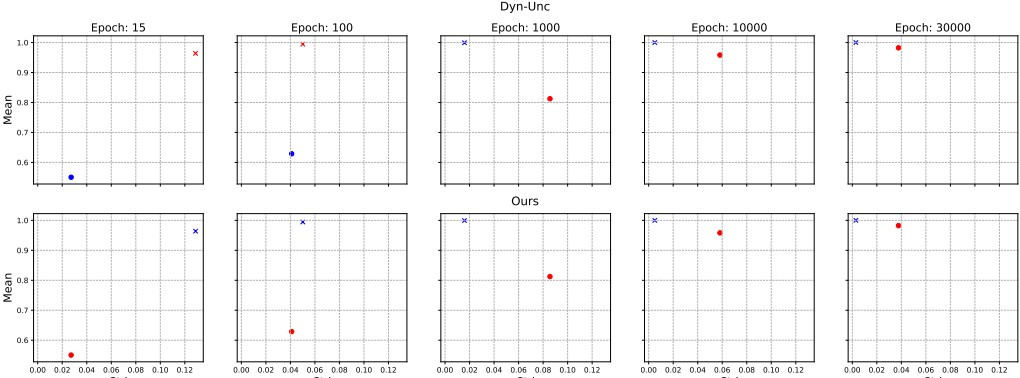

*Figure 25.* Evolution of $x_1, x_2$ by their mean and standard deviation in prediction probabilities at different epochs. The marker 'o' and 'x' stands for $x_1$ and $x_2$, respectively. The red color indicates the sample to be selected, and the blue color indicates the sample to be pruned. Observe that the path that each data point draws resembles is of moon-shape. Here $T = 30,000$.

