# OpenReview forum: "Lightweight Dataset Pruning without Full Training via Example Difficulty and Prediction Uncertainty"
_ICML.cc/2025/Conference — ICML 2025 poster_

### Official Review · Reviewer_rkgP · 2025-02-14

**Overall Recommendation:** 4

**Summary:**

This paper aims to pruning datasets in the early stages of training, without the need to train on the entire dataset. To achieve this, the authors propose a new scoring metric - the DUAL Score, which simultaneously considers sample difficulty and prediction uncertainty. To address potential sample distribution bias under extreme pruning rates, they introduce a Beta distribution-based adaptive sampling method for pruning ratios, ensuring that the selected subset better represents the overall data distribution.

# **update after rebuttal**
I would like to thank the authors for their sincere efforts to address my concerns. I have increase my score.

**Claims And Evidence:**

In general, I agree with the authors' proposed motivation that "Many existing methods require training a model with a full dataset over a large number of epochs before being able to prune the dataset, which ironically makes the pruning process more expensive than just training the model on the entire dataset."

The majority of the claims made in the submission are supported by clear and convincing evidence. The authors provide extensive empirical results across multiple benchmark datasets—CIFAR-10, CIFAR-100, and ImageNet-1K—as well as under challenging conditions like label noise, image corruption, and cross-architecture scenarios. Moreover, the theoretical analysis (although it appears somewhat supplementary, which I don't consider problematic) reinforces the claim that the DUAL Score can identify critical samples earlier than methods based solely on prediction uncertainty.
However, while the evidence is largely compelling, a few claims could benefit from additional validation. I have provided my further recommendations in the following review.

**Essential References Not Discussed:**

See Other Strengths And Weaknesses.

**Experimental Designs Or Analyses:**

Generally speaking, the authors have done a good job. However, it's worth noting that I believe they should provide a graph plotting test accuracy against method runtime (with wall-time on the x-axis), to clearly demonstrate the performance differences between various methods in terms of wall-time, thereby locating the contribution of the DUAL method.

In my view, this setting provides the most intuitive representation of each method's capability to reduce computational overhead during pruning. When implementing this setting, the authors should also include [1] InfoBatch and [2] IES. The authors can compare the methods under both high and low pruning rate scenarios, selecting representative methods from Table 1 for this experiment.

[1] InfoBatch: Lossless Training Speed Up by Unbiased Dynamic Data Pruning, ICLR 2024.

[2] Instance-dependent Early Stopping, ICLR 2025.

**Methods And Evaluation Criteria:**

The DUAL Score combines example difficulty with prediction uncertainty, allowing for early identification of influential samples without requiring a full training cycle. Additionally, the adaptive Beta Sampling strategy is designed to dynamically balance the selection of difficult, informative samples and easier ones, ensuring that the pruned subset remains representative of the overall data distribution even at high pruning ratios.

The evaluation criteria employed in the study are both robust and appropriate for the application. By leveraging standard benchmark datasets such as CIFAR-10, CIFAR-100, and ImageNet-1K, the authors provide a comprehensive assessment of their approach across a range of complexities and scales. The inclusion of realistic scenarios—such as experiments under label noise, image corruption, and cross-architecture generalization—further underscores the practical relevance and robustness of the method.

**Other Comments Or Suggestions:**

See Other Strengths And Weaknesses

**Other Strengths And Weaknesses:**

My primary concern is that the authors only discuss recent work on static dataset pruning. Despite some differences in settings, I strongly recommend comparing the proposed method with [1] InfoBatch (ICLR'24 Oral) and [2] Instance-dependent Early Stopping (IES) (ICLR'25 Spotlight). These are important works in dataset pruning that shouldn't be excluded from the baselines. Both works have straightforward code implementations and methodologies, making fair comparisons feasible. Given the rebuttal timeline, I believe it would be sufficient to include results on the two CIFAR datasets in Table 1.

In the field of dataset pruning, my general perspective is that for high pruning rates, pre-training pruning (similar to coreset) is preferable, while for low pruning rates, Dynamic Data Pruning is the better choice. If the authors' proposed method underperforms compared to the aforementioned methods at low pruning rates, that's acceptable - I suspect DUAL would achieve superior performance at high (or extremely high) pruning rates. The authors can confidently note that these Dynamic Data Pruning methods become ineffective at high removal rates, denoting such cases with "-" in their results.

I commend the paper's clear organization and the authors' substantial effort on evaluation. In particular, I appreciate their focus on addressing an important challenge in the field: the computational inefficiency of many existing coreset selection methods. This is indeed a significant problem that warrants attention.

[1] InfoBatch: Lossless Training Speed Up by Unbiased Dynamic Data Pruning, ICLR 2024.

[2] Instance-dependent Early Stopping, ICLR 2025.

**Questions For Authors:**

Not applicable.

**Relation To Broader Scientific Literature:**

Many existing methods require training a model with a full dataset over a large number of epochs before being able to prune the dataset, which ironically makes the pruning process more expensive than just training the model on the entire dataset. Key research includes Data Pruning/Core Set studies by:

Huggins et al. (2016) Coresets for scalable...

Paul et al. (2021) Finding important examples early in training

Krishnateja et al. (2021) ``Glister''

Xia et al. (2022, 2024) on moderate and refined core set approaches

**Theoretical Claims:**

This paper is empirical research and does not make significant theoretical claims. I have reviewed what could be considered the theoretical component of the paper - the Theorem for DUAL Pruning in Appendix D. This section does not present any novel theoretical contributions; rather, the authors combine two previously explored aspects - Difficulty and Uncertainty - to facilitate efficient and effective selection.
Furthermore, I have reservations about the Beta Sampling method proposed by the authors, as it seems to introduce an excessive number of hyperparameters that could be difficult to control (although the authors can demonstrate these hyperparameters are not sensitive).

However, I want to emphasize that the lacking of theoretical analysis is NOT a shortcoming. I do not look the paper require novel theoretical contributions or theoretical guarantees.

---

> ### Author Rebuttal · Authors · 2025-04-01
>
> We sincerely thank the reviewer for their valuable and constructive feedback. Below, we address the concerns raised.
>
> # Comparison with Dynamic Data Pruning Methods
>
> Thank you for pointing out the relevant references, especially [1, 2]. To address your concerns, we conducted several experiments comparing recent dynamic pruning methods with static approaches, including DUAL pruning.
>
> Before discussing the experimental results, let us first highlight two major differences between static data pruning and dynamic data pruning.
>
> 1. Compared to static pruning, dynamic pruning maintains access to the entire original dataset throughout training, **allowing it to fully leverage all available information in the original dataset**.
> 2. While both aim to improve training efficiency, their underlying goals differ slightly. Static data pruning seeks to identify a “fixed” subset that reduces the dataset size while preserving as much information about the original dataset as possible. This subset can then serve as a new, independent dataset, reusable across various model architectures and experimental setups. In contrast, dynamic data pruning enhances training efficiency within a **single training session** by pruning data dynamically on the fly. However, this approach requires storing the entire original dataset, making dynamic pruning less memory-efficient and not reusable.
>
> ## Standard Training
>
> We conducted experiments on CIFAR-10, 100 with ResNet-18, following the same hyperparameters as in Section 4.1 of our paper. All reported results are averaged over five runs. We first tested dynamic random pruning, which dynamically prunes randomly selected samples from the entire dataset at each epoch. Notably, dynamic random pruning significantly outperformed all static baselines, achieving test accuracies of 91.82% on CIFAR-10 and 72.8% on CIFAR-100 at a pruning ratio of 90%. We also evaluated [1, 2] and the results are provided [here](https://vo.la/XAAkQb). Overall, dynamic methods consistently outperform static baselines. However, at lower pruning ratios (e.g., on CIFAR-10), DUAL can outperform dynamic methods under a similar computational budget.
>
> As mentioned, we believe this performance gap stems from **differences in accessible information**: static methods are limited to 10% of the data, while dynamic methods use the full dataset. Consequently, the performance gap widens even further at aggressive pruning ratios, contrary to the reviewer’s expectations. To validate this, we plotted how often each sample was seen during training. The plot [here](https://vo.la/jhIWIZ) shows that static methods are confined to a subset, while dynamic ones use nearly all data—rendering direct comparison somewhat unfair. Indeed, dynamic pruning methods might be better compared with scheduled batch-selection approaches, such as curriculum learning, rather than static pruning methods.
>
> ## Label Noise Setting
>
> We also evaluated these methods under label noise. In fact, [2] concluded that **their method cannot prune any samples (corrupted or not) when label noise is introduced**. Similarly, [1] tends to retain harder (and often noisy) samples, as it removes only easy examples during training. In contrast, DUAL effectively filters noisy samples, improving performance even beyond full-data training.
>
> We conducted experiments on CIFAR-100 with a 40% label noise setting (full-train test accuracy: 52.74%) to verify this explanation. DUAL achieves **over 70% test accuracy** at a 50% pruning ratio, whereas InfoBatch achieves only 51.24% accuracy with a similar number of iterations. Under similar iterations, random dynamic pruning achieves 51.81% test accuracy, which still outperforms random static pruning (see Table 7 in Appendix). Lastly, IES [2] prunes only 1.7% of samples during training (consistent with the original report in their paper), resulting in 51.95% test accuracy. Furthermore, our static method can create fixed subsets in which **nearly all noisy samples have been removed, resulting in high-quality datasets that can be preserved for future use.**
>
> # Minor Comments
>
> - For details on the Beta sampling method and its hyperparameters, please refer to the response to Reviewer YpaU.
> - As suggested, we plotted test accuracy against total training time. The plots are available [here](https://vo.la/OeGesx). Results show that DUAL efficiently prunes data while achieving SOTA performance.
>
> We will include these findings and additional discussions in the revised version.
>
> ---
>
> [1] InfoBatch: Lossless Training Speed Up by Unbiased Dynamic Data Pruning, ICLR 2024.
>
> [2] Instance-dependent Early Stopping, ICLR 2025.

---

> > ### Comment · Reviewer_rkgP · 2025-04-02
> >
> > I would like to thank the authors for their sincere efforts to address my concerns.
> > I am inclined to increase my score by +1 (as a result, 4).
> > I would like to seeing all the revisions in the updated version.

---

> > > ### Author Response · Authors · 2025-04-02
> > >
> > > Thank you for your response. We are glad to hear that our explanations were helpful and adequately addressed your concerns. We also appreciate your suggestions for improvement and plan to incorporate them into our next revision.
> > >
> > > Best regards,
> > >
> > > Authors

---

### Official Review · Reviewer_TH5d · 2025-03-13

**Overall Recommendation:** 2

**Summary:**

This paper presents a dataset pruning method designed to reduce the computational burden of the pruning process. The authors introduce a strategy that leverages both difficulty and prediction uncertainty to efficiently select a coreset at an early stage of training. The effectiveness of the approach is validated across multiple benchmark datasets. However, the novelty of the method is somewhat limited, and the comparative experiments are not entirely comprehensive.

**Claims And Evidence:**

The main claim in Section 3.2 is unclear and confusing.

The authors attempt to compare the differences in the “Dyn-Unc score” at epochs 60 and 90. However, inconsistencies between the subcaptions of Figure 2, the main caption, and the color bars make it difficult to interpret their intended meaning.

Additionally, the authors do not specify how the mean and standard deviation are calculated—whether these values are computed at epoch 60 or 90 remains unclear.

Moreover, some claims in this section appear to be incorrect. Specifically, the authors argue that to use the Dyn-Unc score at epoch 60 for selecting samples with high Dyn-Unc scores at epoch 90, they should target the samples in the bottom-right region. However, according to Figure 2, the correct region should be the **top-right**, not the bottom-right.

**Essential References Not Discussed:**

No

**Experimental Designs Or Analyses:**

As mentioned earlier, the results regarding computational load are incomplete. The paper does not provide a thorough comparison of training costs with other pruning methods, especially on large-scale datasets like ImageNet. Without a direct comparison to all baseline methods in terms of computational efficiency, the claim that the proposed approach reduces pruning costs remains unsubstantiated.

**Methods And Evaluation Criteria:**

The proposed method does not actually reduce the computational cost of pruning, as it still requires training the original ImageNet for 60 epochs—equivalent to the full training process of (SOTA) methods like TDDS and CCS. This limitation likely explains why the authors only provide comparative results (e.g., Figure 4) on CIFAR rather than ImageNet.

Furthermore, in Figure 4, the authors do not compare their method with all relevant baselines. The results appear to be selectively presented, raising concerns about the completeness of the comparison.

**Other Comments Or Suggestions:**

There are some typos and misleading annotation in figures.

For example, in **Figure 2**, the caption and description are inconsistent, making it difficult to understand the intended comparison of the **Dyn-Unc score** at **Epoch 60 and 90**.

Additionally, in **Figure 4**, the **y-axis label** “total subset training time” is misleading and should be corrected to **“total time spent”**.

**Other Strengths And Weaknesses:**

The novelty of this work is limited, as it primarily offers a small improvement over the existing Dyn-Unc method. Additionally, the effectiveness in reducing computational load is not significant, as the proposed approach still requires extensive training (e.g., 60 epochs on ImageNet), which is comparable to the full training process of existing SOTA pruning methods like TDDS and CCS.

**Questions For Authors:**

Please refer to the comments above.

**Relation To Broader Scientific Literature:**

NA

**Theoretical Claims:**

yes.

---

> ### Author Rebuttal · Authors · 2025-04-01
>
> We appreciate your time and insightful comments. Below, we address the concerns and clarify any confusion raised.
>
> # 1. Figures 2 & 3
>
> First, we apologize for any confusion in Figures 2 and 3. Revised figures are available [here](https://vo.la/msMJZE) (see Figure2_revised, Figure3_revised). In both figures:
>
> - The Y-axis is the mean at epoch $T$: $mean_T(\mathbf{x},y)\coloneqq\bar{\mathbb{P}}=\frac{1}{T}\sum_{t=1}^T\mathbb{P}_t(y\mid\mathbf{x})$
> - The X-axis is standard deviation at epoch $T$: $std_T(\mathbf{x},y)\coloneqq\sqrt{\frac{\sum_{t=1}^T[\mathbb{P}_t(y\mid\mathbf{x})-\bar{\mathbb{P}}]^2}{T-1}}$.
>
> In Figure 2, we mislabeled the colorbars. Both the labels are revised as ‘Dyn-Unc Score’.
>
> Figure 2a shows that if we calculate the Dyn-Unc score at epoch 60, the data points with the highest scores tend to evolve toward the **top-right region** by the end of training (epoch 90). However, this is not desirable for Dyn-Unc, as it prioritizes uncertain points (rightmost region) **at the end of training** (Figure 2b right). To effectively target that region **earlier** in training (e.g., at epoch 60), we should instead focus on the **bottom-right region**, since those points eventually move to the rightmost area by epoch 90.
>
> Hence, we modify the Dyn-Unc score by multiplying it with 1-prediction mean, which helps us better sample this target region at an earlier stage, as shown in Figure 3. Also, the bold outline and the colorbar label of Figure 3 should be corrected to indicate **Epoch 60**.
>
> Again, we sincerely apologize for the confusion this may have caused.
>
> # 2. Computational Efficiency in ImageNet
>
> First, we would like to clarify that CCS and TDDS require full training (90 epochs) to compute their scores on ImageNet. This is explicitly mentioned in Section 5.2 of TDDS [1] and in Appendix B of CCS [2]. The 60 epochs refer to the “post-pruning” training used for coreset, as described in [1]. For a fair comparison, we train 300,000 iterations on coresets, following the setup of CCS.
>
> For further validation, we implemented CCS with AUM score calculated at epoch 60. As shown in the table below, the test accuracy is significantly lower than the original performance. Moreover, at low pruning ratios, the performance is even worse than random pruning (see Table 2 in our paper).
>
> | Pruning Ratio | 30% | 50% | 70% | 80% | 90% |
> | --- | --- | --- | --- | --- | --- |
> | CCS (ep 60) | 70.97 | 69.88 | 66.27 | 63.50 | 56.79 |
>
> Therefore we can assert that our method is both time-efficient and superior compared to CCS. Also, we want to emphasize that saving 30 epochs of full ImageNet dataset training is a significant improvement. Training for 90 epochs on NVIDIA A6000 takes approximately 20h, **thus reduction of 30 epochs can save about 6-7h.**
>
> There are a few subtle points for TDDS. In TDDS, **the epochs used for full dataset training is 90, then they perform an exhaustive search** to determine that around 30 epochs is optimal for pruning ImageNet by 70-90%. We evaluated their method across all pruning ratios using the reported score computation epoch. As a result, TDDS has a shorter overall training time when we ignore the exhaustive search; however, the important thing is that both their reported results and our reproduced experiments show that **our method significantly outperforms TDDS** in test accuracy.
>
> # 3. Regarding Figure 4.
>
> We can explicitly compare other baselines, such as CCS and TDDS, with DUAL in terms of total time spent as shown in Figure 4. However, many of the baseline plots are omitted because these methods require full training on the original dataset, which leads to **excessive overlap in the curves**. We will clarify this point more explicitly in the revision. We also plotted the test accuracy against the total time [here](https://vo.la/OeGesx), which hopefully addresses your concerns.
>
> While we could also create a version of Figure 4 using ImageNet to compare our effectiveness with other SOTA methods such as CCS and D2, we present the CIFAR results in Figure 4 as it highlights the improvement achieved by our method.
>
> # 4. Novelty of DUAL
>
> The key novelty of DUAL is its time efficiency. Unlike most pruning methods that require full training to estimate sample difficulty, DUAL identifies informative samples much earlier. To address performance drops at high pruning ratios, we use Beta sampling to include easier samples. While the goal aligns with CCS and BOSS, our approach is distinct—it leverages prediction mean and a non-linear shift in the distribution mode. This allows us to prioritize harder samples at low or medium pruning ratios and easier ones at high ratios, contributing to our SOTA performance.
>
> # 5. Minor comments
>
> Thank you for pointing out the typo. We will fix them in the new version.
>
> ---
>
> [1] Spanning training progress: Temporal dual-depth scoring (tdds) for enhanced dataset pruning, CVPR 2024.
>
> [2] Coverage-centric coreset selection for high pruning rates, ICLR 2023.

---

### Official Review · Reviewer_YpaU · 2025-03-14

**Overall Recommendation:** 4

**Summary:**

- This paper introduces a new method Difficulty and Uncertainty Aware Lightweight (DUAL) that combines Dyn-Unc with a measure of prediction confidence over training.
- The authors further introduce pruning-ratio-adaptive Beta sampling, which boosts performance at all pruning ratios and particularly helps at very high pruning rates
- The authors find that DUAL+Beta outperform existing pruning methods particularly at high pruning rates.

**Claims And Evidence:**

Yes

**Essential References Not Discussed:**

No

**Experimental Designs Or Analyses:**

Yes, the evaluation method looks consistent with prior data pruning work.

**Methods And Evaluation Criteria:**

- The authors did a thorough job of comparing their method to previous data pruning methods on the relevant benchmarks.

**Other Comments Or Suggestions:**

None

**Other Strengths And Weaknesses:**

- Overall I think this is a strong contribution that is clearly explained and evaluated.
- My main criticism is that the authors seem to emphasize the contribution of DUAL more than the contribution of the beta sampling, but looking at the performance numbers adding beta sampling to Dyn-Unc makes up most of the performance gain compared to adding the extra term that changes Dyn-Unc to DUAL. I think the authors should add Dyn-Unc + beta as a row in table 1 so that this is more clear. That result is present in the beta section and the appendix, but you should make it more clear what the benefit of each component is separately in table. I also feel in some parts of the text the importance of the beta sampling is a little understated.

**Questions For Authors:**

- How were the constants of the beta distribution determined? Is there an automatic way to do this or did you empirically try many possibilities?
- Have you looked at how correlated the particular examples selected by DUAL+beta at high pruning rates are with the examples selected by existing (but more complicated) methods like D2 and CCS? I think it would be interesting to add to your conclusion/discussion if you think these metrics are all getting at something similar from different angles or if selected example sets are more disjoint.

**Relation To Broader Scientific Literature:**

The authors discuss prior work well and evaluate their methods against prior methods.

**Theoretical Claims:**

N/A

---

> ### Author Rebuttal · Authors · 2025-04-01
>
> Thank you for your constructive review and insightful suggestion.
>
> Before we address your concerns regarding our beta sampling method, we would like to emphasize the novelty of the DUAL score lies in its time efficiency. Many existing pruning techniques require full training to estimate example difficulties, which makes the pruning process more expensive than training on the original dataset. Our DUAL can identify important samples more quickly, reducing the total time cost to less than that of a single training run. Beta sampling is applied to mitigate performance drops at high pruning ratios by selecting a larger portion of easier samples.
>
> ### **Details for Beta Sampling Design**
>
> - **The choice of Beta PDF.**
>
>     The domain of Beta distribution is [0, 1], which naturally aligns with the range of prediction means. As the PDF decreases at the tails, ensuring that samples with extreme scores have a negligible chance of being selected. While other distributions, e.g. Gaussian, could also be used for modeling, their support is $\mathbb{R}$, which means they can assign a non-negligible probability to values far outside the desired range unless the standard deviation becomes extremely small.
>
> - **The choice for hyperparameters of the Beta Distribution.**
>
>     When the pruning ratio is set to 0, $\alpha$ and $\beta$ is configured so that the mean of the Beta distribution ($\frac{\alpha}{\alpha + \beta}$) aligns with the prediction mean of the highest-scoring sample. This helps target high score samples at low pruning ratios.
>
>     To include easier samples at the high pruning ratios, we set parameters $\alpha$ and $\beta$ to depend on the pruning ratio. While BOSS changes them so that the Beta distribution's mode scales linearly with the pruning ratio $r$, we employ a non-linear scaling by raising $r$ to the power of $c_D$. This approach creates a PDF that is almost stationary at low pruning ratios and moves to the easier region polynomially as the pruning ratio increases.
>
>     The hyperparameter $c_D$ is chosen based on the relative complexity of the dataset. We assume that the larger the dataset and the more samples per class there are, the easier the whole dataset is. Higher $c_D$ decreases $\beta$ and thus increases the mean and decreases the variance of Beta distribution. For a more difficult dataset, easy data has to be sampled more, so $c_D$ should be larger (refer to Fig. 15 of App. C).
>
>     Recall that the variance of the Beta function is given by $\frac{\alpha \beta}{(\alpha + \beta)^2 (\alpha + \beta + 1)}$. According to our definition of the constant $C$ in Equation (5), where $C=\alpha+\beta$, increasing $C$ leads to a lower variance. This results in a more focused sampling distribution that concentrates on a specific region, improving the effectiveness of the sampling process by reducing unnecessary spread. The impact of the $C$ value is illustrated [here](https://vo.la/TftlTS).
>
>     In conclusion, we fix a moderate value of $C$ as 15 across all experiments, and tune only $c_D$ based on the guidelines outlined above, which is not extensive. Furthermore, as demonstrated in Section 4.4.1 of our paper, the choice of hyperparameters including $c_D$ remain robust across a wide range of values.
>
>
> ### **Comparison with Dyn-Unc + Beta Sampling**
>
> First, we clarify that the performance gain by adding a beta sampling is higher for DUAL, not for Dyn-Unc. Table 4 should be corrected as Dyn-Unc+beta (+25.66) and DUAL+beta (+31.51) for CIFAR-10.
>
> Beta sampling applied to Dyn-Unc utilizes the prediction mean at epoch 200 where Dyn-Unc is computed. Here, all samples are almost learned thus their prediction means become concentrated to 1. Therefore, Dyn-Unc with Beta sampling selects a much more easier samples, resulting in an inferior performance compared to DUAL.
>
> We illustrate selected samples [here](https://vo.la/NGuzVi). This figure clearly shows that Dyn-Unc+Beta actually selects easier samples. Furthermore, we test the Dyn-Unc+Beta for all pruning ratios and confirm that it is consistently worse than DUAL+Beta. The performance table can be found [here](https://vo.la/TAvedH).
>
> ### **Correlation between DUAL and D2, CCS**
>
> We appreciate this insightful suggestion. We visualize the selected subset by each method and examine the intersection [here](https://vo.la/BShVWs). CCS applies AUM with stratified sampling and a hard cutoff (pruning ratio, cutoff ratio): (70%, 20%), (80%, 40%), (90%, 50%). This approach completely excludes the most difficult samples while always retaining the easiest ones. D2 uses the forgetting score for graph-based sampling, applying the same hard cutoff as CCS. However, since it removes samples with the highest forgetting scores, it fails to eliminate the most difficult ones. DUAL selects uncertain and difficult samples, thereby excluding the hardest ones while simultaneously expanding coverage toward easier samples using Beta sampling.

---

### Official Review · Reviewer_3gz4 · 2025-03-14

**Overall Recommendation:** 3

**Summary:**

This paper proposed a dataset pruning score named as Difficulty and Uncertainty-Aware Lightweight (DUAL). The main idea is two fold. First, it combines the data difficulty and data uncertainty into one numerical measure for pruning. This extends existing work on uncertainty based data pruning such as Dyn-Unc (He et al. 2024) by simultaneously considering both difficulty and uncertainty. Second, to mitigate the accuracy degradation on high pruning ratio, the paper proposed to use Beta distributions fitted by the DUAL score as the way to determine the likelihood to prune the data instead of directly pruning data with worse DUAL score. A theoretical analysis was conducted to prove DUAL score can prone data more efficiently compared to Dyn-Unc based pruning strategy and extensive experimental evaluation on CIFAR-10, CIFAR-100, ImageNet-1k shows DUAL not only achieves high accuracy but also needs fewer training epoches (i.e. more efficient) and more robust compared to other state-of-the-art pruning methods.

**Claims And Evidence:**

1. One of the major issue of existing pruning metrics e.g. Forgetting score is that to get the pruning metrics itself require full dataset of training, which makes pruning itself become more computationally costly compared to training the image classification model itself. The proposed DUAL method showed that it can reduce the training time needed to calculate pruning score and simultaneously obtained state-of-the-art accuracy on the pruned dataset. Both theoretic and empirical evidence are provided to justify the claimed advantage of DUAL.
2. The proposed DUAL method still suffer from severe accuracy degradation when pruning ratio is high. The paper claims that such degradation can be effectively mitigated with Beta distribution based random sampling instead of hard thresholding. This is supported by experimental evaluation where DUAL + Beta-sampling consistently improve performance with DUAL only based pruning.
3. The proposed DUAL metric shows more robustness against label noise and image distortion. Experimental evaluation shows that DUAL method can better retain accuracy and identify noisy samples especially when pruning ratio is low (which means samples that are most likely to be pruned is also the noisy ones).
4. The paper also demonstrated the proposed DUAL score is generalizable across different model architectures. In particular, when a simpler model (e.g. ResNet18) is used for pruning the dataset, a complex model (e.g. ResNet50) trained on the pruned dataset can still achieve good performance.

**Essential References Not Discussed:**

As mentioned in previous question, the paper SIMS [2] should be cited and compared in the paper.

[2] Grosz, Steven, et al. "Data pruning via separability, integrity, and model uncertainty-aware importance sampling." International Conference on Pattern Recognition. Cham: Springer Nature Switzerland, 2024.

**Experimental Designs Or Analyses:**

The paper provided a thorough analysis on the effectiveness of DUAL. This experiment includes 1) accuracy obtained on pruned dataset by different methods under different pruning ratios (accuracy); 2) data pruning on dataset with artificially introduced label noise and image distortion (robustness); 3) cross architecture generalization where pruning model and classification model is different (generalization) Ablation study is provided to justify the importance of different hyperparameters and beta distribution fitting strategy. There are a few experiment results that need further explanation.
1. In Table 1, DUAL+\beta sampling consistently yield smaller variance. Is the variance computed over different sampling over the same beta distribution i.e. repeated line 1398 in Appendix C.2 algorithm 1 or different pruning runs altogether i.e. repeated Appendix C.2 algorithm 1? Is there any insights on why using beta sampling yield more stable results?
2. In Table 1 and 2, under high pruning ratio, using Beta sampling significantly improve the performance of DUAL, which is actually worse than many other methods. I'm wondering why beta sampling is so effective and if other distribution of sampling can be considered.
3. In Table 3, it is surprising that pruning using ResNet-18 achieved better performance than pruning using ResNet-50, given the classification model is actually ResNet-50. This seems to suggest pruning using simpler model is preferable. Or there are other causes for ResNet-50 to under perform such as lack of proper selection of hyperparameters? Typically if training with full dataset, ResNet-50 should outperform ResNet-18 on CIFAR-100. So even under pruning setting, I would expect ResNet-50 pruning and classification should still be better. Perhaps another baseline where pruning and classification is done by ResNet-18 can be added.
4. In Table 4, for CIFAR-10 the delta value in row 'Dyn-Und' and 'Ours' do not seem to be correct. Please verify.
5. In Table 4, for both CIFAR-10 and CIFAR-100, the improvement due to beta sampling is more significant for DUAL than Dyn-Unc. Is this expected?
6. I don't understand why each row of figure 2 has two plots. Figure 2(a) says 'score calculated at epoch T=60', so why there is another figure in the same row with T=90? Similarly, Figure 2(b) says 'score calculated at epoch T=90', why there is another figure in the same row with T=60?

**Methods And Evaluation Criteria:**

The paper utilized three different image classification datasets and conducted various experiments following the flow of pruning + classification. Mean accuracy and standard deviation are the primary metrics used to assess performance. And the metrics is measured under different pruning ratio. The paper also measured total training time vs. pruning ratio to demonstrate the efficiency of DUAL. While I believe the evaluation is comprehensive, it can be strengthened by evaluating on more challenging benchmark dataset such as iNaturalist [1], which exhibits more unbalanced and long-tail distribution of different image classes.

[1] Van Horn, Grant, et al. "The inaturalist species classification and detection dataset." Proceedings of the IEEE conference on computer vision and pattern recognition. 2018.

**Other Comments Or Suggestions:**

Typos:
- line 407 left column, the text says 'we vary it from 3 to 7', however, the figure 6 only shows results from 3 to 6.
- line 381 right column, 'asses' should be 'assess'.

**Other Strengths And Weaknesses:**

[Other strengths]
+ The DUAL score showing strength under noisy label and corrupted images is particularly encouraging especially under low pruning rate as it is non-uncommon to have small amount of polluted data samples in dataset so that we can use DUAL to prune a small number of samples and better
+ Figure 4 is a powerful demonstration of the efficiency of the proposed method as it considers all the time needed from pruning to training classifier. And under 30% pruning ratio, there is still about 15% reduction of total time spent. This is a significant data point I do not see in many other data pruning literature.
+ State-of-the-art accuracy achieved on CIFAR-10, CIFAR-100, and ImageNet-1K data pruning task.
+ A theoretical analysis that is also supported by empirical evidence on the efficiency of the pruning metrics.

[Other weaknesses]
- There is no qualitative results on the pruned and kept samples, which can provide some intuition on the data pruned and kept by DUAL.
- Two important ingredients: uncertainty and beta-sampling are heavily inspired by prior work. The uncertainty part uses the same formulation as Dyn-Unc. This may have put a dent on the overall novelty of the pruning metric.
- The dataset evaluated are small-scale. As suggested earlier, a medium/large scale and challenging dataset should be considered for demonstrating the effectiveness of DUAL.

**Questions For Authors:**

1. Since DUAL score use multiplication of the two measure i.e. difficulty and uncertainty, how will DUAL metric handle high difficulty and low uncertainty, or low difficulty and high uncertainty samples? In general, do we expect difficulty and uncertainty always correlate with each other?
2. Have the authors considered other ways to construct the beta distribution? Is there any insights on why the proposed form may be better than other design?
3. In line 146 right column, should 'bottom-right' be 'bottom-left'?

**Relation To Broader Scientific Literature:**

The paper has discussed and compared with a comprehensive collection of methods on data pruning. One of the main contribution of the paper is using beta-distribution based sampling to adjust pruning under high pruning ratio. Besides the BOSS paper mentioned in the paper, another paper shared similar idea is [2], which uses importance sampling as an ingredient for pruning. This paper should be compared in the paper.

[2] Grosz, Steven, et al. "Data pruning via separability, integrity, and model uncertainty-aware importance sampling." International Conference on Pattern Recognition. Cham: Springer Nature Switzerland, 2024.

**Theoretical Claims:**

The paper provides a theoretical claim that under certain assumptions, DUAL metric takes less iterations than Uncertainty-only metric to identify the data sample that needs to be pruned. A thorough analysis is provided to prove the theorem, which is mostly sound. The claim supported empirical observation where DUAL metric can achieve higher accuracy with fewer number of training iteration. I have is the theorem assumes a linear model for classifying the data sample. I'm curious if the authors have any insights on whether the analysis can be extended to nonlinear separable cases.

---

> ### Author Rebuttal · Authors · 2025-04-01
>
> We sincerely appreciate your insightful and valuable feedback. We address the given concerns and questions below.
>
>
> ### **1. Experiment with a more challenging dataset**
> Thank you for your suggestion. Due to time constraints, we experimented on a randomly sampled 20% subset of iNaturalist 2017 on ResNet-50, achieving 30.17% test accuracy. We applied random pruning and our DUAL+Beta method to prune 50% of this subset. Selected 50% subset is re-trained from scratch, and random pruning shows 12.6% test accuracy, while our method achieved 18.07%, showing its potential on challenging datasets.
> We additionally conducted experiments on the long-tailed CIFAR-10 and CIFAR-100, following [1]. Competitors are Random, EL2N, Dyn-Unc, and CCS for all pruning ratios. The table is attached [here](https://vo.la/UHJYgf).
>
>
> ### **2. Extension to nonlinear separable cases**
> The key insight of our analysis is that the prediction variance of uncertain samples is lower than that of easy samples at the early stage; so we use the prediction mean to diminish the score of easy ones.
> In the non-linear case, the decision boundary is likely more complex than a hyperplane. However, the gist of our approach should still hold: samples near the decision boundary would exhibit higher variance over training epochs, while those far from it should stabilize quickly.
> One possible extension is to analyze the behavior of the DUAL score in non-linear models, e.g., deep neural networks. One might focus on how the feature representation evolves over training and thus how uncertainty propagates. A potential approach is to generalize the method in the representation learning framework. Instead of tracking the prediction variance and mean, one could study these in an appropriate feature space, e.g., the penultimate layer of the network. Such extension is out of scope, but we consider this an interesting future direction.
>
>
> ### **3. Stability of DUAL+Beta**
> We clarify that it is different runs altogether, i.e., repeated Appendix C.2 Algorithm 1. Beta sampling selects easier samples that tend to be more typical, leading to more stable performance than selecting the most uncertain and difficult samples. This phenomenon is observed in other methods used with beta sampling (see Table 4 in our paper).
>
>
> ### **4. Regarding Beta Sampling**
> We provide a detailed explanation for beta sampling in our response to Reviewer YPaU.
>
>
> ### **5. Under-performance of ResNet-50**
> We initially used ResNet-18's training parameters due to time constraints. We retrained ResNet-50 with batch size of 256, SGD with 0.9 momentum and 5e-4 weight decay, and a 0.3 lr with 3 epochs warm-up. With these parameters, full dataset test accuracy shows 80.1%. We also add a row indicating the case from R18 to R18 [here](https://vo.la/avcZFx), reflecting your suggestion.
>
>
> ### **6. Clarification of Table 4 & Figure 2.**
> Sorry for the confusion. Delta value for Dyn-Unc is 25.66, and for Ours is 32.14.
> We provide corrected Figure 2 and a detailed explanation in the response to Reviewer TH5d.
>
>
> ### **7. Missed essential reference**
> The SIM score [2] is composed of three factors: Class Separability (cluster overlap), Data Integrity ($\ell_2$-norm of feature representations), and Model Uncertainty (prediction consistency across independent models). All factors are computed at epoch 20 using a snapshot approach. As a result, stabilizing the SIM score requires 10 independent runs, incurring high computational costs. In contrast, our method leverages training dynamics to calculate the DUAL score within a single run, requiring fewer epochs.
> SIMS also proposes a ratio-adaptive sampling strategy, applying importance weights over the original score distribution. However, it assumes a normal distribution of scores, which does not hold in practice (see Fig. 2 of [2]). In contrast, our sampling method, by not relying on any specific score distribution, remains robust across diverse datasets. We will add a citation of SIMS [2] and this comparison in the new version.
>
> ### **8. Qualitative results on the pruned and kept samples**
> Thanks for your suggestion. We visualize pruned and selected samples by DUAL on ImageNet-1K [here](https://vo.la/iCZkWL). DUAL tends to retain uncertain and challenging samples while discarding some typical instances and extremely difficult examples.
>
> ### **Other Questions**
> Q1) First, we emphasize that DUAL prioritizes samples with high uncertainty and high difficulty. Hence, the two cases you mentioned cannot be distinguished by the DUAL score alone. However, our Beta sampling favors samples with low difficulty and high uncertainty over those with high difficulty and low uncertainty.
>
> Q3) As in our response to Review TH5d, ‘bottom-right’ is correct.
>
> ---
> [1] Learning Imbalanced Datasets with Label-Distribution-Aware Margin Loss, NeurIPS 2019.
>
> [2] Data pruning via separability, integrity, and model uncertainty-aware importance sampling, ICPR. Cham: Springer Nature Switzerland 2024.

---

### Decision · Program_Chairs · 2025-05-01

**Decision:**

Accept (poster)

**Comment:**

This paper introduces a lightweight dataset pruning framework based on the proposed DUAL score, which combines example difficulty and prediction uncertainty to identify informative samples early in training, significantly reducing computational overhead compared to existing approaches that require full training cycles. The authors provide both theoretical justification and extensive empirical validations, showing that DUAL matches the state-of-the-art accuracy under various conditions—including label noise, image corruption, and cross-architecture generalization.  While the core ideas are rooted in extensions of earlier work such as Dyn-Unc, the integration and practical improvements are notable. I recommend it for acceptance.